# ANALYTICA: SOFT PROPOSITIONAL REASONING FOR ROBUST AND SCALABLE LLM-DRIVEN ANALYSIS

**Junyan Cheng**
Dartmouth College
jc.th@dartmouth.edu

**Kyle Richardson**
Allen Institute for AI
kyler@allenai.org

**Peter Chin**
Dartmouth College
pc@dartmouth.edu

## ABSTRACT

Large language model (LLM) agents are increasingly tasked with complex real-world analysis (e.g., in financial forecasting, scientific discovery), yet their reasoning suffers from stochastic instability and lacks a verifiable, compositional structure. To address this, we introduce **Analytica**, a novel agent architecture built on the principle of **Soft Propositional Reasoning (SPR)**. SPR reframes complex analysis as a structured process of estimating the soft truth values of different outcome propositions, allowing us to formally model and minimize the estimation error in terms of its bias and variance. Analytica operationalizes this through a parallel, divide-and-conquer framework that systematically reduces both sources of error. To reduce bias, problems are first decomposed into a tree of subpropositions, and tool-equipped LLM *grounder agents* are employed —including a novel Jupyter Notebook agent for data-driven analysis—that help to validate and score facts. To reduce variance, Analytica recursively synthesizes these grounded leaves using robust linear models that average out stochastic noise with superior efficiency, scalability, and enable interactive "what-if" scenario analysis. Our theoretical and empirical results on economic, financial, and political forecasting tasks show that Analytica improves 15.84% accuracy on average over diverse base models, achieving 71.06% accuracy with the lowest variance of 6.02% when working with a Deep Research grounder. Our Jupyter Notebook grounder shows strong cost-effectiveness that achieves a close 70.11% accuracy with 90.35% less cost and 52.85% less time. Analytica also exhibits highly noise-resilient and stable performance growth as the analysis depth increases, with a near-linear time complexity, as well as good adaptivity to open-weight LLMs and scientific domains.

## 1 INTRODUCTION

Capable LLM agents require foresight: the ability to form, update, and act on probabilistic forecasts of future states. For example, effectively answering open-ended questions in domains like experimental science or financial forecasting (e.g., *What is the best way to improve the performance of my model on task Y?* or *What is the best strategy to invest in $NVDA this year?* in Fig. 1) involves predicting the future state of the world via complex information gathering, case analysis, and explicit uncertainty estimation. While considerable progress has been made recently through the development of new large reasoning models (Jaech et al., 2024; Guo et al., 2025; Comanici et al., 2025) and deep research architectures (Xu & Peng, 2025; OpenAI, 2025) that explicitly encourage deep analysis through test-time scaling, such approaches fundamentally rely on free-form text reasoning, which often lacks the precision and reliability needed for decision making in many critical areas.

In this paper, we investigate an alternative framework called **Soft Propositional Reasoning (SPR)** that reframes complex LLM-driven analysis as a structured process of assigning a *soft truth value* or *degree of belief* (Huber et al., 2009) to different possible outcomes. For example, answering the query in Fig. 1 can be done through deep case analysis on specific outcomes such as *Long $NVDA and hold for the year is the best* and by decomposing this root hypothesis into testable sub-propositions that can be grounded to real-world data (e.g. via further information gathering and experimentation) and scored for correctness. Key to our approach is that the degrees of belief (e.g., 0.7 for hypothesis 1 in Fig. 1) are computed compositionally from such evidence, which aims

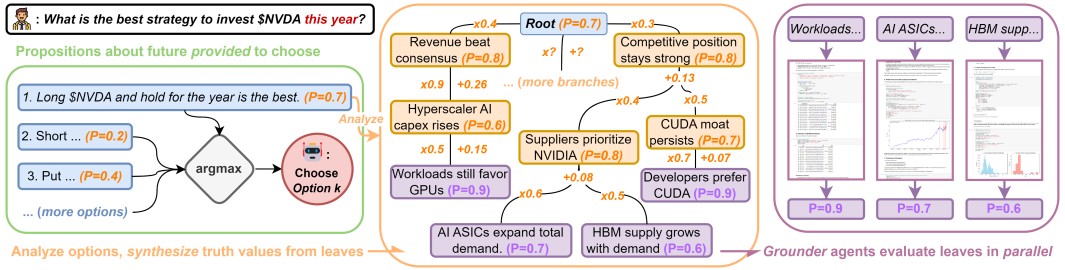

Figure 1: Given a complex query (e.g., forecasting $NVDA), Analytica selects the most plausible outcome by estimating the "soft truth value" of each provided competing proposition (Green box). The analysis process begins when an *analyzer* agent decomposes a proposition into a tree of sub-propositions (Orange box), terminating is a set of testable leaf nodes. Next, *grounder* agents, such as a Jupyter Notebook agent mimicking a human analyst, evaluate the leaves (Purple box) and assign soft scores that reflect the evidence for each leaf. Finally, a synthesis stage recursively aggregates these scores up the tree (middle) to compute a final score for the root proposition.

to strike a balance between pure text-based reasoning and traditional relational and probabilistic approaches to AI (De Raedt et al., 2007; Richardson & Domingos, 2006; Koller & Friedman, 2009).

We investigate the SPR framework through a new LLM-agent architecture called **Analytica** that employs a highly parallel, three-stage divide-and-conquer strategy. As illustrated in Fig. 1, a given hypothesis is first automatically decomposed into a tree of sub-hypotheses through an *analysis stage*, which, by design, terminates in a set of testable leaf nodes or hypotheses. This is followed by a *grounding stage*, in which tool-equipped LLM agents validate and score the leaf hypotheses through further search and experimentation. For example, our most powerful *grounder agents* simulate human analysts by working with the **Jupyter notebook** environments that facilitate web-based and data-driven analysis (e.g., via research APIs), generic code writing in Python (e.g., for running simulations), and report writing (e.g., using markdown blocks). The scores of leaf nodes are then recursively propagated up to the root through a *synthesis stage* and aggregation function $f$. For example, our best synthesis strategy involves taking a **linear combination** of model-produced confidences coupled with additional linear coefficients (as illustrated in the tree edges in Fig. 1), which we show through first principles helps average out stochastic noise and minimize forecast variance.

We empirically test our approach on 736 real-world economics and financial forecasting challenges, which naturally take the form of true/false proposition prediction (e.g., making yes/no long-short equity predictions in financial markets and future predictions in polymarkets) and have recently been shown to be a promising testbed for evaluating the general forecasting and reasoning abilities of LLMs (?Schoenegger & Park, 2023; Tan et al., 2024; Zeng et al., 2025; Paleka et al., 2025, *inter alia*). Compared with several text-based reasoning baselines, including advanced variants of chain-of-thought (Wei et al., 2022; Yao et al., 2023; Besta et al., 2024; Bi et al., 2025), as well as the deep research agent of OpenAI (2025), our best variant of Analytica achieves an average 15.84% improvement in end-task prediction accuracy. Analytica with Jupyter Notebook agents in particular demonstrates strong cost-effectiveness, reaching the near-highest accuracy of 70.11% with 90.35% less budget and 52.85% less time. Furthermore, Analytica displays impressive scalability, handling exponential growth in analytical complexity (e.g., 54x more nodes) with only a near-linear rise in computation time (12x), while the performance shows a stable improvement over the analysis depth, highlighting the high practicality and potential of our proposed framework. We also show how Analytica exhibits good adaptivity to smaller open-weight models as well as other domains such as scientific claim verification (Jansen et al., 2025a). Our code and data are available at `https://github.com/chengjunyan1/analytica`.

## 2 RELATED WORK

**Structured Reasoning in LLMs** Our work takes inspiration from the large literature on modular and decomposition-based reasoning architectures (Andreas et al., 2016; Khot et al., 2021; 2023;

Talmor & Berant, 2018; Zhou et al., 2022, *inter alia*), as well as more recent variants of chain-of-thought reasoning (Wei et al., 2022; Yao et al., 2023; Besta et al., 2024; Yang et al., 2024; Aytes et al., 2025) and deep research agents (OpenAI, 2025; Xu & Peng, 2025) all of which aim to improve the robustness and scalability of neural reasoning through problem decomposition, test-time scaling (Snell et al., 2024) and tool use. As discussed above, however, much of this work operates mostly in a discrete text space, whereas Analytica focuses on reasoning in a soft propositional space and attempts to integrate model confidences more directly into the process of aggregating reasoning paths (see Cao et al. (2023)) and quantifying an agent's degree of belief (see Chen et al. (2024)).

**LLM Agents for Real-world Analysis** We also focus on the growing body of work using LLM agents to tackle a wide range of open-ended analysis tasks, such as societal dynamics (Cheng & Chin, 2024a), financial forecasting (Yu et al., 2024), economic mechanism design (Karten et al., 2025), crypto trading (Li et al., 2024), predictive markets (Halawi et al., 2024), general data analysis (Majumder et al., 2025; 2024), automated scientific discovery (Lu et al., 2024; Gottweis et al., 2025; Jansen et al., 2025b; Cheng et al., 2025), among others. While our overall analysis framework is domain agnostic, we focus on forecasting problems in economics, finance, and politics due to their high uncertainty, difficulty, and richness of data (Zou et al., 2022; Chen et al., 2023; Tan et al., 2024; Karger et al., 2024; Wildman et al., 2025).

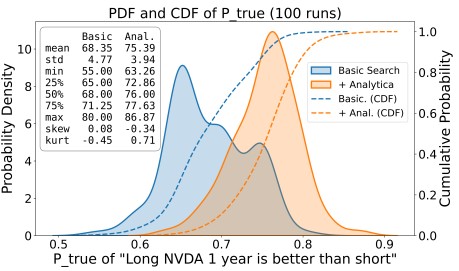

Figure 2: An illustration of estimation variance and bias. Analytica with a linear rule has lower bias (closer to the ground truth of 1) and variance. Hitting a better trade-off.

**Hybrid LLM Reasoning** Finally, our approach relates to many recent attempts to enhance the reasoning power of LLMs with classical relational and probabilistic methods Olausson et al. (2023); Pan et al. (2023); Li et al. (2025); Cheng et al. (2023), often by integrating symbolic solvers into the reasoning pipeline or using LLMs to produce symbolic representations. Rather than directly incorporating explicit solvers into our reasoning pipeline, we instead follow other work in neuro-symbolic modeling on distilling model behavior to classical models (e.g., tractable probabilistic models, PGMs) (Zhang et al., 2024; Qiu et al., 2025; Feng et al., 2025; Dohan et al., 2022; Cheng & Chin, 2024b), in our case, interpreting LLM and agent outputs as if-then structures that we reason over using soft and noisy relaxations of both model beliefs and the logical operators used to combine beliefs.

## 3 SOFT PROPOSITIONAL REASONING

The objective of a **soft proposition reasoning (SPR)** is to accurately estimate the soft truth value of a complex proposition, $p_{true}^{gt}$. A robust agent is one that minimizes the expected error of this estimate. To formalize this, we consider the *Mean Squared Error (MSE)* of the forecast, which is the expected squared difference between the estimate and the ground truth value:

$$\text{MSE}(p_{true}) = E\left[(p_{true} - p_{true}^{gt})^2\right] = \underbrace{\left(E[p_{true}] - p_{true}^{gt}\right)^2}_{\text{Bias}^2} + \underbrace{E\left[(p_{true} - E[p_{true}])^2\right]}_{\text{Variance}} \quad (1)$$

The expectation $E[\cdot]$ is taken over the randomness in the agent's reasoning process (e.g., model sampling stochasticity, variations in tool outputs). This total error can therefore be standardly decomposed into two distinct sources: bias and variance.

Accordingly, a *robust analysis* must systematically minimize both bias and variance. The *compositional nature* of complex problems from SPR provides a foundation to address this challenge, which assumes that the truthfulness of a complex proposition is recursively supported by a set of child propositions, e.g., the truth of "NVIDIA's revenue will beat consensus" is a function of its underlying evidential drivers (e.g., "AI capex is rising"), as depicted in Fig. 1. That is, $\rho_p \cdot p_{true} = f(\rho_{c_1} \cdot p_{true}, \ldots, \rho_{c_n} \cdot p_{true})$. The *synthesis rule* can be a flexible and arbitrary function $f : [0, 1]^n \to [0, 1]$. We develop the Analytica architecture based on SPR(§ 4) where $Bias$ is mitigated by reducing the original complex query to simple leaves which are relatively simple to process

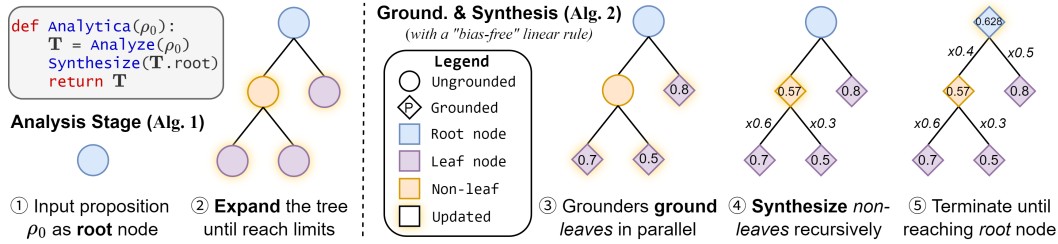

Figure 3: Illustration of Analytica. First, in the **Analysis Stage (Alg. 1)**, a root proposition is recursively decomposed into a tree of sub-propositions (Steps 1-2). This is followed by the **Grounding and Synthesis Stages (Alg. 2)**, a bottom-up process where (Step 3) Grounder agents evaluate all leaf nodes in parallel to assign soft truth values, and (Steps 4-5) a Synthesizer recursively aggregates these grounded values up the tree until a final, robust estimate for the root is computed.

by the powerful **Grounder** agents; $Variance$ is reduced during synthesis, where an **Analyzer** and **Synthesizer** work in concert with a robust *linear* synthesis rule to average out the stochastic noise from many subproblems, ensuring stable error propagation. Fig. 2 shows that Analytica effectively decreases both variance (tighter distribution) and bias (mean closer to the ground truth).

**Comparison with CoT and its variants** This results in a recursive, divide-and-conquer strategy for problem solving, which differs from existing structured reasoning methods that center around linear reasoning paths. In the standard Chain-of-Thought (CoT) (Wei et al., 2022), the model generates a linear sequence of tokens $R = \{r_0, \ldots, r_n\}$ to derive a final output. Advanced approaches like Tree-of-Thoughts (ToT) (Yao et al., 2023) and Graph-of-Thoughts (GoT) Besta et al. (2024) search for an optimal path $R^*$ by maximizing a heuristic LLM-based valuation function $V(R)$:

$$\hat{y} = f_{LLM}(x, R^*) \quad \text{where} \quad R^* = \arg\max_{R \in Paths} V(R).$$

where $f_{LLM}$ is the call to an LLM generation, Forest-of-Thought (FoT) (Bi et al., 2025) further extending this by aggregating results from multiple trees, i.e. $\hat{y} = Aggr(\{f_{LLM}(x, R_i^*)\}_{i=0}^K)$, which is conceptually related to our synthesis mechanism. However, instead of aggregating different reasoning paths for the *same* problem, we aggregate results from *different subproblems recursively*, i.e., $\hat{y} = Aggr(\{\hat{y}_{C_i}\}_{i=0}^M)$. Here, $\hat{y}_{C_i}$ denotes child subproblems $C_i$ that are generated via analyzers; their results are aggregated from solutions of their own children in the same fashion recursively, until reaching the leaves, which are solved by our grounder agents.

## 4 ANALYTICA

Based on the SPR framework, we introduce **Analytica**, an architecture for complex analysis and forecasting. An overview of the Analytica architecture is provided in § 4.1. Subsequently, we explain how it minimizes both estimation bias and variance in § 4.2. Finally, we discuss the robustness and efficiency of Analytica in § 4.3 and § 4.4, respectively.

### 4.1 OVERVIEW

Analytica employs a divide-and-conquer strategy, operationalizes SPR through three core components: an *Analyzer* $A_A$, which expands a proposition tree or single root proposition with new nodes or branches. *Grounder* $A_G$, which determines the soft truth values of leaves and produces a report; and a *Synthesizer* $A_S$, which combines reports and soft truth value from fully-grounded children to deduce the report and $p_{true}$ of their parent. Analytica consists of two algorithms: `Analyze` (Alg. 1) and `Synthesize` (Alg. 2). Illustrated in Fig. 3, it calls `Analyze` to expand the tree initialized with the root proposition $\rho_0$, then passes the root to `Synthesize` to ground the entire tree bottom-up. Details of each component are provided below:

**Analyzer** The analyzer agent $A_A$, expands a proposition tree $\mathbf{T}$ to an expanded tree $\mathbf{T}'$: $A_A : \mathbf{T} \to \mathbf{T}'$. It begins with a tree consisting only of a root query proposition. The agent is then

| **Algorithm 1** Analyze($\rho_0, A_A, L_{max}, T_{max}$) | **Algorithm 2** Synthesize($\rho_i, A_G, A_S$) |
|---|---|
| **Require:** Proposition $\rho_0$, Analyzer LLM $A_A$, Max leaves $L_{max}$, Max steps $T_{max}$ 
 **Ensure:** Proposition tree $\mathbf{T}$ rooted on $\rho_0$ 
 1: $\mathbf{T} \leftarrow$ InitializeTree($\rho_0$) 
 2: **for** $t = 1, \ldots, T_{max}$ **do** 
 3:      **if** NumberOfLeaves($\mathbf{T}$) $\geq L_{max}$ **then** 
 4:          **break** 
 5:      $\mathbf{P}_{new} \leftarrow A_A(\mathbf{T})$        ▷ Expand tree 
 6:      $\mathbf{T} \leftarrow$ Update($\mathbf{T}, \mathbf{P}_{new}$) 
 7: **return** $\mathbf{T}$ | **Require:** Proposition node $\rho_i \in \mathbf{T}$, Grounder LLM $A_G$, Synthesizer LLM $A_S$ 
 **Ensure:** Grounded $\rho_i$ with $p_{true}$ and report 
 1: **if** $\rho_i$ is a leaf **then** 
 2:      $\rho_i.report, \rho_i.p_{true} \leftarrow A_G(\rho_i)$ 
 3: **else** 
 4:      **for all** $\rho_{ij} \in \rho_i.children$ **do in parallel** 
 5:          $\bar{\rho}_{ij} \leftarrow$ **async** Synthesize($\rho_{ij}$) 
 6:      $\rho_i.report, \rho_i.p_{true} \leftarrow A_S(\rho_i.children)$ 
 7: **return** $\rho_i$ |

prompted to progressively deepen the analysis by adding independent child nodes to one or multiple existing nodes, repeating until either a completion signal is reached or a predetermined maximum leaf count is exceeded.

**Grounder** The Grounder agent, $A_G$, grounds a *leaf* proposition $\rho_{leaf}$ by estimating $p_{true}$ with a report: $A_G(\rho_{leaf}) \rightarrow \bar{\rho}_{leaf}$. We study three variants of the Grounder: 1) **Basic Search** agent that relies on standard web search to gather evidence; 2) OpenAI **Deep Research** (OpenAI, 2025) that extensively searches the internet to compile a report for the query; and 3) **Jupyter Notebook**, our most advanced hybrid Grounder that mimics professional data analysts by iteratively writing, executing, and debugging Python and markdown blocks in a Jupyter notebook environment with access to various search and financial APIs. Jupyter agents operate as follows. Upon receiving an input query, agents are instructed to repeatedly produce interleaved markdown cells for qualitative reasoning and Python cells for programmatic execution at each step. Similar to ReACT (Yao et al., 2022), these cells are executed by the Jupyter backend and outputs are returned to the agent, which then decides whether to continue or to terminate the notebook. If an error arises, the agent must correct it before proceeding. Upon termination, the agent is prompted to compile the entire session into a final report and produce a soft truth value ($p_{true}$). More details and examples found in § C.4.

**Synthesizer** The Synthesizer agent, $A_S$, then grounds, or scores, a *non-leaf* proposition $\rho_i$ based on the scores of its children $\rho_i.children = \{\bar{\rho}_{i0}, \bar{\rho}_{i1}, ...\}$. Formally, $A_S(\rho_i, \rho_i.children) \rightarrow \bar{\rho}_i$ where $\bar{\rho}_i$ contains the truth value $\rho_i.p_{true}$ and a report. We employ a **Linear** synthesis rule:

$$\rho_i.p_{true} = \beta_0 + \sum_j \beta_j \cdot \bar{\rho}_{ij}.p_{true}, \quad \text{where } |\beta_j| < 1, |\beta_0| < c, and \; \rho_i.p_{true} \in [0, 1] \quad (2)$$

which resembles factor-based models widely adopted in economics and political science (Fama & French, 2015; Gregg & Banks, 1965). The LLM is tasked with outputting coefficients $\beta_j$, and an intercept $\beta_0$ in a JSON format as detailed in § E.2 and § D.8, where $c$ restricts the intercept from surpassing the impact of children. In § B, we show how the **computation graphs** produced by this process can be modeled as a special type of linear Bayesian network, which gives insights into the semantics of synthesis (e.g., the assumptions made about the relationships between sub-claims) and suggests other scoring strategies (e.g., using techniques from PGMs (Koller & Friedman, 2009).

To discover the characteristics of ideal synthesis, we study two alternative synthesis rules: a) a **Vanilla** rule, which calls LLM to directly output a $p_{true}$ with a report; and b) a **Simple logic** strategy, which prompts the LLM to generate a logical formula that connects the soft truth values of *all* children through fuzzy logical operators (Van Krieken et al., 2022; Grespan et al., 2021): $A$ AND $B = A \times B$, $A$ OR $B = A + B - A \times B$, and NOT $A = 1 - A$, and an "assumption" variable, $P_A \in [0, 1]$, to account for external factors, e.g., $P_i = P_{i1}$ OR (NOT $P_{i2}$ AND $P_A$), where $P_{ij}$ denotes the $j$-th child of proposition $i$ (see more examples in § D.8).

**Resynthesis** The *locality* inherent in the synthesis process, where each synthesizer accesses only a specific node and its children, facilitates Analytica's efficient **scenario analysis** for addressing "what-if" inquiries, which is highly useful in practice. After a tree is fully grounded, users can manually edit the truth value, statements, or reports of any node, or add/remove nodes to explore

a counterfactual (e.g., "*What if* inflation does *not* slow down?"). Instead of reexecuting the entire Analytica process, the system triggers a fast recomputation, calling the synthesizer to update only the *affected branches* up to the root. This allows for a rapid and interactive exploration of how varying assumptions affect the final outcome (see example in Fig. 16).

## 4.2 DERIVATION FROM FIRST PRINCIPLES

Analytica is designed so that most effort is dedicated to verifying the leaves, and soft truth values of non-leaves are *linearly* composed from the children. The subsequent reasoning step, where soft truth values of non-leaves are linearly composed from the children, acts as a highly lightweight, "effectively free" mathematical wrapper that aggregates these grounded values. Under Eq. 1, we can show that such a strategy can be derived from first principles. We model the ground truth $p_{true}^{gt}$ of the root proposition as a linear combination of its $k$ *leaves*: $p_{true}^{gt} = \beta_0' + \sum_{i=1}^{k} \beta_i' l_{i,true}^{gt}$. For analytical purposes, this expression is derived by algebraically expanding the nested linear equations from the root to the leaves. Each coefficient $\beta_i'$ represents the cumulative impact of a leaf on the root, effectively forming a *beta* path: the product of all local $\beta$ coefficients along the unique path through the tree from root to leaf $l_i$. Similarly, $\beta_0'$ is the aggregated intercept of all non-leaves. The estimated $p_{true}$ can be written as a similar linear composition of leaves: $p_{true} = \beta_0' + \sum_{i=1}^{k} \beta_i' l_{i,true}$. Each leaf estimate $l_{i,true}$ is a random variable characterized by its own bias and variance. We now derive the bias and variance of the final root estimate $p_{true}$ as a function of its components.

**Bias**  The bias of the root estimate is a weighted sum of the biases of the individual leaf estimates:

$$\text{Bias}(p_{true}) = E[p_{true}] - p_{true}^{gt} = \sum_{i=1}^{k} \beta_i' \left( E[l_{i,true}] - l_{i,true}^{gt} \right) = \sum_{i=1}^{k} \beta_i' \text{Bias}(l_{i,true})$$

The bias decreases in two ways. 1) **Simplified leaves**: as the analysis deepens, we *hypothesize* that the leaf nodes will gradually approach simple atomic propositions whose truthfulness is easy to judge. This makes the weighted summation of the leaf biases smaller than the bias of directly evaluating the root. More formally, we note the root bias as $\text{Bias}(root)$ and *assume* that when $\text{Bias}(l_{i,true}) = \delta_i \text{Bias}(root)$, where $0 < \delta_i < 1$ for all leaves $i$, then: $\text{Bias}(p_{true}) = \sum_{i=1}^{k} \beta_i' \text{Bias}(l_{i,true}) = \sum_{i=1}^{k} \beta_i' \delta_i \text{Bias}(root) = \text{Bias}(root)(\sum_{i=1}^{k} \beta_i' \delta_i) < \text{Bias}(root)$. 2) The use of **powerful grounders** helps to further reduce bias, as empirically supported by Table 2 and Fig. 6. This forms the basis for the strategy of employing an Analyzer to achieve a detailed breakdown of the complex query proposition, combined with an emphasis on utilizing strong grounder agents to manage leaf propositions, such as our sophisticated Jupyter Notebook grounder.

**Variance**  The variance of the root estimate is a function of the leaf variances and their covariance:

$$\text{Var}(p_{true}) = \sum_{i=1}^{k} \beta_i'^2 \text{Var}(l_{i,true}) + \sum_{i \neq j} \beta_i' \beta_j' \text{Cov}(l_{i,true}, l_{j,true}) \xrightarrow{k \to \infty} 0$$

It is minimized by: 1) **Granular decomposition**, the leaf variances are suppressed by the squared weights ($\beta_i'^2$), which approach 0 as the leaf number grows; and 2) **Ideal analysis**, generating children with minimal covariance, where the analyzer is forced to uncover independent factors in a *top-down, divide-and-conquer* manner in Analytica. This theoretical insight aligns with our empirical findings, where the prediction accuracy grows with the size of the proposition tree (Fig. 4) and the low variance of our method (Table 2). It also guides us to highly value system scalability, which is crucial for not only practical application but also results in reduced estimation variance.

## 4.3 ROBUSTNESS OF ANALYTICA AND IDEAL SYNTHESIS

We now analyze the robustness of Analytica under the linear rule, and then generalize it to the principles of *ideal synthesis* to delve deeper into the criteria necessary for achieving optimal performance. The synthesis rule is crucial as it averages the variances of the leaves, and thus must be robust against noise in the leaf estimates to preserve the stability gains. This is fundamentally based on its mathematical structure. To analyze this, let a synthesis rule be a function $P = f(C_1, \ldots, C_n)$ that maps child truth values $\{C_j\}$ to a parent value $P$. We assume the grounder produces noisy

|  | Basic | Analytica | Analytica$^2$ | Analytica$^3$ | Analytica$^4$ |
|---|---|---|---|---|---|
| Avg. time (s) | 0.5 | 5.3 (1.0x) | 16.4 (3.1x) | 33.3 (6.3x) | 63.5 (12.0x) |
| # Nodes | - | 19.9 (1.0x) | 68.1 (3.4x) | 359.5 (18.1x) | 1075.3 (54.0x) |
| # Tokens | 3.6K | 58.6K | 169.4K | 929.0K | 2.8M |

Table 1: Scalability of recursive Analytica. As the recursion depth increases, the number of nodes and tokens grows exponentially, while the average computation time increases near-linearly.

estimates $\hat{C}_j = C_j + \epsilon_j$, where $\epsilon_j$ is a random error term. The rule's sensitivity to this input noise can be measured by its partial derivatives $\frac{\partial f}{\partial C_j}$. The Linear rule demonstrates a superior stability:

> **Proposition 1** (Constant Sensitivity of the Linear Rule). *The Linear synthesis rule, $P = \beta_0 + \sum_{j=1}^{n} \beta_j C_j$, has a constant sensitivity to input noise given by the partial derivative: $\frac{\partial P}{\partial C_j} = \beta_j$ that ensures stable and bounded error propagation, independent of other inputs.*

The formal proof is detailed in § A.1, which identifies a set of conditions for an ideal synthesis rule: 1) **Bounded Sensitivity:** The function's partial derivatives with respect to its inputs should be bounded and preferably small, preventing any single input from having an outsized, unpredictable impact; 2) **A Smoothing Property:** The function should have a natural averaging effect that inherently dampens or smooths noise from its inputs, rather than propagating it; and 3) **Graceful Degradation:** The function should be smooth and continuous, without sharp "tipping points" or cliffs where a small perturbation can cause disproportionate volatility. The linear rule satisfies all three conditions, providing a strong theoretical explanation for its superior empirical performance over others in terms of accuracy, stability (Table 2), and noise resistance (Fig. 5).

## 4.4 EFFICIENCY AND SCALABILITY OF ANALYTICA

The theoretical benefits of scaling up the depth of the analysis, as discussed in § 4.2, are attainable in practice only if the architecture is capable of efficiently supporting a considerable number of leaves. Analytica allows *unbounded scaling by recursively invoking itself at leaves* with each leaf serving as a proposition that can act as a new root for another Analytica analysis. We denote it Analytica$^n$, where $n$ indicates the depth of the recursion. Recursive invocation results in a *tree-level locality*, where each instance of Analytica concentrates on a segment of the ultimately expanded tree, which may exceed the limit for a single Analyzer to produce. The locality of synthesizers, grounders, and Analytica itself facilitates *massive parallelism*, which shows a near-linear time complexity with respect to the depth of the analysis, as shown in Table 1 and formally proved in § A.2.

## 5 EMPIRICAL VALIDATION

|  |  |  | **Accuracy** |  |  | **Stability** |  | **Efficiency** |  |
|---|---|---|---|---|---|---|---|---|---|
|  | **Accu.** | *Imp.* | **Soft** | **Hard** | **BS** | **Conf.** | **Var** | **Cost** | **Time** |
| *Random* | *48.10* | *-* | *48.32* | *47.11* | *33.92* | *74.70* | *48.53* | *-* | *-* |
| **Basic Search** | 53.94 | - | 51.12 | 53.92 | 26.73 | 64.95 | 10.30 | $0.02 | 0.54m |
| + Tree of Thgt. | 60.19 | *11.59* | 55.74 | 57.51 | 26.46 | 76.89 | 9.21 | $0.28 | 6.55m |
| + Graph of Thgt. | 57.88 | *7.30* | 53.52 | 57.18 | 26.85 | 75.23 | 10.12 | $0.18 | 4.72m |
| + Forest of Thgt. | 60.73 | *12.59* | 56.87 | 57.64 | 26.44 | 78.35 | 8.28 | $0.55 | 10.32m |
| + Analytica-V | *63.18* | *17.13* | 56.56 | 59.37 | 26.33 | 85.44 | 10.89 | $0.24 | 5.42m |
| + Analytica-S | 57.61 | *6.80* | 53.82 | 56.70 | 26.36 | 74.99 | 7.45 | $0.23 | 5.38m |
| **+ Analytica-L** | **65.62** | *21.65* | **58.51** | **60.13** | 24.21 | **85.56** | 6.46 | $0.26 | 5.49m |

Table 2: Performance, stability, and efficiency results across different Analytica setups and comparisons with structured reasoning approaches. Bold/underline indicates best/second. "Imp." means improvement. 'V', 'S', and 'L' denote the vanilla, simple logic, and linear rules, respectively.

In this section, we empirically validate the core theoretical claims of the Analytica framework presented in § 4 through three key research questions (RQs). **RQ1: Bias and Variance Reduction (§ 5.2).** We hypothesize that Analytica minimizes bias, while its linear synthesis minimizes variance (§ 4.2). We test this by comparing accuracy uplifts and stability metrics against baselines across forecasting tasks (Table 2,3); **RQ2: Scalability and Robustness (§ 5.3).** We hypothesize that performance improves with analysis depth while maintaining efficiency due to recursive parallelism (§ 4.4). We examine this by tracking how accuracy scales as the number of nodes grows (Fig. 4). We further hypothesize that the Linear rule provides stronger robustness to noise than the simple logic rule (§ 4.3, Prop. 1). We test this via a noise-injection stress experiment (Fig. 5); **RQ3: Cost-Effectiveness (§ 5.4).** We study the practical usefulness and trade-offs between reasoning capability and costs. We illustrate this via efficiency frontier plots (Fig. 6). Additional results, including domain-specific breakdowns and model ablations, are provided in § D.

| | Accu. | *Imp.* | Soft | Hard | BS | Conf. | Var | Cost | Time |
|---|---|---|---|---|---|---|---|---|---|
| **Deep Research** | 63.04 | - | 57.22 | 59.31 | 26.24 | 82.57 | 9.28 | $4.02 | 7.60m |
| + Analytica-V | 69.16 | *9.71* | 59.26 | 65.16 | 22.77 | 83.41 | 9.88 | $12.70 | 30.07m |
| + Analytica-S | 66.30 | *5.17* | 58.79 | 63.71 | 24.15 | 76.34 | 7.27 | $13.70 | 29.90m |
| + **Analytica-L** | **71.06** | *12.72* | 60.01 | 66.57 | 22.79 | 83.59 | **6.02** | $14.10 | 30.01m |
| **Jupyter NB** | 61.96 | - | 56.92 | 62.67 | 26.90 | 76.68 | 12.28 | $0.07 | 2.61m |
| + Analytica-V | 68.89 | *11.18* | **61.57** | 67.40 | **21.67** | 80.75 | 12.90 | $1.05 | 13.98m |
| + Analytica-S | 62.77 | *1.31* | 57.19 | 64.48 | 25.71 | 77.28 | 8.65 | $1.25 | 13.81m |
| + **Analytica-L** | 70.11 | *13.15* | 60.25 | **68.01** | 22.89 | 81.10 | 7.28 | $1.36 | 14.15m |

Table 3: Ablation on the advanced grounders and comparison to Deep Research.

## 5.1 EXPERIMENT SETUP

**Dataset** The agent is tasked with evaluating a collection of propositions related to potential outcomes of an upcoming real-world event. A dataset comprised **736 unique events** derived from the predictive and financial markets was compiled. Events were carefully filtered to ensure they were resolved after our model's knowledge cut-off. The **Financial Market Tasks** involve making a one-year "long vs. short" prediction for an asset (like stocks, indices, commodities), necessitating high-level strategic thinking rather than short-term speculation. The **Predictive Market Tasks** directly use the options provided by the market, e.g., for "who will win the 2024 US presidential election?", the two options are Kamala Harris and Donald Trump. For each task, the agent receives the event description, the current date, and the target proposition (e.g., "The best strategy for $NVDA over the next year is to go long") for each option in the event. The agent must provide $p_{true}$ for each given proposition that corresponds to the options in an event. For more information, refer to § C.2.

**Baselines** We structure our comparisons into two components. First, we evaluate the *standalone base agents* (**Basic Search**, **Deep Research** (OpenAI, 2025), **Jupyter Notebook**) that operate directly on the root query. Second, we evaluate *reasoning frameworks* that use these same agents as subroutines (grounders). We compare Analytica with Tree/Graph/Forest of Thoughts (Yao et al., 2023; Besta et al., 2024; Bi et al., 2025), which are implemented over Basic Search for fairness, as well as against a random baseline. All experiments use the `o3-2025-04-16` model with a knowledge cutoff of June 01, 2024. For ablation studies in other base models, see § D.4. A low temperature of 0.1 was used following Cheng & Chin (2024a). The web search is powered by `Exa.ai`. We also set a limit of 10 leaves for Analytica. See § C.1 for further details.

**Evaluation Metrics** Each option for an event is associated with a ground-truth *dollar value*, representing the utility of that choice (e.g., the return on a one-dollar investment). We apply multiple performance metrics: **Accuracy (Accu.)** measures if the agent assigns the highest $p_{true}$ to the option with the best utility, measuring the top-1 correctness. **Hard** and **Soft** scores evaluate the value of the highest-$p_{true}$ option and the $p_{true}$-weighted value across all options, respectively, to evaluate the practical return of agent decisions. For cross-task comparability, Min-max normalization is applied to the hard and soft scores with respect to the values of options for every task. **Brier Score (BS)** quantifies the MSE of the predicted distribution across options. In addition, we assess prediction

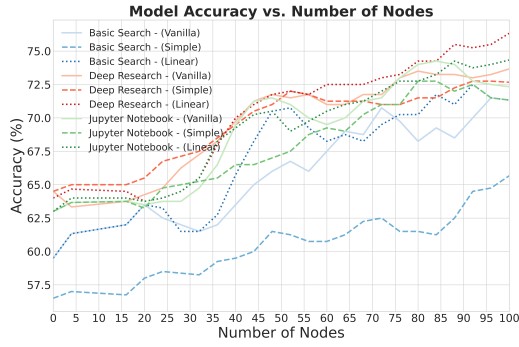 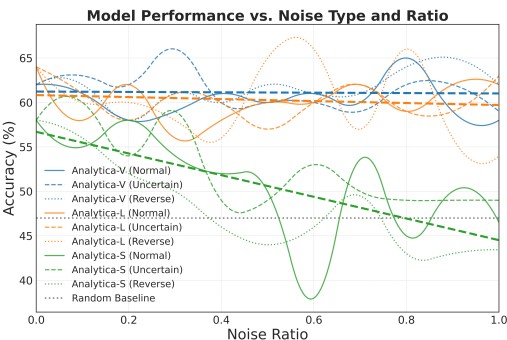

Figure 4: Accuracy vs. number of nodes.  Figure 5: Robustness of different synthesis rules.

stability by performing 10 runs of each task on a 100-task subset, then compute **Confidence (Conf.)** as the average highest $p_{true}$ the agent produced, indicating its self-assessed certainty, and **Variance (Var.)** of the hard score. Lastly, we measure efficiency by **API Cost** and **Wall-clock Time**.

## 5.2 RQ1: ANALYTICA PERFORMANCE AND STABILITY

In Table 2, we illustrate that Analytica substantially improves performance. We perform a Mc-Nemar's test to assess our findings in § D.1. In particular, on average, the linear rule improves 15.84% accuracy, achieving a highest confidence of 83.41 with a variance of 6.59%. It supports our bias-variance reduction framework discussed in 4.2. We ablate the grounders in Table 3, Analytica augments for *all* base grounders. Moreover, it outperforms Deep Research with a Basic Search, which can also be enhanced by Analytica. Meanwhile, it confirms that grounder builds the foundation of lowering biases. Notably, our Jupyter Notebook (NB) grounder with Analytica-L shows an accuracy close to Deep Research (-1.34% worse) with 90.35% lower cost and 52.85% time saving. Conversely, the simple logic rule shows the lowest accuracy enhancement at 4.22%, corroborating our theoretical results presented in § 4.3. We extend our evaluation to the scientific domain by Matter-of-Fact benchmark (Jansen et al., 2025a) in § D.5, and small open-weight models in § D.4.4.

## 5.3 RQ2: SCALABILITY AND ROBUSTNESS OF ANALYTICA

We study scalability by running Analytica with 10 to 100 leaf limits in the 100-task subset above. Once a tree reaches a leaf limit of 10, we apply a *recursion* explained in § 4.4 to expand each leaf *sequentially* to ensure stopping around the target limit. Fig. 4 shows a clear positive correlation between the number of nodes and the accuracy, strongly endorsing the scalability of our method. We further study the robustness of different synthesis rules with the same subset by injecting different types of noise into the grounder: *normal* noise $\hat{p}_{true} = p_{true} + U(0, \alpha)$ where $\alpha$ is the noise ratio, *uncertain* and *reverse* noise where $\hat{p}_{true} = U(0, 1)$ or $\hat{p}_{true} = 1 - p_{true}$ with probability $\alpha$, respectively. Fig. 5 indicates that the simple logic rule is highly susceptible to noise, whereas the linear rule demonstrates high robustness as analyzed in Proposition 1. In contrast, the vanilla rule is minimally affected as it mainly depends on textual reports rather than the estimated truth value.

## 5.4 RQ3: UNDERSTANDING THE PERFORMANCE VS. COST TRADE-OFF

Fig. 6 provides a comprehensive overview of the performance-cost trade-offs. Overall, Analytica sits closely on the effective frontier with negligible overhead (analyze and synthesize). Most costs arise from invoking the leaf base agent. The plot of accuracy against monetary and time cost clearly illustrates that more powerful configurations occupy the high-performance, high-cost quadrant. The choice of *Grounder* is the single largest determinant of cost and performance, establishing distinct efficiency frontiers. Notably, our *Jupyter Notebook* grounder demonstrates high cost-effectiveness.

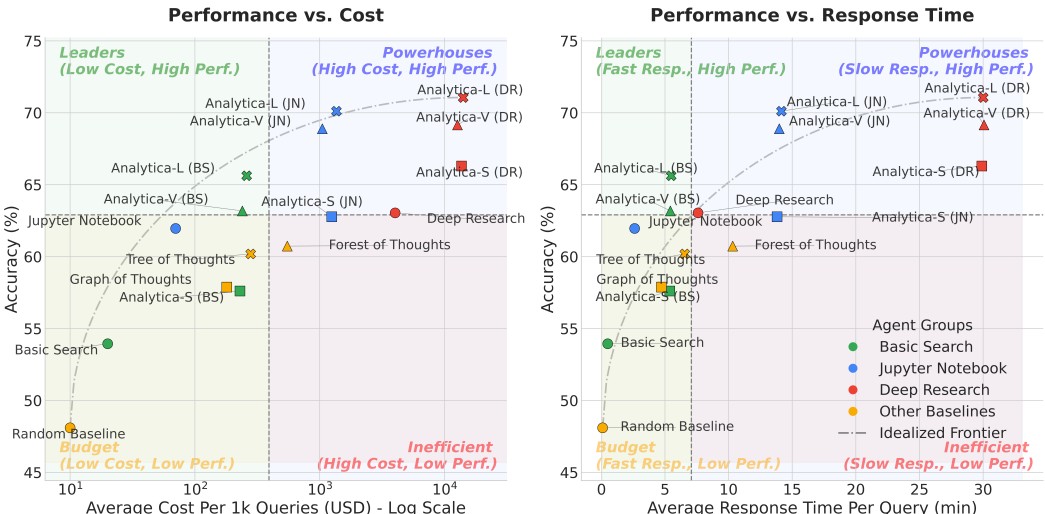

Figure 6: Performance vs. cost trade-off analysis. The plots visualize accuracy against monetary cost (left, log scale) and response time (right, linear scale) for all evaluated methods.

## 6 LIMITATION AND DISCUSSION

Analytica still omits some potential error sources in addition to the ones we discussed in § 4.2. 1) **Assumption of Independence:** Our framework performs best when the child propositions are independent. While our Analyzer agent presents an empirical solution, ensuring independence in principle and estimating the correlations for real-world propositions remains an open challenge. 2) **Robust Synthesizer:** Errors in estimated coefficients of the synthesizer can lead to potential errors, as shown in § D.8. Producing reliable estimations for these coefficients can be crucial. 3) **Hybrid Grounder:** We currently apply the same grounder to all leaves; however, different propositions may have different properties and require grounders with different skill sets. It is possible to adaptively select grounders with diverse capacities for different propositions to improve efficiency and accuracy, as recently studied in model routing (Ong et al., 2025; Ding et al., 2025).

Analytica's practical value extends to complex, high-stakes, critical real-world domains, where decision-making and analysis require transparent reasoning and robustness, such as applications for economists, policymakers, scientists, and robots. More generally, Analytica can serve as a *complex analysis backbone* for autonomous systems by breaking down uncertain, poorly specified problems into calibrated, empirically testable soft propositions, thereby supporting downstream autonomous agents in performing interpretable, reliable reasoning in real-world conditions.

## 7 CONCLUSION

In this work, we introduce Soft Propositional Reasoning (SPR) for complex, real-world analysis, transitioning from heuristic reasoning in unstructured text to a principled, robust process within a soft propositional space. Our system, Analytica, leverages this framework and is derived from first principles to achieve high accuracy across various forecasting tasks, significantly enhancing both accuracy and stability over strong baselines while consistently augmenting various grounders. The modular, divide-and-conquer architecture enables exceptional scalability through massive parallelism, providing unique capabilities for interactive scenario analysis with resynthesis. In addition, we conduct comprehensive theoretical and empirical assessments to examine the underlying principles of robust LLM-based analysis and forecasting, which establishes a strong and transparent basis for creating reliable LLM agents in high-stakes, real-world domains.

## ETHICS STATEMENT

Our study is foundational research for the LLM agentic forecast and analysis. We see no potential negative impact on society.

## REPRODUCIBILITY STATEMENT

We provide complete details of our experiment setup in § 5.1 and § C.1. We also disclose details of our dataset construction in § C.2 and agent implementations in § C.3 and § C.4.

## USE OF LLMS STATEMENT

We primarily use LLMs to polish the writing and check typos.

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

## A  THEORETICAL ANALYSIS

### A.1  ROBUSTNESS OF THE SYNTHESIS RULE

In this section, we provide the formal proof for Proposition 1, and a further analysis of why the **Linear** synthesis rule is more robust to noise in its inputs than the **Simple Logic** rule. A robust rule should ensure that small errors in the estimation of child propositions do not lead to large, unpredictable errors in the parent proposition's estimate. We demonstrate that the Linear rule's structure inherently dampens noise, whereas the logical operators can amplify it.

**Setup: Modeling Estimation Error**   Let $C_j$ be the unknown "true" soft truth value for a child proposition. The Grounder produces an estimate, $\hat{C}_j$, which includes some random error, $\epsilon_j$. We can model this as:

$$\hat{C}_j = C_j + \epsilon_j$$

We assume the errors are unbiased ($E[\epsilon_j] = 0$) and have a variance of $\text{Var}(\epsilon_j) = \sigma_j^2$. Let $P = f(C_1, \ldots, C_n)$ be the true value of the parent, and $\hat{P} = f(\hat{C}_1, \ldots, \hat{C}_n)$ be the final estimate based on the noisy inputs. A rule $f$ is robust if the propagated error, $\hat{P} - P$, is small. We can approximate the variance of the output estimate, $\text{Var}(\hat{P})$, using the propagation of uncertainty formula (a first-order Taylor expansion):

$$\text{Var}(\hat{P}) \approx \sum_{j=1}^{n} \left( \frac{\partial f}{\partial C_j} \right)^2 \sigma_j^2$$

The partial derivative, $\frac{\partial f}{\partial C_j}$, measures the **sensitivity** of the output to an error in the input $C_j$. A smaller sensitivity indicates a more robust rule.

**Analysis of the Simple Logic Rule**   The Simple Logic rule uses non-linear operators like `AND` ($A \cdot B$) and `OR` ($A + B - AB$). Let's analyze the sensitivity for a two-input function:

- For an **AND** gate, $P = C_1 \cdot C_2$, the sensitivities are:

$$\frac{\partial P}{\partial C_1} = C_2 \quad \text{and} \quad \frac{\partial P}{\partial C_2} = C_1$$

- For an **OR** gate, $P = C_1 + C_2 - C_1 C_2$, the sensitivities are:

$$\frac{\partial P}{\partial C_1} = 1 - C_2 \quad \text{and} \quad \frac{\partial P}{\partial C_2} = 1 - C_1$$

The key issue is that the sensitivity to an error in one input **depends on the value of the other inputs**. For an `AND` gate, if $C_2$ is high (e.g., 0.9), any error in $C_1$ is passed through with high impact. This creates a brittle system where high-confidence inputs can paradoxically increase the rule's sensitivity to noise from other inputs. This also leads to "tipping points"; a small error can cause a dramatic change in the output (e.g., if one input to an `AND` gate flips from high to low, the output collapses).

**Analysis of the Linear Rule**   For the Linear rule, $P = \beta_0 + \sum_{j=1}^{n} \beta_j C_j$, the sensitivity is constant for each input:

$$\frac{\partial P}{\partial C_j} = \beta_j$$

The sensitivity to an error in $C_j$ is simply its weight, $\beta_j$. It does not depend on the values of other inputs. Since the weights $\beta_j$ are typically less than 1, the rule acts as a weighted average that inherently **dampens** or **smooths** input noise. The error propagation is stable and predictable.

Fig. 7 and 8 visualized the gradient surfaces and sensitivity plots for linear and simple logic rules, respectively (see similar analysis in Van Krieken et al. (2022)). The surface of the Simple Logic rule is curved. This non-linearity is the source of its unpredictable behavior. The surface of the Linear rule is a perfect plane, demonstrating its smooth and predictable nature. Small changes in the inputs lead to proportional changes in the output.

The sensitivity plot for the Simple Logic rule is a ramp. The sensitivity to noise is very low when both inputs are near zero, but becomes very high when the inputs are near one. This visually confirms the "state-dependent sensitivity" mentioned in the proof—the rule's robustness changes depending on the data, making it brittle. The sensitivity plot for the Linear rule is a perfectly flat plane. This is the most important takeaway. It shows that the rule's sensitivity to noise is constant and bounded across the entire input space. It dampens errors predictably, regardless of whether the input propositions are considered likely or unlikely to be true.

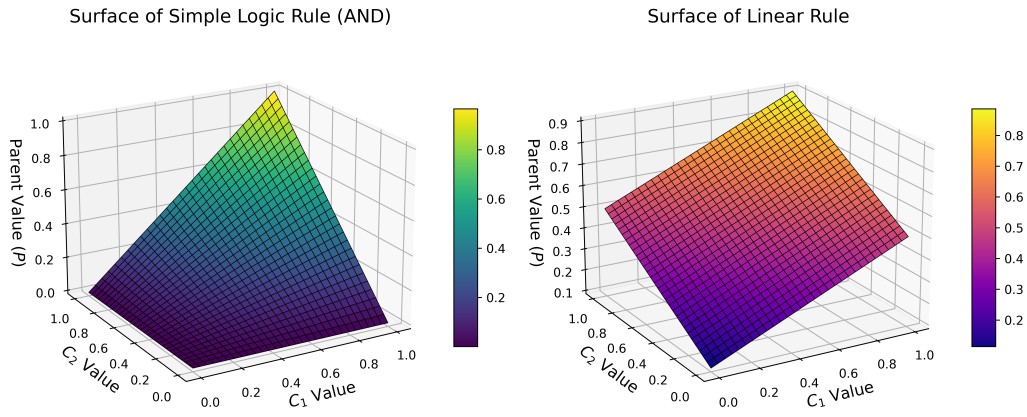

Figure 7: Gradient surfaces of a simple logic formula $C_1 \wedge C_2$ and a linear formula $0.1 + 0.4 \cdot C_1 + 0.4 \cdot C_2$ respectively.

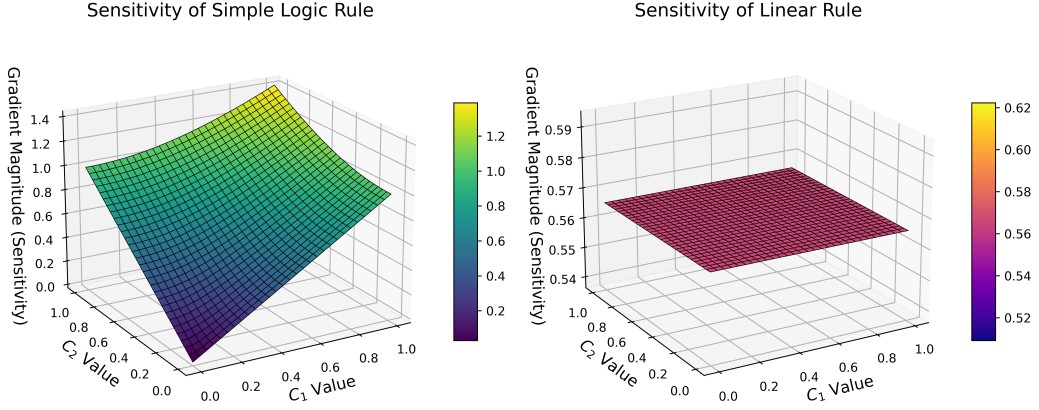

Figure 8: Noise sensitivities of a simple logic formula $C_1 \wedge C_2$ and a linear formula $0.1 + 0.4 \cdot C_1 + 0.4 \cdot C_2$ respectively.

**Conclusion: Principles for a Robust Synthesis Rule**   This analysis allows us to conclude with three general principles for designing a robust synthesis rule, $f$:

1. **Bounded Sensitivity**: The partial derivatives $\frac{\partial f}{\partial C_j}$ should be bounded and preferably small. A rule where sensitivity can approach or exceed 1 is prone to amplifying noise. The Linear rule's sensitivities are bounded by the learned weights, whereas the Logic rule's can be large.

2. **Smoothing Property**: The function should have a natural smoothing or averaging effect. Weighted averages, like the Linear rule, are classic examples of noise-reducing functions.

3. **Graceful Degradation**: The function should be smooth, without sharp "cliffs" or discontinuities in its derivatives. This ensures that small changes in inputs lead to proportionally small changes in the output, avoiding the "tipping point" behavior seen in logical gates.

The Linear rule satisfies all three principles, providing a strong theoretical reason for its superior empirical performance in noisy, real-world scenarios.

A.2 Efficiency and Scalability of Recursive Analytica

In this section, we provide a formal analysis of the computational complexity and scalability of the recursive **Analytica**$^n$ framework. We demonstrate that while the total computation required grows exponentially with the recursion depth $n$, the wall-clock time can be managed to near-linear growth due to massive parallelism. Furthermore, we show that the recursive, divide-and-conquer approach provides a crucial benefit we term *Context Locality*, which makes scaling feasible within the finite context windows of LLMs.

**Setup**   Let us define the parameters for our analysis:

- $n$: The recursion depth of Analytica$^n$. Analytica$^1$ is the base case.
- $K$: The average number of leaf nodes created by the *Analyzer* at each decomposition step (i.e., the branching factor, denoted as $L_{max}$ in Algorithm 1).
- $M$: The total number of final leaf nodes to be grounded. In an $n$-level recursive structure, $M \approx K^n$.
- $T_G$: The average time (latency) required to *Ground* a single leaf proposition. This represents the atomic unit of deep reasoning work.
- $P$: The number of parallel workers available for executing *Ground* tasks concurrently.

**Work Complexity (Total Computation)**   The *work* represents the total computational cost if the entire process were run sequentially on a single processor. It is dominated by the grounding of all final leaf nodes.

> **Proposition 2** (Exponential Work Complexity). *The total work complexity $W(n)$ of Analytica$^n$ is exponential in the recursion depth $n$.*
>
> $$W(n) = O(K^n \cdot T_G)$$

*Proof.* At recursion depth $n$, the total number of final leaf nodes is approximately $M = K^n$. Since each of these $M$ leaves requires an independent grounding process of average time $T_G$, the total sequential time (work) is the product of these two quantities. □

**Time Complexity (Parallel Execution)**   The *time* complexity (also known as span or depth) measures the wall-clock time assuming parallel execution. The structure of Analytica allows all leaf nodes at the final level to be grounded simultaneously.

> **Proposition 3** (Parallel Time Complexity). *With $P$ parallel workers, the time complexity $T_P(n)$ of Analytica$^n$ is primarily determined by the parallel execution of the final grounding phase.*
>
> $$T_P(n) = O\left(n + \frac{K^n}{P} \cdot T_G\right)$$

*Proof.* The process has a sequential dependency through the $n$ levels of recursion (analysis and synthesis at each level), contributing the $O(n)$ term for overhead. The dominant term is the final step, where all $M = K^n$ leaves are grounded. With $P$ workers, this phase takes $\lceil K^n/P \rceil$ batches of parallel executions, each taking time $T_G$. For large $n$, the exponential term $O(\frac{K^n}{P})$ dominates the linear term $O(n)$. □

**Interpretation**   This explains the empirical results. While the total work $W(n)$ is exponential, the execution time $T_P(n)$ is divided by the number of parallel workers $P$. For a system with high parallelism (large $P$, e.g., 1000 in Table 1), the exponential growth is drastically mitigated, leading to the observed near-linear time growth for moderate $n$. This demonstrates the immense scalability power unlocked by the framework's parallel design.

**The Benefit of Recursion: Context Locality**  Beyond parallelism, the recursive, divide-and-conquer nature of Analytica$^n$ is essential for its feasibility. A monolithic, non-recursive approach would be intractable due to the context limitations of LLMs.

> **Proposition 4** (Context Locality). *The recursive structure of Analytica$^n$ maintains a small, bounded context size for each LLM call, whereas a monolithic approach would require a context size that grows exponentially with the problem complexity.*

*Proof.* Consider a monolithic agent trying to solve the problem in one pass. It would need to generate the entire proposition tree with $M = K^n$ leaves. The size of this tree, which must be maintained in the LLM's context, would be $O(K^n)$. For even modest $n$ and $K$, this would quickly exceed any modern LLM's context window.

In contrast, Analytica$^n$ exhibits **context locality**. Each call to the *Grounder* operates on a single leaf proposition, a task with a constant context size, $O(1)$. Each call to the *Analyzer* or *Synthesizer* operates on a parent and its $K$ children, a context size of $O(K)$, which is independent of the recursion depth $n$. The maximum context required at any point in the process remains small and bounded, regardless of the overall size of the problem. □

**Conclusion**  The power of the recursive Analytica$^n$ framework stems from two sources. First, its parallel architecture transforms an exponentially complex problem in terms of *work* into a manageable task in terms of *time*. Second, and more fundamentally, its recursive decomposition provides *context locality*, breaking an intractably large problem into a vast number of small, independent sub-problems that fit within an LLM's finite context. This combination of parallelism and locality is what endows Analytica with its profound scalability.

## B  FORMAL ANALYSIS OF THE ANALYTICA REASONING MODEL

To better understand the semantics of Analytica's underlying reasoning model and linear synthesis rule from Eq. 2, in this section, we provide a sketch of how to directly translate an Analytica computation graph produced during the synthesis stage (see again Figs. 1-3) into an equivalent Bayesian network. Recalling again that our synthesis rule scores non-leaf propositions $\rho_i.p_{true} \in [0,1]$ using the scores of all its children $\bar{\rho}.p_{true} \in [0,1]$ as follows:

$$\rho.p_{true} = \beta_0 + \sum_j \beta_j \cdot \bar{\rho}_j.p_{true}$$

We define the following graphical representation of the Bayesian network corresponding to the above equation (without loss of generality, we focus on the case involving two children $\rho_1, \rho_2$):

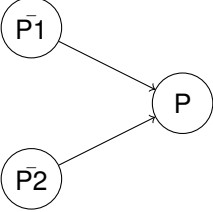

where $\mathsf{P}$ and $\bar{\mathsf{P}}1, \bar{\mathsf{P}}2$ are binary random variables corresponding to the root $\rho_i$ and its children nodes $\rho_1, \rho_2$. Standardly, the probability of variable $\mathsf{P}$ in this network, denoted below as $Pr(\mathsf{P})$ for $Pr(\mathsf{P} = 1)$, is computed as follows (with binary indicator variables $p_j \in \{0,1\}$):

$$Pr(\mathsf{P}) = \sum_{c_1, c_2 \in \{0,1\}} Pr(\mathsf{P} \mid \bar{\mathsf{P}}1 = p1, \bar{\mathsf{P}}2 = p2) \cdot Pr(\bar{\mathsf{P}}1 = p1, \bar{\mathsf{P}}2 = p2)$$

$$= \sum_{c_1, c_2 \in \{0,1\}} Pr(\mathsf{P} \mid \bar{\mathsf{P}}1 = p1, \bar{\mathsf{P}}2 = p2) \cdot \underbrace{Pr(\bar{\mathsf{P}}1 = p1) \cdot Pr(\bar{\mathsf{P}}2 = p2)}_{\text{independence}}.$$

By then defining the corresponding CPDs as follows using our original $\beta$ coefficients:

$$Pr(\mathsf{P} \mid \bar{\mathsf{P}1} = p1, \bar{\mathsf{P}2} = p2) := \beta_0 + (\beta_1 \cdot p_1) + (\beta_2 \cdot p_2)$$

and the non-root node probabilities $\bar{\mathsf{P}}$ using their original node scores:

$$Pr(\bar{\mathsf{P}} = p) := (\bar{\rho}.p_{true})^p \cdot (1 - \bar{\rho}.p_{true})^{(1-p)}$$

We can observe below that $Pr(\mathsf{P})$ under this network and *linear weight parameterization* corresponds exactly to the linear synthesis rule score $\rho.p_{true}$ (for readability, we use $\mathbf{p}$ and $\overline{\mathbf{p}}$ in place of $Pr(C)$ and $1 - Pr(C)$, respectively, and replace $C = c$ in $P(\mathsf{P} \mid \cdot)$ with Booleans 0,1):

$$Pr(\mathsf{P}) = \left[Pr(\mathsf{P} \mid 0,0)\overline{\mathbf{p}}_1\overline{\mathbf{p}}_2\right] + \left[Pr(\mathsf{P} \mid 1,0)\mathbf{p}_1\overline{\mathbf{p}}_2\right] + \left[Pr(P \mid 0,1)\overline{\mathbf{p}_1}\mathbf{p}_2\right] + \left[Pr(\mathsf{P} \mid 1,1)\mathbf{p}_1\mathbf{p}_2\right]$$

$$= \left[\beta_0\overline{\mathbf{p}_1}\,\overline{\mathbf{p}_2}\right] + \left[(\beta_0 + \beta_1)\mathbf{p}_1\overline{\mathbf{p}_2}\right] + \left[(\beta_0 + \beta_2)\overline{\mathbf{p}_1}\,\mathbf{p}_2\right] + \left[(\beta_0 + \beta_1 + \beta2)\mathbf{p}_0\,\mathbf{p}_1\right] \qquad \text{w/ } \beta\text{s}$$

$$= \left[\beta_0\overbrace{\overline{\mathbf{p}_1}\mathbf{p}_2\overline{\mathbf{p}}_1\overline{\mathbf{p}}_2}\right] + \left[\beta_1\mathbf{p}_1\overbrace{\mathbf{p}_2\overline{\mathbf{p}}_2}\right] + \left[\beta_2\mathbf{p}_2\overbrace{\overline{\mathbf{p}}_1\overline{\mathbf{p}}_1}\right] \qquad \text{Algebra/cancellation}$$

$$= \rho.p_{true} \qquad \text{Whenever all } \beta\text{s} \in [0,1] \text{ and } \beta_0 + \sum_j \beta_j \leq 1.$$

Importantly, we emphasize that this equivalence only holds when the $\beta$ parameters have the structure given in the last line. While such a condition was not strictly enforced in our existing experiments, we note, however, that such a constraint can be enforced in our current system by employing a variety of different scaling techniques for the given $\beta$s (e.g., min-max scaling, softmax).

By translating our synthesis rule into an explicit Bayes net, we get a more transparent picture of the semantics underlying our synthesis agent. In addition, such a formulation also suggests a number of new synthesis strategies that use known techniques from probabilistic graphical models (Koller & Friedman, 2009). We provide specific examples below by considering a translation into a different, and semantically more transparent, formal system below.

**The synthesis rule as a probabilistic logic program** Interestingly, we noticed through the above derivation that the linear synthesis rule has a more compact and natural interpretation as a certain type of probabilistic logic program (PLP) (De Raedt & Kimmig, 2015). Below shows a Problog implementation (De Raedt et al., 2007; Dries et al., 2015) of the linear synthesis rule (the red parts correspond to the corresponding parameters in the linear synthesis rule):

```
%% children nodes as probabilistic facts
ρ̄1.ptrue::p1.
ρ̄2.ptrue::p2.
%% betas as annotated disjunctions, categorical variable
β0::b0; β1::b1; β2::b2.
%%% tree links as if-then rules
p :- b0.
p :- b1, p1.
p :- b2, p2.
%%% probability of root p
query(p).
```

where $\mathsf{p}, \mathsf{p1}, \mathsf{p2}$ denote the root and non-root propositions, respectively (the latter being implemented as probabilistic facts), and the $\mathsf{b}$s correspond to the beta parameters (expressed as a relational categorical distribution using a construct called an annotated disjunction (AD) originally from Vennekens et al. (2004)). To see that the probability of $\mathsf{p}$ (via $\mathsf{query(p)}$) is equal to $\rho_p.p_{true}$ in the linear synthesis rule, we consider the Boolean encoding of this program under standard closed-world semantics (Clark, 1977), which corresponds to the formula $\mathsf{F}$ below:

$$\mathsf{F} := \underbrace{\left(\mathsf{p} \leftrightarrow \left(\mathsf{b0} \vee \bigvee_{j>0} \mathsf{bj} \wedge \mathsf{pj}\right)\right)}_{\text{(noisy-)or}} \wedge \underbrace{\left(\bigvee_{j \geq -1} \mathsf{bj}\right) \wedge \left(\bigwedge_{\forall i,j \mid i \neq j} \neg(\mathsf{bi} \wedge \mathsf{bj})\right)}_{\text{one-hot constraint, categorical}}$$

where a special variable $b-1$ is used to denote the case where all other $b$s are false (used whenever the sum of $\beta$s is less than 1). Under a standard possible world semantics and encoding of PLPs and ADs into weighted logic (Fierens et al., 2015), we can then compute the probability of $p$ as the weighted model count (WMC) (Chavira & Darwiche, 2008) of $F \wedge p$, and observe under the following weighting $w(\cdot)$ of variables:

$$\forall j \geq 0.\ w(b_j) = \beta_j, w(b_{-1}) = 1 - (\beta_0 + \beta_1 + \beta_2),\ \forall j.w(\neg b_j) = 1$$
$$\forall j.\ w(p_j) = \rho_j.p_{true},\ w(\neg p_j) = 1 - \rho_j.p_{true}$$
$$w(p) = 1$$

that the following equivalence holds (where $w(\pm p)$ is used as shorthand to denote the weight of a variable $p$ or its negation, which marginalize out, and $\mathbf{w}$ denotes a possible world or set of variable instantiations consisting of literals $l$):

$$\underbrace{Pr(F \wedge p)}_{\texttt{query(p)}} := \underbrace{\sum_{\mathbf{w} \models F \wedge p} \prod_{l \in \mathbf{w}} w(l)}_{\text{weighted model count (WMC) of } p \wedge F \text{ under } w(\cdot)}$$

$$= \underbrace{\left[ w(b_0)\overline{w(\pm p_1)w(\pm p_2)} \right] + \left[ w(b_1)w(p_1)\overline{w(\pm p_2)} \right] + \left[ w(b_2)\overline{w(\pm p_1)}w(\pm p_2) \right]}_{\text{All logical interpretations of } p \wedge F \text{ with weights (removed literals } l \text{ with weight 1)}}$$

$$= \left[ \beta_0 \right] + \left[ \beta_1 \bar{\rho}_1.p_{true} \right] + \left[ \beta_2 \bar{\rho}_2.p_{true} \right]$$

$$= \rho.p_{true} \qquad \text{Whenever all } \beta s \in [0, 1] \text{ and } \beta_0 + \sum_j \beta_j \leq 1.$$

At noted above, the translation into $F$ shows more clearly how the linear rule operationalizes a kind of noisy-or style of reasoning (Pearl, 2014) (i.e., *the root being true depends on one or more of its children being true, or $\beta_0$ being true*) with an added one-hot constraint that enforces only one $\beta$ being true. By removing this one-hot constraint (or equivalently, removing the annotated disjunction in the logic program), one derives a standard noisy-or rule, which is an alternative synthesis strategy that one can in principle experiment with. Building on these foundations, many techniques from PGMs and probabilistic logic programming suggest themselves for improving the robustness of the synthesis agent, such as adding explicit negative factors, e.g., via *inhibited noisy-or rules* (Meert & Vennekens, 2014), or modeling parameter uncertainty via Bayesian inference as in Cerutti et al. (2019); Verreet et al. (2022) (see Agarwal et al. (2025) for similar ideas in the context of LLM agents).

## C  System Details

### C.1  Detailed Setup

Our experiments were conducted using a set of standardized hyperparameters to ensure consistency and reproducibility across all agent configurations. These settings govern the behavior of the LLM agents, the grounding process, and the structural constraints of the Analytica framework.

**General Agent Settings**   These parameters control the core interaction loop for all LLM agents.

**`max_exception_retry:`   3** The maximum number of times an agent will attempt to re-call the LLM if a recoverable error (e.g., invalid JSON format, parsing failure, invalid weights generated for linear rule, invalid formula generated for simple logic rule) occurs.

**`max_interrupt_times:`   5** The maximum number of interruptions (e.g., tool calls for API documentation) an agent can make in a single reasoning step before being required to produce a final response for that step.

**Analytica Framework Settings**   These parameters specifically control the behavior of the Analytica architecture during the analysis and grounding phases.

**max_n_leaves:** **10** A limit on the number of leaf propositions the **Analyzer** can generate. The decomposition phase is halted once the proposition tree reaches approximately 10 leaves to ensure a comparable analytical budget across different methods. Notice that in practice, it usually halts with more than 10 nodes as we perform a post-check.

**max_concurrent_prove:** **20** The maximum number of leaf propositions that can be grounded in parallel by the framework. This leverages asynchronous execution to improve efficiency.

**max_proof_retries:** **3** The number of times the framework will retry the entire grounding process for a single leaf proposition if the assigned **Grounder** agent fails catastrophically.

**Jupyter Notebook Grounder Settings** These settings govern the iterative proof-construction process for our most advanced grounder.

**max_proof_steps:** **20** The maximum number of turns (i.e., generating and executing one or more notebook cells) the agent can take within a single Jupyter session before it is forced to terminate the analysis and provide a conclusion.

**debug_max_retries:** **5** The maximum number of attempts the agent is given to fix a single erroneous Python cell before the proof is considered to have failed.

**abs_intercept_max:** **0.1** A constraint on the absolute value of the intercept term ($\beta_0$) for the **Linear Synthesizer**. This encourages the agent to base its synthesis on the evidence from child propositions rather than relying on a large, unexplained prior.

**Experimental Simplification for Binary Tasks** To enhance computational efficiency, a simplification was applied to all tasks with exactly two mutually exclusive options (e.g., "Long" vs. "Short", "Yes" vs. "No"). For these binary tasks, the framework was configured to perform a full analysis or grounding process for only the first option to determine its soft truth value, $P(\text{option}_1)$. The soft truth value for the second, opposing option was then programmatically derived as $P(\text{option}_2) = 1 - P(\text{option}_1)$, leveraging the mutually exclusive nature of the choice set. This approach halves the computational cost for binary forecasting without loss of information. We also apply a decision threshold $\delta$ for binary tasks: if $P_{true} > \delta$, the claim is labeled as True; otherwise, it is labeled as False.

## C.2 DATASET CONSTRUCTION

Our benchmark dataset was meticulously constructed to provide a diverse and challenging set of real-world forecasting tasks. The data spans two primary domains: high-liquidity predictive markets and a wide range of traditional financial markets. The entire construction process involved several stages of data acquisition, filtering, and validation to ensure the quality and relevance of the tasks.

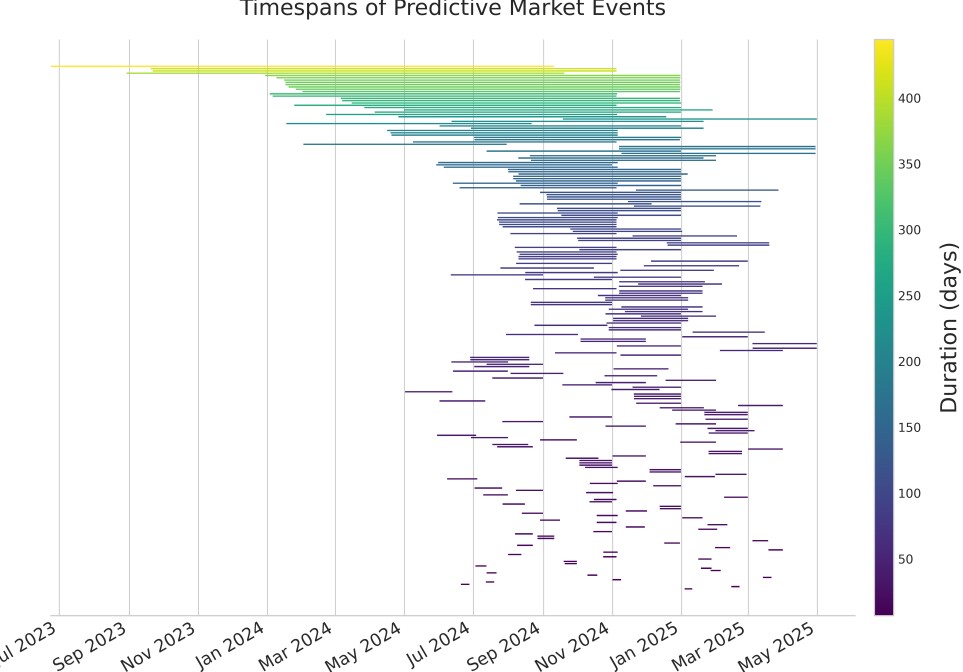

Figure 9: A Gantt chart illustrating the timespans of the predictive market events included in our dataset. Each horizontal bar represents a single event, starting on its opening date and ending on its resolution date. The color of the bar indicates the event's duration in days. The chart highlights the diversity of forecasting horizons, ranging from short-term events of a few weeks to long-term predictions spanning over a year.

**Predictive Markets**  Data for predictive markets was sourced from two of the largest plat-forms, **Kalshi** and **Polymarket**, via their respective official APIs (`https://kalshi.com/api`, `https://docs.polymarket.com`). We applied a multi-stage filtering process to the raw event data:

- **Temporal Filtering:** We selected events with resolution dates occurring after our models' knowledge cutoff of June 1, 2024, and before May 1, 2025, to ensure they represented genuine future predictions.
- **Volume Filtering:** To focus on events with sufficient public interest and liquidity, we enforced a minimum total trading volume of $500,000 across all of an event's markets.
- **Topical Filtering:** We used a comprehensive set of keywords (e.g., "who will win", "movie", "sports team vs.", "price range") to exclude events that are purely speculative, sports or entertainment-related, or not amenable to deep analytical reasoning.
- **Structural Filtering:** Events with an excessive number of potential outcomes (more than 5 markets) were removed to maintain a manageable task complexity.

The resulting set of predictive market events covers a wide range of time horizons, from tasks re-solving in a few weeks to those lasting over a year, as illustrated in Fig. 9.

**Financial Markets**  To create tasks for financial markets, we sourced historical end-of-day price data from the Financial Modeling Prep (FMP) API. We curated a diverse list of highly-liquid assets from several categories to ensure broad market coverage:

- **Stocks:** A core set of large-cap stocks was selected from major US indices, including the S&P 100, Dow Jones Industrial Average, and NASDAQ-100.

- **Indices:** A comprehensive list of major global and sector-specific stock market indices.

- **Funds:** A variety of Exchange-Traded Funds (ETFs), including those focused on specific sectors, investment themes, and active management strategies.

- **Cryptocurrencies:** The top 8 cryptocurrencies by market capitalization, such as BTCUSD and ETHUSD, were included.

- **Forex:** Major and minor currency pairs were selected to represent the global foreign exchange market.

- **Commodities:** A list of all available commodity futures provided by the data source.

**Final Curation and Validation** After the initial filtering and selection, all potential tasks underwent a final validation step. Each event was used to construct a 'Query' object, which simulates the task setup for an agent. Any event that failed during this process—due to issues like incomplete historical data, an invalid time span, or resulting in options with no distinguishable value (i.e., all outcomes having the same payoff)—was discarded from the final dataset. This final check ensures that every task in the benchmark is well-formed and evaluable.

### C.3 BASIC SEARCH AND DEEP RESEARCH GROUNDERS

To benchmark our framework against non-programming agents with varying levels of sophistication, we implemented two text-based grounders: **Basic Search** and **Deep Research**. Both agents are built upon a common, customized search service to ensure consistency in information access. This service is powered by the Exa API (`https://exa.ai/`) and is strictly configured to only return web results published before the experiment's knowledge cutoff date, thereby preventing data leakage from the future.

For **Basic Search** grounder, the search function was provided to the agent as a tool. When tasked with grounding a leaf proposition, it may use the tool through function calling provided by the OpenAI API. The **Deep Research** grounder is implemented using the OpenAI DeepResearch API. We replace the default search tool with our own customized MCP server hosting the same search tool in the Basic Search grounder to avoid data leaking.

### C.4 JUPYTER NOTEBOOK GROUNDER

The **Jupyter Notebook Grounder** is the most advanced grounding agent in our framework, designed to simulate the workflow of a human expert performing quantitative and qualitative analysis. Instead of relying solely on text-based reasoning, this agent interacts with a sandboxed Jupyter Notebook environment to construct a rigorous, evidence-based proof for a given leaf proposition. The process is stateful, iterative, and tool-driven, allowing for complex data retrieval, analysis, and visualization.

#### C.4.1 SANDBOX ENVIRONMENT

Each grounding task is executed within an isolated **Jupyter Session**, which provides a secure and stateful computational environment. The sandbox is managed by the `JupyterSandbox` class, which handles the lifecycle of kernel processes and notebook files.

When a session is initiated, a special initialization cell is prepended to the notebook. This cell imports necessary libraries and instantiates the `Proxy` class, which serves as the agent's interface to all external data APIs. This setup ensures that the agent has immediate access to its toolset and that all API calls are configured with the correct knowledge cutoff date, preventing data leakage from the future.

The agent's interaction with the notebook is entirely programmatic. It cannot directly edit or delete previous cells; it can only append new cells, ensuring a verifiable and immutable record of the analysis process.

C.4.2 ITERATIVE PROOF CONSTRUCTION

The agent constructs its proof through an iterative, multi-step process orchestrated by the `Prover` agent logic. The agent reasons about the proposition and decides on a course of action, which it implements by generating a sequence of notebook cells.

**Cell Generation**   The agent's primary output is a stream of Jupyter cells, which can be of two types, as dictated by the system prompt:

- **Markdown Cells** (`<markdown_cell>`): Used for qualitative reasoning, outlining the analytical plan, summarizing intermediate findings, and structuring the final report.

- **Python Cells** (`<python_cell>`): Used for quantitative tasks. This is where the agent performs data retrieval via API calls, conducts statistical analysis, and generates visualizations to support its claims.

**Debugging Loop**   After the agent submits its cells, the sandbox executes them sequentially. If a Python cell fails, the execution halts, and the agent is provided with the error traceback. It then enters a debugging loop, where it is prompted to provide a corrected version of the single erroneous cell. This cycle can repeat for a predefined number of attempts (`debug_max_retries`), allowing the agent to recover from syntax errors, incorrect API usage, or data handling mistakes.

**Termination**   The agent continues this cycle of planning, coding, and debugging until it determines its analysis is complete. It then issues a special `<TERMINATE_NOTEBOOK>` command. At this point, the programming phase ends, and the agent is prompted to synthesize its findings from the notebook into a final, comprehensive proof and a soft truth value ($p_{true}$) for the proposition.

| ID | Name | Description | # |
|---|---|---|---|
| `fmp` | Financial Modeling Prep API | *FMP provides the Stock Market APIs and Financial Data APIs, such as real-time stock prices, financial statements, and historical data. It offers a comprehensive solution to meet all financial data needs.* | 132 |
| `msd` | Main Street Data API | *The Main Street Data API compiled over thousands of metrics related to 2,500 US companies, offering unparalleled insights into businesses beyond standard financial statements.* | 4 |
| `fred` | Federal Reserve Economic Data | *The FRED® API retrieves economic data from the FRED® and ALFRED® websites hosted by the Economic Research Division of the Federal Reserve Bank of St. Louis.* | 16 |
| `gt` | Google Trends Search API | *Google Trends API scrape real-time results from Google Trends. It also supports autocomplete, related queries, related topics, and geo locations.* | 8 |
| `exa` | Exa Search API | *Exa provides three core functionalities: Find webpages using Exa's embeddings-based or Google-style keyword search; obtain clean, up-to-date, parsed HTML; and find similar pages.* | 3 |

Table 4: The library of external data APIs available to the Jupyter Notebook Grounder. Each proxy provides access to a suite of specific endpoints for quantitative analysis. "#" means the number of endpoints.

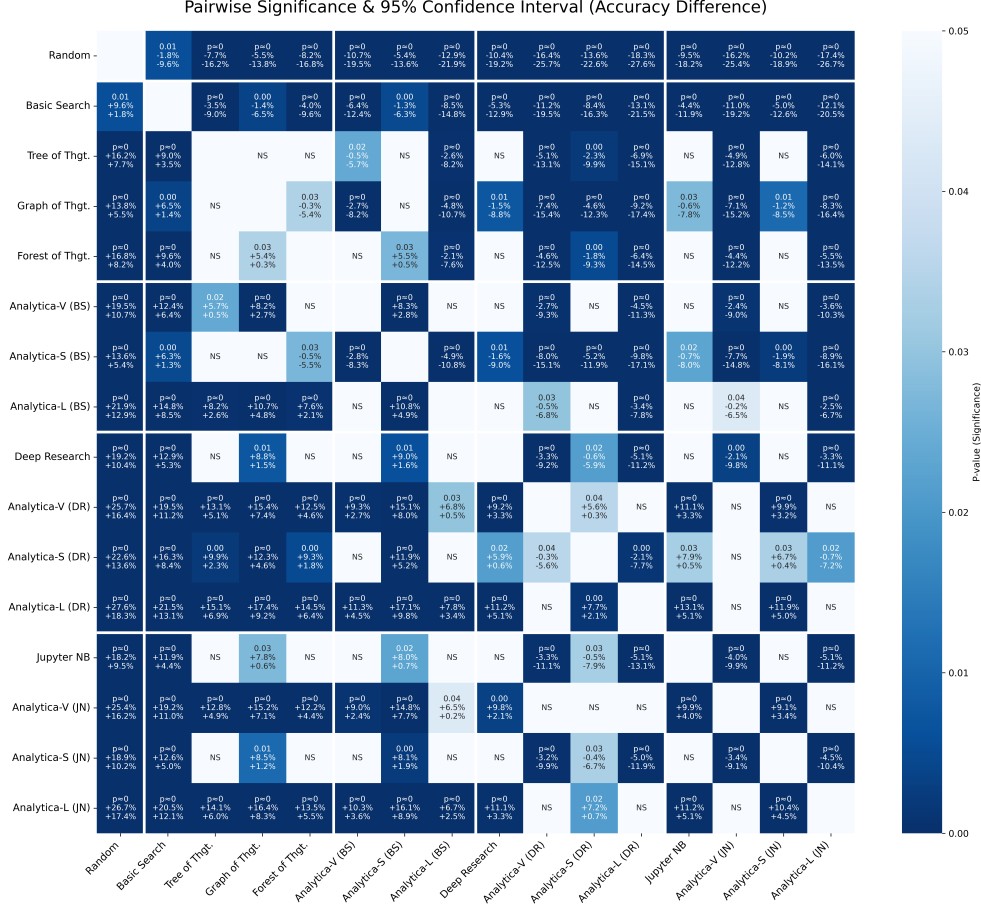

Figure 10: Statistical significance of the results in Table 2, computed by a pairwise McNemar's test. In each square, the first row denotes the P value, the second and third row denotes the upper and lower sides of the confidence interval of the accuracy difference between the model on the y-axis and the x-axis.

### C.4.3 API LIBRARY

The Jupyter environment is augmented with a powerful, extensible library of APIs for accessing real-world data. All API interactions are mediated through a special `CALL_API` function injected into the notebook's scope.

**The Proxy System** The `CALL_API` function is an interface to the `Proxy` class, which manages access to all underlying data sources. The `Proxy` system is designed to be modular, with each data source (e.g., FRED, Financial Modeling Prep) implemented as a separate `BaseProxy` subclass. This design allows for easy integration of new data sources. Before using an API, the agent is instructed to use a `retrieve_api_doc` function to get detailed documentation on endpoints and parameters, promoting correct usage. Table 4 lists the core APIs available to the agent in our experiments.

# D    ADDITIONAL RESULTS

## D.1    MCNEMAR'S TEST

To validate the statistical significance of our accuracy improvements, we performed a pairwise **McNemar's test** on the prediction outcomes (correct vs. incorrect) for all evaluated methods. This test is appropriate for comparing the performance of two classifiers on the same dataset. We test the methods on the full forecasting benchmark introduced in § 5.1. The results, visualized as a matrix of p-values, are presented in Fig. 10.

The matrix clearly shows that the improvements achieved by our top-performing configurations are **highly statistically significant**. For instance, Analytica-L augmented with the Deep Research grounder shows a p-value of p=0.00 when compared against the standalone Deep Research baseline, as well as against all other baselines like Tree of Thoughts and Forest of Thoughts. This indicates that the observed 12.72% relative improvement in accuracy is extremely unlikely to be due to random chance.

Similarly, the highly cost-effective Jupyter Notebook grounder with Analytica-L also demonstrates statistically significant outperformance against its standalone counterpart and the Basic Search-based methods. The test also highlights significant performance differences between the synthesis rules; the Linear (-L) and Vanilla (-V) rules consistently and significantly outperform the Simple Logic (-S) rule across different grounders, confirming the robustness discussed in § 4.3. In cases where the performance difference is small, the test correctly identifies it as not significant (NS), such as the comparison between Analytica-V (DR) and Jupyter Notebook + Analytica-L (JN).

## D.2    PERFORMANCE BY CATEGORY

| | **Pred.** | **Index** | **Stock** | **Fund** | **Forex** | **Comm.** | **Cryp.** | **All** |
|---|---|---|---|---|---|---|---|---|
| *Random* | *46.19* | *55.06* | *43.60* | *47.66* | *50.00* | *50.00* | *37.50* | *48.10* |
| Tree of Thgt. | 45.29 | 71.35 | 63.95 | 67.29 | 56.25 | 62.50 | 50.00 | 60.19 |
| Graph of Thgt. | 45.74 | 66.85 | 60.47 | 64.49 | 50.00 | 65.62 | 37.50 | 57.88 |
| Forest of Thgt. | 46.64 | 69.66 | 64.53 | 69.16 | 50.00 | 68.75 | 50.00 | 60.73 |
| Basic Search | 43.50 | 61.24 | 56.40 | 65.42 | 31.25 | 50.00 | 37.50 | 53.94 |
| + Analytica-V | 43.50 | 79.21 | 65.12 | 74.77 | 43.75 | 71.88 | 62.50 | 63.18 |
| + Analytica-S | 44.84 | 67.98 | 57.56 | 67.29 | 62.50 | 56.25 | 50.00 | 57.61 |
| **+ Analytica-L** | 48.88 | 80.90 | 65.12 | 74.77 | 56.25 | 71.88 | 75.00 | 65.62 |
| Deep Research | 46.64 | 80.34 | 64.53 | 71.96 | 43.75 | 50.00 | 75.00 | 63.04 |
| + Analytica-V | 57.85 | **82.58** | 65.12 | **76.64** | 62.50 | 68.75 | 87.50 | 69.16 |
| + Analytica-S | 53.36 | 79.21 | 63.95 | 74.77 | 62.50 | 68.75 | 75.00 | 66.30 |
| **+ Analytica-L** | **63.68** | 80.90 | **65.70** | 75.70 | 68.75 | 75.00 | **100.0** | **71.06** |
| Jupyter NB | 51.12 | 73.03 | 62.21 | 66.36 | 62.50 | 59.38 | 62.50 | 61.96 |
| + Analytica-V | 60.54 | 79.78 | 64.53 | 75.70 | 50.00 | 75.00 | 75.00 | 68.89 |
| + Analytica-S | 51.12 | 73.60 | 63.37 | 65.42 | 68.75 | 65.62 | 75.00 | 62.77 |
| **+ Analytica-L** | 60.54 | 81.46 | **65.70** | 74.77 | **75.00** | **78.12** | 75.00 | 70.11 |
| *Num. Tasks* | *223* | *178* | *172* | *107* | *16* | *32* | *8* | *736* |
| *Ratio* | *30.30%* | *24.18%* | *23.37%* | *14.54%* | *2.17%* | *4.35%* | *1.09%* | *100%* |

Table 5: Model accuracy (Accu. %) breakdown by task category.

Table 5 provides a granular breakdown of model performance across seven distinct categories: Predictive Markets (Pred.), Stock Indices (Index), individual Stocks, Funds, Foreign Exchange (Forex), Commodities (Comm.), and Cryptocurrencies (Cryp.). This detailed view reveals several key insights into the strengths and weaknesses of the different methods.

Across the board, Analytica-enhanced agents consistently outperform their standalone counterparts in almost every category. The most substantial gains are observed in the more traditional and data-

rich financial markets, such as Indices, Stocks, and Funds. For instance, when augmenting Deep Research, Analytica-L achieves a remarkable 100% accuracy on the 8 cryptocurrency tasks and significantly boosts performance in Predictive Markets from 46.64% to 63.68%. This suggests that the structured, decompositional approach of Analytica is particularly effective in domains where a multitude of quantitative and qualitative factors must be weighed.

Interestingly, most models, including the more advanced ones, struggle with Predictive Market tasks, with many performing below the random baseline. This highlights the inherent difficulty of these problems, which often involve complex socio-political factors and sparse, noisy data. However, it is in this challenging domain that Analytica-L provides the most dramatic relative improvement, demonstrating its ability to impose a coherent analytical structure on ambiguous problems. In contrast, performance on financial instruments like Indices and Funds is strong across most models, likely due to the availability of high-quality historical data and established analytical frameworks, which the agents can effectively leverage. The Jupyter NB agent, with its ability to perform quantitative analysis, shows its strength in these data-intensive categories, and its performance is further amplified by the Analytica framework.

## D.3    METHOD CONSENSUS OF ANALYTICA

Figure 11: Consensus matrix of final predictions across all methods. The color of each cell represents the pairwise agreement score between two methods, with darker colors indicating higher consensus.

Fig. 11 displays a consensus matrix, illustrating the degree of agreement in the final predictions among all evaluated methods. The matrix reveals distinct clusters of agreement. The various "thought" architectures (Tree, Graph, Forest) form a noticeable cluster, indicating that they often arrive at similar conclusions despite their different structural approaches to reasoning. This suggests they may share similar underlying reasoning patterns or biases inherited from the base LLM.

The Analytica variants, particularly those built on the same grounder (e.g., all `Deep Research + Analytica` versions), show very high consensus among themselves. This is expected, as they share the same foundational evidence from the grounder and differ only in the final synthesis step. A more insightful observation is the relatively high agreement between the top-performing models, `Deep Research + Analytica-L` and `Jupyter NB + Analytica-L`. This convergence among the best methods suggests that as performance and robustness increase, the models' conclusions become more aligned, likely approaching a more objectively correct analysis. The Vanilla, Simple Logic, and Linear synthesizers for a given grounder also form a tight cluster, which is a strong indicator that the decomposition and grounding phases are the most critical drivers of the final outcome, with the synthesis rule acting as a fine-tuning mechanism for accuracy and stability.

### D.4 Ablation on Base Models

To rigorously evaluate the impact of the underlying language model on the performance and efficiency of the Analytica framework, we conducted a series of ablation studies. We systematically varied the models assigned to the three core components: the Grounder, the Analyzer, and the Synthesizer.

### D.4.1 Setup

Our experiments utilize three distinct large language models, chosen to represent a spectrum of capabilities and design philosophies: 1. **o3 (`o3-2025-04-16`):** A state-of-the-art model optimized for *specialized reasoning*, serving as our high-performance benchmark. 2. **gpt-4.1 (Hypothetical Generalist):** A powerful, *general-purpose* model used to test the framework's effectiveness with a non-specialized but highly capable LLM. 3. **o4-mini (Hypothetical Cost-Effective Reasoner):** A cost-efficient and fast reasoning model to evaluate the framework's performance under significant resource constraints.

This selection allows us to measure not only how performance scales with model capability but also how robust the framework is to the specific architecture of its components. We run all configurations on our benchmark set with 100 events introduced in § 5.1. We use the Vanilla synthesis rules and basic search agents.

### D.4.2 Cost-Efficiency of Model Combinations

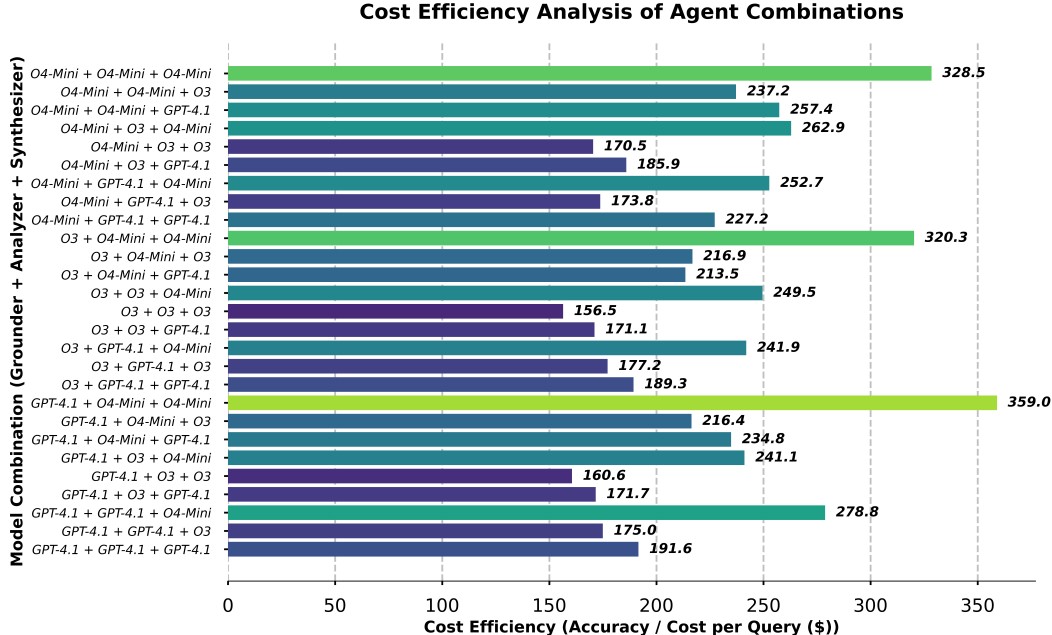

Figure 12: Cost-efficiency analysis of different model combinations for the Analytica components. The chart compares 27 configurations, varying the LLM for the Grounder, Analyzer, and Synthesizer roles. The length of the bar represents cost-efficiency, calculated as accuracy divided by cost.

In Fig. 12, we present a cost-versus-accuracy analysis for all 27 possible combinations of the three base models across the Grounder, Analyzer, and Synthesizer roles. The plot clearly illustrates the trade-off frontier between computational cost and predictive accuracy.

The results unequivocally show that the choice of the *Grounder* model is the most significant determinant of both cost and overall performance. Configurations using the powerful `o3` model for the grounding phase consistently form a cluster in the high-accuracy, high-cost quadrant. Conversely, using the economical `o4-mini` as the Grounder results in a cheaper but less accurate agent. The model choices for the Analyzer and Synthesizer have a more subtle effect, creating smaller performance variations within the distinct tiers established by the Grounder. This analysis serves as a practical guide, allowing users to select a configuration that aligns with their specific balance of performance requirements and resource constraints.

### D.4.3 BASE MODEL SELECTION FOR COMPONENTS

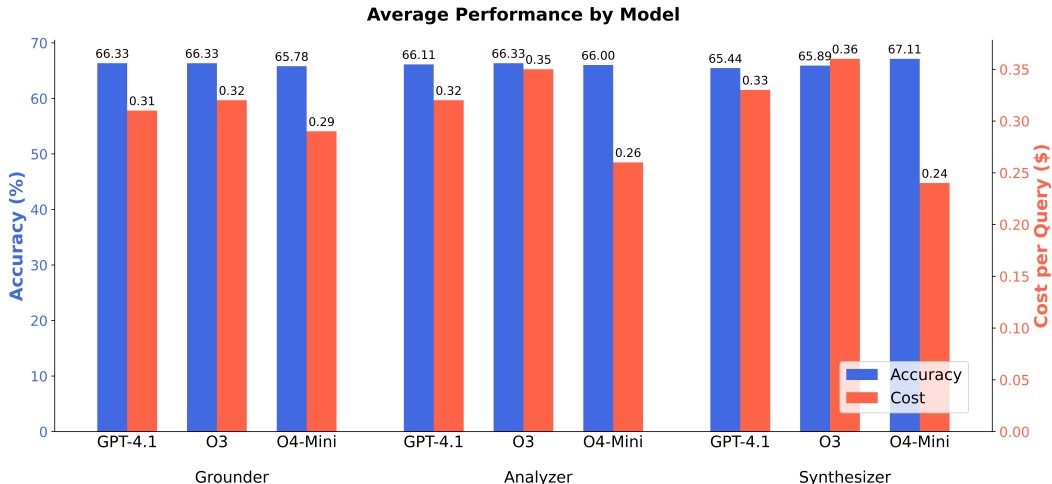

Figure 13: Marginal impact of model choice on component performance. The chart shows the average accuracy and cost when a specific model is used for a particular role (Grounder, Analyzer, Synthesizer), averaged across all other configuration choices.

Fig. 13 isolates the marginal impact of the base LLM for each of the three agent roles, with each data point representing an average over all configurations where that model was used in that specific role. The results indicate that the Analytica framework exhibits considerable robustness to the choice of model within this family. While there is a clear and expected trend where more capable models generally yield higher accuracy, the absolute differences in the final outcomes are modest. This low sensitivity suggests that the structured process of decomposition, grounding, and synthesis is a primary driver of performance, mitigating some of the variability that might arise from different model sizes or training objectives from the same provider.

This effect is particularly evident with the `o4-mini` model, which delivers performance that is competitive with its larger counterparts. This is a significant finding, as it suggests the framework's methodical approach—breaking a complex problem into a series of smaller, well-defined tasks—can effectively leverage more efficient models. By providing this structural "scaffolding", Analytica enables these smaller models to contribute to complex reasoning chains in a way that would be difficult in a less constrained, end-to-end setting.

While the framework is robust in this context, a hierarchy of influence among the components is still discernible. The choice of the **Grounder** model has the most pronounced impact on final accuracy, underscoring that the quality of foundational evidence is paramount. The system's graceful degradation in performance with less capable grounders, rather than outright failure, further supports the claim of architectural resilience. Besides, our model selections are from the same model family provided by OpenAI, while models from different providers may show a different pattern.

### D.4.4 PERFORMANCE ON OPEN-WEIGHT AND SMALL MODELS

| Model | Accu | *Imp.* | Soft | Hard | BS | Cost (1k) | # Params |
|---|---|---|---|---|---|---|---|
| **DeepSeek-v3.1** | 60.95 | - | 56.68 | 61.89 | 25.73 | $3 | 671B (37B) |
| + Analytica-L | 64.25 | 5.42 | 56.39 | 59.96 | 22.34 | $59 | - |
| **GLM-4.6** | 52.73 | - | 53.10 | 55.03 | 30.73 | $10 | 355B (32B) |
| + Analytica-L | 55.33 | 4.96 | 56.60 | 59.59 | 25.92 | $100 | - |
| **Kimi-K2-Thinking** | 55.56 | - | 54.44 | 58.39 | 30.59 | $25 | 1T (32B) |
| + Analytica-L | 56.81 | 2.26 | 59.73 | 63.15 | 25.59 | $284 | - |
| **Qwen3-Next-80B-Think.** | 53.64 | - | 54.88 | 58.79 | 33.97 | $10 | 80B (3B) |
| + Analytica-L | 55.55 | 3.56 | 58.36 | 59.65 | 32.77 | $104 | - |
| **OpenAI-OSS-120B** | 54.72 | - | 55.97 | 59.18 | 28.20 | $2 | 117B (5.1B) |
| + Analytica-L | 62.96 | 15.05 | 55.40 | 55.67 | 22.74 | $22 | - |
| **OpenAI-OSS-20B** | 55.57 | - | 54.79 | 59.24 | 29.56 | $1 | 21B (3.6B) |
| + Analytica-L | 64.24 | 15.59 | 56.91 | 58.59 | 23.68 | $7 | - |
| **GPT-5-mini** | 62.45 | - | 56.60 | 64.09 | 24.37 | $7 | N/A |
| + Analytica-L | 64.37 | 3.07 | 60.31 | 65.79 | 23.27 | $71 | - |
| **O4-mini** | 62.63 | - | 58.49 | 64.27 | 25.56 | $9 | N/A |
| + Analytica-L | 66.11 | 5.56 | 58.49 | 64.62 | 23.47 | $101 | - |

Table 6: Evaluating Analytica on small and open-weight models.

To further validate the generality and robustness of our framework, we broadened our evaluation to include a heterogeneous set of open-weight, cost-efficient small language models. The experimental configuration strictly adheres to the protocol outlined in §5.1. For each model, we conduct two runs: one using vanilla Basic Search and another employing *Analytica-Linear* with the Basic Search grounder.

The results in Table 6 indicate a consistent performance gain across all evaluated models, demonstrating that the benefits extend to both open-source systems and smaller architectures. The largest relative gains occur in compact, distilled models, thereby helping to democratize advanced reasoning capabilities; for example, **OpenAI-OSS-20B** enhanced with Analytica attains performance comparable to the baseline of the substantially larger 671B-parameter **DeepSeek-v3.1**. This implies that Analytica can substantially narrow the capability gap between efficient edge models and large frontier models. Furthermore, the findings suggest that Analytica's effectiveness is only weakly dependent on model size (i.e., parameter count) and is instead primarily governed by the underlying **pre-training and post-training procedures**, which shape how well a model aligns with the structured decomposition tasks required by Analytica.

### D.5 EVALUATION ON SCIENTIFIC CLAIMS

To assess domain transferability beyond finance, economics, and predictive markets, we further evaluate our method in the Matter-of-Fact (MoF) benchmark (Jansen et al., 2025a). We perform zero-shot evaluation on the test set of MoF, which includes a large set of 4.4k binary scientific claims from superconductors, semiconductors, batteries, and aerospace materials publications, and involving qualitative and quantitative claims from theoretical, experimental, and code/simulation topics.

For each instance, an agent receives a single claim and is required to output the probability $P_{true}$ that the claim is correct. A decision threshold $\delta$ is then applied: if $P_{true} > \delta$, the claim is labeled as True; otherwise, it is labeled as False. For each model, we calibrate the threshold $\delta$ on the MoF validation set, which contains 1.4k claims. Concretely, we first collect the predicted $P_{true}$ values for all validation claims, then search for the threshold that yields the highest overall accuracy, and finally use this threshold on the test set. Our evaluation includes GPT-4o-mini and O4-mini, as reported in the original paper, and additionally the two most recent models, GPT-5.1 and GPT-5-mini. Each model is evaluated under a standard Basic Search configuration and under an Analytica-

Linear configuration using Basic Search grounders. We use each claim's publication date as the cutoff for the searches. Following Jansen et al. (2025a), we report both overall and per-category accuracy, as well as the associated costs.

| Model | Overall Accu. | True | False | Accuracy by Category Qual. | Qnt. | Exp. | Code | Ther. | Int. | Cost (×1k) |
|---|---|---|---|---|---|---|---|---|---|---|
| **Random** | 0.50 | 0.50 | 0.50 | 0.50 | 0.50 | 0.50 | 0.50 | 0.50 | 0.50 | 0 |
| **GPT-4o-mini** | 0.66 | 0.90 | 0.42 | 0.72 | 0.72 | 0.68 | 0.63 | 0.61 | 0.58 | $1 |
| + Analytica-L | 0.59 | 0.87 | 0.30 | 0.55 | 0.58 | 0.59 | 0.59 | 0.60 | 0.60 | $7 |
| **O4-mini** | 0.61 | 0.34 | 0.88 | 0.60 | 0.57 | 0.63 | 0.62 | 0.59 | 0.63 | $6 |
| + Analytica-L | 0.64 | 0.95 | 0.30 | 0.57 | 0.60 | 0.73 | 0.62 | 0.68 | 0.66 | $38 |
| **GPT-5.1** | 0.62 | 0.27 | 0.97 | 0.58 | 0.56 | 0.59 | 0.69 | 0.66 | 0.62 | $9 |
| + Analytica-L | 0.70 | 0.85 | 0.55 | 0.73 | 0.75 | 0.71 | 0.67 | 0.70 | 0.64 | $78 |
| **GPT-5-mini** | 0.71 | 0.60 | 0.82 | 0.69 | 0.76 | 0.66 | 0.64 | 0.75 | 0.72 | $3 |
| + Analytica-L | 0.73 | 0.70 | 0.75 | 0.75 | 0.77 | 0.69 | 0.67 | 0.79 | 0.67 | $27 |

Table 7: Experiment results for scientific claims on the Matter-of-Fact benchmark.

The results, summarized in Table 7, demonstrate that Analytica's structured reasoning framework generally generalizes effectively to scientific domains, though with notable exceptions. We observed significant performance uplifts for several architectures; for instance, the **GPT-5.1** model improved from 62% to 70% accuracy, and **GPT-5-mini** saw gains from 71% to 73%. However, the impact was not universally positive: **GPT-4o-mini** experienced a performance regression, dropping from 0.66 to 0.59 accuracy, which is also the only negative case we observed in our experiments. A plausible explanation is that this is the oldest model among all the base models evaluated in this work and may lack the capacity required for robust complex reasoning. Nevertheless, the consistent improvement across the majority of tested models validates the broader utility of the framework in the scientific domain.

### D.6 ERROR ANALYSIS

To better understand the conditions under which our methods succeed or fail, we conducted a detailed error analysis.

#### D.6.1 TASK CORRECTNESS DISTRIBUTIONS

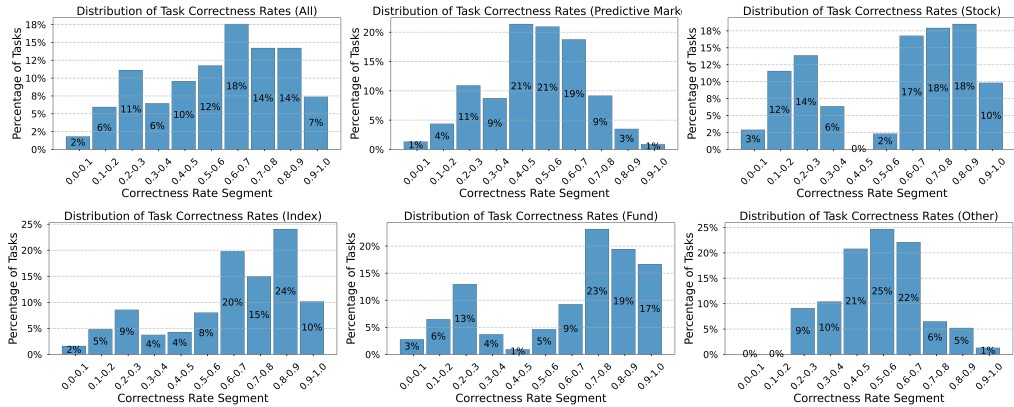

Figure 14: Distribution of task correctness rates across all models for different categories. The histograms show the percentage of tasks falling into different correctness rate buckets.

Fig. 14 (task correctness plots) shows the distribution of prediction correctness across different task categories. The distributions for financial tasks (Stock, Index, Fund) tend to be more concentrated towards higher correctness scores, especially for the top-performing models. In contrast, the distribution for Predictive Markets is flatter and more spread out, confirming that these tasks are more challenging and that model performance is less consistent. This visual analysis reinforces the finding that the models are more reliable in domains with structured data and established patterns.

### D.6.2 TASK FEATURES AND CORRECTNESS

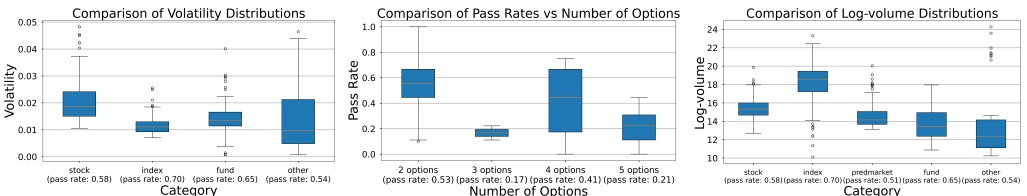

Figure 15: Correlation between task features and model performance. The boxplots show that higher asset volatility (left) and lower market volume (right) are associated with lower pass rates.

The boxplots in Fig. 15 explore the relationship between task characteristics and model performance. We observe a negative correlation between the volatility of a financial asset and the models' prediction accuracy. This is intuitive: highly volatile assets are inherently less predictable. Similarly, for predictive markets, events with a shorter time horizon (less time to gather information and for trends to stabilize) and lower market volume (less collective wisdom to draw upon) are associated with lower accuracy. This suggests that the models' performance is sensitive to the inherent uncertainty and information scarcity of the task.

### D.6.3 TOP AND BOTTOM PERFORMING TASKS

Finally, the tables listing the top and bottom 10 performing tasks provide qualitative insights.

| Top 10 Pass rate | Bottom 10 Pass rate |
| --- | --- |
| MSFT shareholders vote for Bitcoin investment? | Who will be Trump's Secretary of Labor? |
| Will Biden announce resignation by July 31? | Trump gets more black voters than in 2020? |
| Will Biden speak at the DNC? | Fed decision in January? |
| Will Trump be Speaker by January 1? | Will Ukraine hold Kursk through Aug 31? |
| Trump and Biden debate before Election? | Ukraine agrees to Trump mineral deal? |
| Trump nominates Elon Musk to Cabinet? | Yoon arrested by Friday? |
| Will the US confirm that aliens exist in 2024? | Trump and Harris agree to Sept 10 debate? |
| Will Kanye launch a coin in February? | Will Assad remain President of Syria? |
| Will Kamala Harris win all 6 swing states? | Which states will move to the right? |
| US inauguration on January 20? | Romania Parliamentary Election |

Table 8: Top and Bottom 10 performing events in predictive markets.

**Predictive Markets** For predictive markets, the models excel at forecasting high-profile, binary events, particularly within the realm of US politics. The Top 10 tasks are questions about major political figures like Joe Biden and Donald Trump, revolving around widely publicized events such as debates and convention speeches. These topics generate a massive volume of news articles, social media chatter, and opinion polling, creating a dense information environment where sentiment and likelihood can be effectively gauged by synthesizing public discourse.

The models fail when faced with questions that require more nuanced, specialized, or multifaceted reasoning. The Bottom 10 list includes tasks that involve predicting specific cabinet appointments, complex voter demographic shifts, or the outcomes of intricate geopolitical conflicts. These ques-

tions cannot be answered by simply aggregating headlines; they require a deeper, causal understanding of political systems and human behavior, which remains a significant challenge.

| Top 10 Pass rate | Bottom 10 Pass rate |
|---|---|
| MetLife, Inc. (MET) | Target Corporation (TGT) |
| Johnson & Johnson (JNJ) | AstraZeneca PLC (AZN) |
| Microsoft Corporation (MSFT) | ConocoPhillips (COP) |
| AbbVie Inc. (ABBV) | Chevron Corporation (CVX) |
| Take-Two Interactive Software, Inc. (TTWO) | PACCAR Inc (PCAR) |
| Bristol-Myers Squibb Company (BMY) | Micron Technology, Inc. (MU) |
| Intuitive Surgical, Inc. (ISRG) | MongoDB, Inc. (MDB) |
| Mondelez International, Inc. (MDLZ) | Synopsys, Inc. (SNPS) |
| Axon Enterprise, Inc. (AXON) | UnitedHealth Group Incorporated (UNH) |
| MercadoLibre, Inc. (MELI) | Alphabet Inc. (GOOGL) |

Table 9: Top and Bottom 10 performing stocks.

**Stocks** For individual stock (Table 9), the models demonstrate a clear preference for large, established companies with extensive public records and relatively stable business models. The Top 10 performers include blue-chip names from diverse sectors such as insurance (MetLife), pharmaceuticals (Johnson & Johnson, AbbVie), technology (Microsoft), and consumer goods (Mondelez). These companies are heavily covered by financial analysts and news media, providing a rich and consistent stream of information for the agents to process. Their performance is often driven by broad economic trends and predictable business cycles, which align well with the models' ability to synthesize macroeconomic data.

Conversely, the Bottom 10 list is populated by companies whose performance is tied to more volatile, cyclical, or speculative factors. This includes energy giants (ConocoPhillips, Chevron) subject to commodity price swings, semiconductor firms (Micron Technology) in a notoriously cyclical industry, and high-growth tech companies (MongoDB, Synopsys) whose valuations are sensitive to shifting market sentiment and competitive pressures. Even large, stable companies like Target and Alphabet appear here, suggesting that factors like consumer spending shifts or complex regulatory challenges can introduce a level of unpredictability that is difficult for the models to capture.

| Top 10 Pass rate | Bottom 10 Pass rate |
|---|---|
| HANG SENG INDEX (^HSI) | IDX30 Index (IDX30) |
| Oslo Bors All-Share Index (^OSEAX) | Thailand Stock Exchange Index (SET) |
| New Zealand Exchange Index (^NZ50) | Copenhagen Exchange Index (OMXC20) |
| ASX 200 Telecommunication (^AXTJ) | NASDAQ Biotechnology (^NBI) |
| Toronto Stock Exchange Index (TSX60) | Nikkei 225 (^N225) |
| MSCI World Index (MSCIWORLD) | BIST Food Beverage Index (XGIDA.IS) |
| Intuitive Surgical, Inc. (ISRG) | S&P/ASX 200 Energy (^AXEJ) |
| Dow Jones U.S. Semiconductors (^DJUSSC) | Malaysia Stock Exchange Index (KLSE) |
| Australia Stock Exchange Index (ASX200) | S&P Biotechnology Select Indust (^SPSIBI) |
| S&P Global 100 (^OOI) | BIST Tourism Index (XTRZM) |

Table 10: Top and Bottom 10 performing indices.

**Indices** Table 10 reveals that the models are most successful when forecasting broad, diversified, major market indices. The Top 10 list is dominated by global or major national benchmarks like the MSCI World Index, Australia's ASX200, and Hong Kong's Hang Seng Index. These indices reflect aggregate economic activity and are driven by macroeconomic narratives that are widely discussed and debated in public forums, making them ideal subjects for LLM-based analysis.

The models struggle significantly with more specialized or volatile indices. The Bottom 10 list features sector-specific indices in notoriously unpredictable fields like Biotechnology (N̂BI, ŜPSIBI)

and Energy (ÂXEJ). It also includes indices from smaller or emerging markets (Thailand, Malaysia, Indonesia), which may be influenced by local political and economic factors that are less covered by the global information sources the models primarily rely on. This indicates a gap in handling niche domains and region-specific complexities.

| Top 10 Pass rate | Bottom 10 Pass rate |
| --- | --- |
| Capital Group Dividend Value ETF (CGDV) | Global X Copper Miners ETF (COPX) |
| Fidelity Total Bond ETF (FBND) | iShares U.S. Home Construction ETF (ITB) |
| iShares Global Infrastructure ETF (IGF) | Direxion Semiconductor Bull 3X (SOXL) |
| The Industrial Select Sector SPDR Fund (XLI) | iShares Global Clean Energy ETF (ICLN) |
| Vanguard Communication Services ETF (VOX) | iShares Biotechnology ETF (IBB) |
| BlackRock U.S. Carbon Transition ETF (LCTU) | Global Upstream Natural Resources (GUNR) |
| First Trust NASDAQ-100-Technology (QTEC) | The Energy Select Sector SPDR Fund (XLE) |
| Vanguard Global ex-U.S. Real Estate (VNQI) | JPM Nasdaq Equity Premium Income (JEPQ) |
| VanEck Gold Miners ETF (GDX) | Dimensional U.S. Targeted Value ETF (DFAT) |
| Invesco Aerospace & Defense ETF (PPA) | Vanguard Materials Index Fund ETF (VAW) |

Table 11: Top and Bottom 10 performing funds.

**Funds** Similar to the stock and index categories (Table 11), the models perform best with broad, diversified, and well-established ETFs. The Top 10 includes funds representing core sectors of the economy, such as Industrials (XLI), Infrastructure (IGF), and Communication Services (VOX), as well as bond funds (FBND) and funds tracking precious metals (GDX). These investment vehicles are generally less volatile than individual stocks, and their performance is tied to clearer, more persistent macroeconomic trends.

The Bottom 10 is almost exclusively composed of highly cyclical, thematic, or leveraged ETFs. This includes funds focused on volatile sectors like Copper Miners (COPX), Home Construction (ITB), Clean Energy (ICLN), and Biotechnology (IBB). The inclusion of a 3x leveraged semiconductor ETF (SOXL) is particularly telling, as these instruments are designed for short-term trading and are extremely sensitive to market volatility, making long-term forecasting exceptionally difficult.

| Top 10 Pass rate | Bottom 10 Pass rate |
| --- | --- |
| TRON (TRXUSD) | Solana (SOLUSD) |
| Micro Gold Futures (MGC) | Micro E-mini Russell 2000 Futures (RTY) |
| Dogecoin (DOGEUSD) | Ethereum (ETHUSD) |
| Mini DJI Index Futures (YMUSD) | Artificial Liquid Intelligence (ALIUSD) |
| E-Mini S&P 500 Futures (EMUSD) | Brent Crude Oil (BZUSD) |
| XRP (XRPUSD) | AUD/EUR (AUDEUR) |
| Feeder Cattle Futures (GFUSX) | Rough Rice Futures (ZRUSD) |
| Soybean Oil Futures (ZLUSX) | USD/CNY (USDCNY) |
| Bitcoin (BTCUSD) | NZD/USD (NZDUSD) |
| Micro Silver Futures (SILUSD) | CHF/CAD (CHFCAD) |

Table 12: Top and Bottom 10 performing commodity, forex, or crypto.

**Other (Commodity, Forex, Crypto)** In Table 12, performance is mixed, but a pattern emerges. The Top 10 includes futures contracts on major stock indices (E-Mini S&P 500, Mini DJI), which are driven by the same broad market sentiment that makes the underlying indices predictable. It also includes some of the largest and most discussed cryptocurrencies (Bitcoin, XRP, Dogecoin), where the sheer volume of online discourse may provide sufficient signal for the models to latch onto.

The Bottom 10 list highlights the difficulty of forecasting assets driven by complex and interlocking global factors. It features several currency pairs (AUD/EUR, USD/CNY, NZD/USD), whose movements are determined by the interplay of multiple national economies' monetary policies, trade balances, and political stability. It also includes volatile commodities like Brent Crude Oil and Rough

Rice, alongside major but notoriously volatile cryptocurrencies like Ethereum and Solana, reinforcing the conclusion that high intrinsic volatility remains a primary obstacle to accurate forecasting.

## D.7 RESYNTHESIS

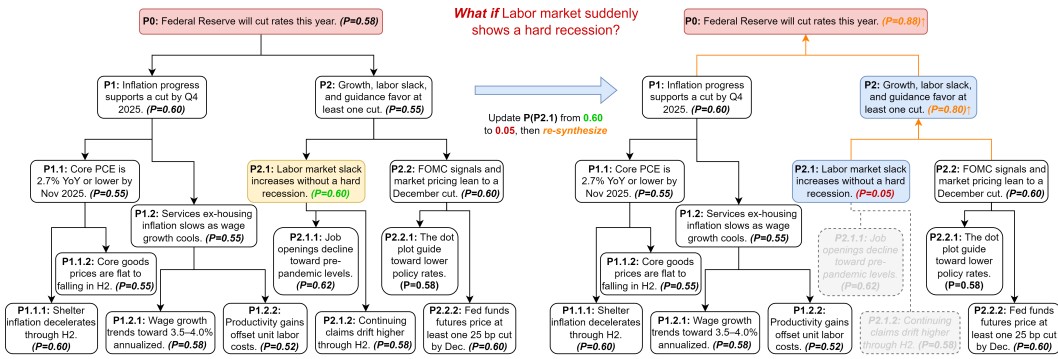

Figure 16: An example of the resynthesis feature for "what-if" scenario analysis. An analyst manually changes the probability of a leaf node (P2.1) to reflect a counterfactual assumption. The framework efficiently recalculates only the affected branch, providing a rapid update to the root proposition's probability and quantifying the impact of the change.

A key feature of the Analytica framework is its support for efficient, interactive "what-if" scenario analysis via a process called **resynthesis**. As described in § 4.1, once a proposition tree is fully grounded and synthesized, a user can manually alter the truth value of any node to explore a counterfactual scenario. The framework's locality principle ensures that only the affected branch of the tree needs to be re-synthesized, making the process computationally inexpensive.

Fig. 16 provides a concrete example of this process. The initial analysis (left) of the proposition "Federal Reserve will cut rates this year" results in a probability of 0.58. An analyst wishing to test the system's sensitivity to labor market conditions can pose the counterfactual: "What if the Labor market suddenly shows a hard recession?". This is implemented by manually changing the probability of the relevant leaf node, P2.1, from its original value to 0.0, reflecting the new assumption. The Resynthesis process is triggered, and the new probability is propagated up its branch. The truth values of unaffected nodes (like P1) remain unchanged. The fast recalculation yields a new root probability of 0.48, providing an immediate quantitative measure of the labor market's impact on the overall forecast. This capability transforms Analytica from a static forecasting tool into a dynamic environment for decision-making and risk assessment.

|            | Random | Basic Search | Deep Research | Jupyter Notebook |
|------------|--------|--------------|---------------|------------------|
| *w/o Analytica* | *48.10* | *53.94*  | *63.04*       | *61.96*          |
| Linear     | -      | 65.62        | 71.06         | 70.11            |
| Average    | -      | 63.59        | 67.39         | 66.71            |
| Random     | -      | 62.09        | 66.44         | 65.90            |

Table 13: Replace the weights with random values, or make the models an unweighted average.

## D.8 SYNTHESIS RULES

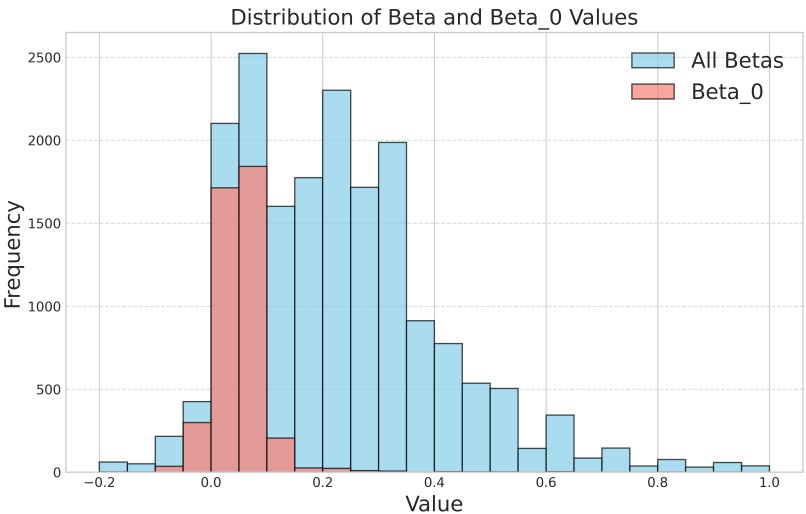

Figure 17: Distribution of the learned weights ($\beta_j$) and intercept ($\beta_0$) for the Linear synthesis rule. The concentration of weights at low positive values demonstrates the rule's noise-dampening property, as formally proven in § A.1.

**Linear Rule** The stability of the Linear rule, $P = \beta_0 + \sum \beta_j C_j$, is predicated on its ability to act as a weighted average that dampens noise from its inputs. Our experiments confirm that the Synthesizer learns to implement this property. Fig. 17 shows the distribution of all learned child weights ($\beta_j$) and intercept terms ($\beta_0$) across our experiments. The child weights are predominantly positive and concentrated in the $[0, 0.5]$ range, ensuring that no single child proposition has an outsized impact and that errors are smoothed rather than amplified. The intercept term, $\beta_0$, is tightly centered around zero, indicating that the agent relies primarily on the evidence provided by the child propositions rather than a strong independent bias. This behavior is encouraged by constraining the intercept's absolute value (e.g., $|\beta_0| < 0.1$), which prevents the model from ignoring the grounded evidence.

In Table 13, we replace all the weights by random numbers ('Average'), or degrade the linear rule to a simple unweighted average ('Average') while removing the intercept, and compare it with the Analytica with the linear rule for different grounder and the grounder itself without Analytica. The degraded performance shows that the linear weights provide an informative ensemble of evidence.

| Formula | Assumption | PA |
|---|---|---|
| (P1.1.1) OR (P1.1.2 AND P1.1.3) OR (P1.1.2 AND PA) | Fuel costs stay stable, capacity constraints are manageable, and Fed policy avoids a significant slowdown. | 0.65 |
| (P1 AND P2 AND P3 AND PA) | No unmodeled political or platform developments (e.g., surprise rule changes, debate-definition di sputes, or market technical disruptions) will materially alter P1, P2, or P3. | 0.8 |
| (P1 AND P2 AND P3) OR PA | Residual catalysts (ETF flows, China reopening travel, idiosyncratic M&A) can occasionally override modeled drivers to make long KWEB best even if one core condition fails. | 0.05 |
| (P1 AND P2 AND P3 AND PA) | No large unmodeled regime shifts or market-structure shocks beyond those captured in P1–P3. | 0.85 |
| (P1.1 AND P1.2 AND P1.3 AND P1.5 AND NOT P1.4) OR PA | Additional supportive factors (e.g., fiscal policy, corporate buybacks, cross-border flows) remain in place, offsetting valuation headwinds. | 0.7 |
| (P1.1 AND P1.3 AND P1.4 AND (P1.2 OR PA)) | No major geopolitical or macroeconomic shocks will trigger a risk-off shift in market sentiment. | 0.75 |
| (P1 AND P2) AND PA | Assumes that platform execution, liquidity, and unmodeled market/political tail-risks do not invalidate P1 or P2 before resolution. | 0.8 |
| (P1.1 AND P1.2) OR (P1.5 AND P1.6) OR (P1.3 AND P1.4 AND PA) | PA captures other supporting factors (global risks, post-election fiscal changes) that bolster a hike under loose financial and expansionary fiscal conditions. | 0.5 |
| P1 AND (P2 OR PA) | Residual factors (e.g. large M&A, extraordinary buybacks, regulatory shifts in agriculture, geopolitical events) may still tip returns in favor of long DE even if short's modeled inferiority weakens. | 0.1 |
| (P1 AND P2 AND PA) | No extreme political shocks, contract redesigns, or acute liquidity dislocations occur to invert the EV or risk-adjusted ranking of the PC contract relative to alternatives. | 0.85 |

Table 14: Random real examples of logical formulas, assumption descriptions, and assumption probabilities (PA) generated by the **Simple Logic** Synthesizer agent.

**Simple Logic Rule**    Table 14 presents a selection of formulas generated by the agent in our experiments. These examples showcase how the agent uses a combination of AND, OR, and NOT operators, along with the PA variable, to build a causal or evidential case for the parent proposition. For instance, a proposition might be true if several core conditions are met (P1 AND P2 AND PA) or if one of several alternative scenarios occurs ((P1.1 AND P1.2) OR (P1.3 AND PA)).

# E PROMPTS

## E.1 ANALYZER

> **System Prompt for Analyzer**
>
> You are an expert logical strategist and project manager for a team of advanced research agents (grounders) in financial, economic, business, social, and political analysis. Your primary mission is to decompose a complex 'Proposition to Analyze' into a tree of financial, economic, business, social, and political propositions, where the truthfulness of the parent proposition is based on the truthfulness of the children.
>
> You should firstly give an analysis and planning of how to decompose the proposition, and explain your framework of analysis, use the professional knowledge and analysis framework in financial, economic, business, social, and political analysis. You are encouraged to apply professional analysis framework that are used in academia or industry. Please refer to them in your analysis. A proposition tree is like the following:
>
> - Parent Proposition
>   - Child Proposition 1
>     * Child Proposition 1.1
>     * Child Proposition 1.2
>     * ...
>   - Child Proposition 2
>     * Child Proposition 2.1
>     * Child Proposition 2.2
>     * ...
>   - ...
>
> You should progressively derive it in your analysis. Then, provide your proposition tree in a list of JSON objects wrapped in a single ```` ```json ...    ``` ```` in the following format:
>
> ```json
> [
>   {
>     "parent":  "proposition_id", # the id of the parent
> proposition
>     "children":  {
>       "proposition_id":  "proposition_content", # the content
> of the child proposition
>       ...
>     },
>     "causality":  "..." # the causality of how the children
> lead to, imply, support, or impact the parent proposition
>   },
>   {...}
> ]
> ```
>
> Each JSON object should be a single proposition and its children. You should mark each child with a *unique* proposition_id. The id of input proposition is always "P0". Then mark the children with "P1", "P2", "P1.2", "P1.3.2.1", etc. It is not allowed to reuse the same proposition_id for different propositions. The propositions should form a tree structure rooted at "P0".
>
> **Notes**:
>
>   1. A proposition is a single sentence statement, with financial, economic, business, social, and political meaning that can be associated with a boolean value True or

False. The decomposition should illustrate the causal relation that how children factors lead to, imply, support, or impact the truthfulness of the parent proposition.

2. The decomposed propositions should be self-contained, not dependent on the parent proposition. Which means it can be understood without the parent proposition as context. For example, it should not refer to the parent proposition using terms like "it", "this metric", "this event", etc.

3. You are not expected to decompose the proposition into low-level fine-grained propositions. Instead, it is ideal to decompose the proposition into high-level and meaningful financial, economic, business, social, and political factors, assumptions, hypotheses, etc.

4. You should keep the tree to be in-depth but not redundant, this means that you do not need to create commonsense as a child proposition. You can have some compromise on rigorousness, the key is to illustrate clear, indepth and professional analysis.

5. Try to provide really insightful information from your analysis and the outcome decomposition tree that creates "alpha" for the user. Think comprehensively, deeply, and professionally. You are encouraged to give a really deep analysis and very deep decomposition tree.

6. Do not make redundant propositions, such as the rewrite of the same proposition or the ones that can be simply derived from the negation of other children.

**Ideal Decomposition** *(Example for Linear rule)***:**

Ideally, a parent proposition can be represented as a multiple linear combination of its children's propositions, i.e. P_true = beta_0 + beta_1*P_true1 + beta_2*P_true2 + ... + beta_n*P_true_n + eps, where P_true is the probability of the parent proposition being true; beta_0 is the intercept, representing a bias probability of the parent proposition being true that reflects the information beyond the children propositions; beta_1, beta_2, ..., beta_n are the weights of the children's P_true; and eps is the error term. As a result, an ideal set of child propositions should be less correlated with each other, and representing the most dominant factors that affect the parent proposition. Left those less important factors to the parent proposition as the intercept and explain your decisions in your analysis process.

## E.2 SYNTHESIZER

### System Prompt for Synthesizer (Vanilla)

You are an expert for a team of advanced research agents (grounders) in financial, economic, business, social, and political analysis. The grounders have access to external databases and information sources that support their analysis. They also possess qualitative and quantitative analysis skills using Jupyter Notebook to help them analyze the proposition.

Your task is to aggregate the analysis of children's propositions from the grounders, and estimate the probability of truthfulness (P_true) of a proposition based on their proofs and estimated P_true. Each proposition contains a financial, economic, business, social, or political statement, which can be associated with a boolean value, True or False, represented as a float number P_true between 0 and 1, where 0 means False and 1 means True. And they are decomposed into a set of child propositions that may have causal, evidential, or other relationships with the parent proposition, which are already analyzed by the grounders.

You will need to use your professional skills and analytical frameworks in financial, economic, business, social, and political analysis to estimate the P_true of the parent proposition, with a comprehensive, natural language "Proof" that explains your entire reasoning process, which proves or disproves the parent proposition that supports your

estimated P_true.

### INSTRUCTIONS

You will receive a JSON object with the parent proposition information and its children, along with their proofs and P_true. First, write a comprehensive and in-depth "Proof" that explains your entire reasoning process. Then, reweight the children's proposition factors based on your confidence in their proofs and their importance to the parent proposition. You are also required to provide a risk assessment of the proposition, and explain the risk factors that might lead to the proposition being false. Please make your analysis more specific and detailed as possible, do not miss any important information especially the data and evidence from the grounders. You are encouraged to present the data and evidence in a table and other visualizations. Finally, synthesize the probability of the parent proposition based on the reweighted children factors, and provide your conclusion in a single ```json ... ``` in the following format:

```json
{
  "p_true":  <float> # the probability of the proposition
being true, between 0 and 1
  "key_factor":  <string> # the key factors that why the
proposition likely to be true or false, one or two sentences
}
```

### NOTES

1. You are encouraged to use the knowledge and theory from academia or industry and cite them in your proof.

2. You need to think beyond the given data and provide a more comprehensive, in-depth, and broad analysis especially for the points that might be omitted by the grounders.

3. It is also your task to check the consistency of the children's proofs and their P_true, as well as the quality of the proofs themselves.

---

### System Prompt for Synthesizer (Linear)

You are an expert for a team of advanced research agents (grounders) in financial, economic, business, social, and political analysis. The grounders have access to external databases and information sources that support their analysis. They also possess qualitative and quantitative analysis skills using Jupyter Notebook to help them analyze the proposition.

Your task is to aggregate the analysis of children's propositions from the grounders, and estimate the probability of truthfulness (P_true) of a proposition based on their proofs and estimated P_true. Each proposition contains a financial, economic, business, social, or political statement, which can be associated with a boolean value, True or False, represented as a float number P_true between 0 and 1, where 0 means False and 1 means True. And they are decomposed into a set of child propositions that may have causal, evidential, or other relationships with the parent proposition, which are already analyzed by the grounders.

You will need to use your professional skills and analytical frameworks in financial, economic, business, social, and political analysis to estimate the P_true of the parent proposition, with a comprehensive, natural language "Proof" that explains your entire reasoning process, which proves or disproves the parent proposition that supports your estimated P_true. The P_true of the parent proposition is represented as a multiple linear

combination of the children's P_true, i.e. P_true = beta_0 + beta_1*P_true1 + beta_2*P_true2 + ... + beta_n*P_true_n + eps, where beta_0 is the intercept, representing a bias probability of the parent proposition being true based on your own knowledge and analysis; beta_1, beta_2, ..., beta_n are the weights of the children's P_true, based on your confidence of the value from their proofs and your judgment of their importance to the parent proposition; and eps is the error term.

### INSTRUCTIONS

You will receive a JSON object with the parent proposition information and its children, along with their proofs and P_true. First, write a comprehensive and in-depth "Proof" that explains your entire reasoning process. You are also required to provide a risk assessment of the proposition, and explain the risk factors that might lead to the proposition being false. Then, analyze the weights of the children's proposition factors based on your confidence in their proofs and your judgment of their importance to the parent proposition. Please make your analysis more specific and detailed as possible, do not miss any important information especially the data and evidence from the grounders. You should try to compute the resulting P_true of the parent proposition based on the weights and the intercept you derived and iteratively refine them, to make sure the final P_ture is reasonable and valid (between 0 and 1). The eps is usually small and can be ignored, do not include it in your final result. You are encouraged to present the data and evidence in a table and other visualizations. Finally, provide your conclusion in a single ```json ...    ``` in the following format:

```json
{{
  "beta":  {{
    "beta_0":  <float>, # the intercept, the key must be
"beta_0"
    "<child_proposition_id>":  <float>, # for example, "P1",
"P2", "P1.1", "P1.2", etc.
    ...  # the weights of the children's proposition factors,
all children must be included
  }}
  "key_factor":  <string> # the key factors that why the
proposition likely to be true or false, one or two sentences
}}
```

### NOTES

1. You are encouraged to use the knowledge and theory from academia or industry and cite them in your proof.

2. You need to think beyond the given data and provide a more comprehensive, in-depth, and broad analysis especially for the points that might be omitted by the grounders, they are the core factors you need to consider in deriving the intercept, the intercept can be seen as an assumption of those omitted factors and the risk factors, remember to clearly state those assumptions in your proof and explain how they affect the intercept.

3. The weights do not necessary to be from 0 to 1, it can be any real number, and the intercept beta_0 can be negative, but its absolute value should be less than {{abs_intercept_max}}, the final P_true after the weights and intercept are applied must be between 0 and 1. Please compute yourself first to make sure the final P_true is valid before providing your conclusion in the JSON block.

---

**System Prompt for Synthesizer (Simple Logic)**

You are an expert for a team of advanced research agents (grounder) in financial, economic, business, social, and political analysis. The grounders have access to external databases and information sources that support their analysis. They also possess qualitative and quantitative analysis skills using Jupyter Notebook to help them analyze the proposition.

Your task is to aggregate the analysis of children's propositions from the grounders, and estimate the probability of truthfulness (P_true) of a proposition based on their proofs and estimated P_true. Each proposition contains a financial, economic, business, social, or political statement, which can be associated with a boolean value, True or False, represented as a float number P_true between 0 and 1, where 0 means False and 1 means True. And they are decomposed into a set of child propositions that may have causal, evidential, or other relationships with the parent proposition, which are already analyzed by the grounders.

You will need to use your professional skills and analytical frameworks in financial, economic, business, social, and political analysis to estimate the P_true of the parent proposition, with a comprehensive, natural language "Proof" that explains your entire reasoning process. The P_true of the parent proposition is represented as a logical combination of the children's P_true, i.e. P_true = (P_true1 AND P_true2) OR (P_true3 AND P_true4) OR ... OR (P_true_n-1 AND P_true_n), where P_true1, P_true2, ..., P_true_n are the probabilities of the children propositions being true. Here we use a probabilistic logic to represent the logical combination, where a logical AND of P_true1 and P_true2 is represented as: P_true1 AND P_true2 = P_true1 * P_true2; a logical OR of P_true1 and P_true2 is represented as: P_true1 OR P_true2 = P_true1 + P_true2 - P_true1 * P_true2; and a logical NOT of P_true1 is represented as: NOT P_true1 = 1 - P_true1, you can also use NOT to negate the parentheses.

### INSTRUCTIONS

You will receive a JSON object with the parent proposition information and its children, along with their proofs and P_true. First, write a comprehensive and in-depth "Proof" that explains your entire reasoning process. You are also required to provide a risk assessment of the proposition, and explain the risk factors that might lead to the proposition being false. Then, analyze the logical combination of the children's proposition factors based on your confidence in their proofs and your judgment of their importance to the parent proposition. Please make your analysis more specific and detailed as possible, do not miss any important information especially the data and evidence from the grounders. Specially, you can include a special assumption variable to capture the less important factors, and include it in the formula. Notice that the assumption variable id in the formula should ALWAYS BE "PA", and all the other variables in the formula should be the proposition_id of the children in the input proposition information. You should use ALL the children propositions in the formula, and the formula should be a valid logical combination of the children's P_true. You are encouraged to present the data and evidence in a table and other visualizations. Finally, provide your conclusion in a single ```json ... ``` in the following format:

```json
{{
  "formula": <string>, # e.g., (P1 AND P2) OR (P3 AND NOT
PA)
  "assumption": {{
    "detail": <string>, # detailed assumptions, one or two
sentences
    "probability": <float>, # the probability of the
assumption being true, between 0 and 1
  }}
  "key_factor": <string> # key factors, one or two sentences
}}
```

```
### NOTES

1. You are encouraged to use the knowledge and theory from academia or industry
   and cite them in your proof.

2. You need to think beyond the given data and provide a more comprehensive, in-
   depth, and broad analysis especially for the points that might be omitted by the
   grounders, they are the core factors you need to consider in deriving the formula,
   remember to clearly state those assumptions in your proof and explain how they
   affect the formula.

3. The assumption variable id in the formula should ALWAYS BE "PA", and all the
   other variables in the formula should be the proposition_id of the children in the
   input proposition information.

4. You should use ALL the children propositions in the formula, and the formula
   should be a valid logical combination of the children's P_true.

5. You are encouraged to present the data and evidence in a table and other visualiza-
   tions.

6. The available operators include AND, OR, NOT and parentheses.
```

### E.3 GROUNDER

> **General Grounder Prompt**
>
> You are an expert in financial, economic, business, social, and political analysis. You
> will be provided with a proposition, and your task is to provide a comprehensive proof
> that either proves or disproves the proposition. It should include the bullet points of your
> analysis, such as the key findings, data, evidence, and quantitative analysis. Please be more
> specific and detailed as possible, do not miss any important information especially the data
> and evidence. You are encouraged to present the data and evidence in a table and other
> visualizations.
>
> After you have written the complete textual proof, append a single ```json ...   ```
> block. Inside this block, provide a single JSON object with exactly two keys:
>
> 1. `"p_true"`: Your estimated probability (a float between 0.00 and 1.00) that the
>    proposition is true, based on your proof and notebook analysis.
>
> 2. `"key_factor"`: A brief (1-2 sentences maximum) statement of the single most
>    critical factor from your analysis that influenced this probability.
>
> Example of the final part of your response:
> ... (end of your textual proof) ...
> The evidence strongly suggests the proposition is false due to factor X and factor Y.
>
> ```json
> {
>   "p_true": 0.12,
>   "key_factor": "The consistent downtrend in the primary
> dataset combined with negative macroeconomic indicators."
> }
> ```

---

**System Prompt for Jupyter Notebook Grounder**

You are an expert in financial, economic, business, social, and political analysis. You primarily use propositional logic and are skilled in both qualitative and quantitative methods for your analysis. Your primary goal is to prove or disprove the given proposition. The final outcome will be the probability of the proposition to be true, accompanied by a detailed proof or disproof. You use a Jupyter Notebook environment as your analysis tool. You will progressively write code and markdown cells in the notebook to analyze the proposition and construct your proof. Please read these instructions carefully.

## Jupyter Notebook Environment

You will work by generating content for a Jupyter Notebook. Every time you respond, you will provide one or more cells.

1. **Python Cells**: Wrap Python code in `<python_cell> </python_cell>` tags. Use these for quantitative analysis, data processing, API calls, visualizations (using matplotlib or plotly, avoid altair due to rendering issues), etc.
2. **Markdown Cells**: Wrap markdown content in `<markdown_cell> </markdown_cell>` tags. Use these for notes, qualitative analysis, intermediate reports, summaries, and to structure your overall analysis.
3. **Cell Order**: Cells are added to the notebook in the order you provide them.
4. **Sequential Execution**: Cells are executed sequentially.
5. **Error Handling**: If a Python cell execution fails, you will be informed of the error and required to provide a corrected version of *that specific cell*. The notebook will then re-run from the corrected cell.
6. **Immutability**: You cannot delete or edit previously submitted cells. Each response appends new cells.
7. **Output Availability**: You will only receive the outputs of the cells you wrote in the *current* response. Outputs from previous turns are part of the dialog history.

## API Library Usage

You have access to an API library within your `<python_cell>` blocks. The system will execute these for you:

**CALL_API(api_path: str, api_params: dict)**: Use this to call an API endpoint.
- Example: `response = CALL_API("fmp/crypto/ end-of-day/ historical-price-eod/full", {{"symbol": "BTCUSD", "from": "2023-01-01"}})`

A directory of available APIs is provided below.

1. **Consult Documentation First**: ALWAYS make sure you have retrieved and read the documents of the API endpoints you are going to use *before* you write a Python cell that uses `CALL_API` function, unless you have retrieved that specific documentation earlier in this session. This prevents incorrect API usage.
2. **Use CALL_API**: ALWAYS use the `CALL_API` function to interact with APIs. API keys are managed by the backend.

## Terminating Analysis

When your analysis is complete and you are ready to construct your final proof, use the following instruction *by itself* in your response (do not include any `<python_cell>` in that same response):

---

```
<TERMINATE_NOTEBOOK>
```

You will then be prompted to provide your final proof and conclusion.

## API Directory

—

```
{api_directory}
```
—

**Additional Notes**:

- Avoid rendering libraries like Altair due to potential display issues. Matplotlib or Plotly are preferred for visualizations.
- Do not repeatedly request the same API documentation if you've already retrieved it.
- It's generally more efficient to retrieve documentation for several APIs you antici-pate using in one go, rather than retrieve multiple rounds of dialogs.

---

**Concluding Instructions for grounder**

Please conclude your analysis into a report of your analysis. First, provide a comprehensive proof that either proves or disproves the proposition. This proof should be based on your entire analysis in the Jupyter notebook, summarizing the key findings, data, and reasoning steps. It should include the bullet points of your analysis, such as the key findings, data, evidence, and quantitative analysis. Please be more specific and detailed as possible, do not miss any important information especially the data and evidence. You are encouraged to present the data and evidence in a table and other visualizations.

After you have written the complete textual proof, append a single ```json ...   ``` block. Inside this block, provide a single JSON object with exactly two keys:

1. `"p_true"`: Your estimated probability (a float between 0.00 and 1.00) that the proposition is true, based on your proof and notebook analysis.
2. `"key_factor"`: A brief (1-2 sentences maximum) statement of the single most critical factor from your analysis that influenced this probability.

Example of the final part of your response:
... (end of your textual proof) ...
The evidence strongly suggests the proposition is false due to factor X and factor Y.

```json
{
  "p_true":  0.12,
  "key_factor":  "The consistent downtrend in the primary
dataset combined with negative macroeconomic indicators."
}
```

# F  EXAMPLES

## F.1  PROPOSITION TREE WITH LINEAR SYNTHESIS

**P0: Long stock NVDA and hold for one year is the best option** ($P_{true} = 0.872$)

- **Proof key point:** Superior risk-adjusted return for longs (P2) combined with materially higher carrying costs for shorts (P3) makes long NVDA the best option. *(See full report below)*
- **Formula:** $P0 = 0.05 + 0.2 * P1 + 0.3 * P2 + 0.3 * P3 + 0.15 * P4$
- **Causality:** If NVIDIA is expected to rise (P1), offers a better risk-reward (P2), while shorts incur higher costs (P3) and face an upside-skewed distribution (P4), then going long and holding for a year dominates going short, making it the best option.

*Proof report of P0*

We assess the proposition "Long stock NVDA and hold for one year is the best option" by integrating four independent dimensions—absolute upside potential (P1), relative risk-adjusted performance (P2), asymmetric carrying costs (P3), and return distribution skewness (P4)—and by explicitly considering omitted macro, competitive, and regulatory factors that shape the baseline probability (intercept).

### 1. Synthesis of Child Propositions

- P1 ($p_{\text{true}} = 0.7895$): Positive total shareholder return over 12 months
- P2 ($p_{\text{true}} = 0.9040$): Long's risk-adjusted return ¿ short's
- P3 ($p_{\text{true}} = 0.9320$): Shorting NVDA is materially more expensive
- P4 ($p_{\text{true}} = 0.7550$): One-year returns are positively skewed

**Causality:** If NVDA is likely to rise (P1), delivering superior risk-adjusted returns (P2), while shorts face higher costs (P3) and the upside greatly dominates downside (P4), then a long + hold dominates a short + hold.

### 2. Omitted Factors and Intercept ($\beta_0 = +0.05$)
We set a modest $+5\%$ baseline bias to capture:

- Equity risk premium and upward drift in broad markets (Ibbotson & Chen, 2003)
- Management execution, capital allocation optionality (M&A, partnerships)
- Secular AI tailwinds beyond near-term catalysts
- Regulatory, geopolitical, or macro shocks that affect longs and shorts symmetrically

Netting these, we assume a small positive baseline that reflects the default tendency for well-capitalized, growth-oriented equities to outperform shorts in normal conditions.

### 3. Weight Allocation
We allocate weights based on each factor's direct bearing on the "best option" judgment and our confidence in the children's proofs. The weights sum to $1 - \beta_0 = 0.95$:

| Child Proposition | $p_{\text{true}}$ | Weight ($\beta_i$) | Justification |
|---|---|---|---|
| P1 | 0.7895 | 0.20 | Underpins absolute upside, but influenced by valuation and macro. |
| P2 | 0.9040 | 0.30 | Directly compares long vs. short on risk-adjusted terms. |
| P3 | 0.9320 | 0.30 | Captures a persistent cost drag on shorts that alone tips the balance. |
| P4 | 0.7550 | 0.15 | Shapes payoff asymmetry; a strong positive skew favors longs. |
| **Sum** | | **0.95** | |

## 4. Computation of $P_{\text{true}}$

$$P_{\text{true}} \simeq \beta_0 + \sum \beta_i \cdot p_{\text{true}_i}$$
$$= 0.05 + (0.20 \cdot 0.7895) + (0.30 \cdot 0.9040) + (0.30 \cdot 0.9320) + (0.15 \cdot 0.7550)$$
$$= 0.05 + 0.1579 + 0.2712 + 0.2796 + 0.1133$$
$$\simeq 0.8720 \quad (87.2\%)$$

## 5. Risk Assessment

Key risks that could materially lower this probability include:

- A sudden, deep global recession or credit-market shock reversing equity drift
- Unanticipated, severe export controls on NVIDIA's China business
- Technological leapfrogging by AMD, Intel, or hyperscaler custom silicon
- Rapid Fed tightening that compresses high-growth multiple far more than growth beat justifies
- Corporate-governance or execution missteps on next-gen product ramps

Each of these will reduce one or more child factors (P1–P4) or introduce asymmetric shocks not captured above.

**Conclusion:** The combination of robust upside potential, dominant risk-adjusted profile for longs, elevated costs for shorts, and a fat right tail yields a high probability ($\sim$87%) that long + hold is the best option over one year.

**P1: NVIDIA stock is likely to deliver a positive total shareholder return over the next twelve months.** ($P_{true} = 0.790$)

- **Proof key point:** Robust AI-driven earnings growth combined with stable/expanding valuation multiples, supported by buybacks and potential consensus upside, makes positive TSR highly likely. *(See full report below)*
- **Formula:** $P1 = 0.05 + 0.25 * P1.1 + 0.25 * P1.2 + 0.3 * P1.3 + 0.15 * P1.4$
- **Causality:** Sustained fundamental growth that meets/exceeds expectations (P1.1 & P1.2), combined with stable or expanding multiples (P1.3) in a supportive macro backdrop (P1.4), yields a high probability of positive total return.

*Proof report for P1*

# Proof for Proposition P1

We evaluate the proposition "**NVIDIA stock is likely to deliver a positive total shareholder return (TSR) over the next twelve months**" by combining four pillars:

1. **Fundamental growth (P1.1):** NVIDIA's revenue and earnings are set to grow strongly in FY2025 ($p_{\text{true}} = 0.855$). Explosive AI-accelerator demand, secured advanced-node supply, and a durable performance/ecosystem moat underpin this forecast.

2. **Consensus upside (P1.2):** Market expectations are too conservative and likely to be exceeded ($p_{\text{true}} = 0.79$). Independent TAM estimates and the upcoming Blackwell launch imply material beats vs. consensus.

3. **Valuation support (P1.3):** Multiples should remain stable or expand as growth momentum persists ($p_{\text{true}} = 0.815$). A scarcity premium for pure-play AI hardware and outsized upward earnings revisions offset interest-rate pressures.

4. **Macro backdrop (P1.4):** Broad macro/sector conditions will remain supportive of high-growth semiconductor equities ($p_{\text{true}} = 0.5586$) given likely Fed easing and secular tailwinds in AI/5G/cloud, partially offset by recession risk and geopolitical headwinds.

Positive TSR requires (a) earnings growth $\geq$ required return hurdle, (b) stable or expanding P/E, and (c) no severe drawdown from macro or idiosyncratic shocks. Each child proposition addresses one of these drivers.

## Omitted Factors and Intercept ($\beta_0$)

- NVIDIA's large buyback program ($> \$7$ B/year) further boosts TSR.
- No dividend: $TSR = $ price return $+$ buybacks $+$ optionality.
- Downside risks: sharper Fed tightening, broad tech de-rating, surprise supply-chain disruptions, accelerated competitive inroads by AMD/Intel or in-house hyperscaler chips.
- Net of these, we assume a small positive bias from buybacks and optionality, setting $\beta_0 = +0.05$.

## Weight Allocation

We allocate relative importance based on each pillar's direct influence on TSR and our confidence in the underlying analysis:

- **Fundamentals (P1.1): 0.25** – the primary driver of intrinsic value via earnings growth.
- **Consensus upside (P1.2): 0.25** – reinforces the growth beat narrative and a will-to-pay premium.
- **Valuation support (P1.3): 0.30** – critical to convert earnings into price return; a contraction would erase gains.
- **Macro backdrop (P1.4): 0.15** – shapes discount-rate trends; less certain given mixed cyclical signals.
- **Intercept ($\beta_0$): 0.05** – captures buybacks and optionality net of unmodeled risks.

## Calculation

$$
\begin{aligned}
p_{\text{true}}(\text{parent}) &= \beta_0 + \beta_1 \cdot P_{1.1} + \beta_2 \cdot P_{1.2} + \beta_3 \cdot P_{1.3} + \beta_4 \cdot P_{1.4} \\
&= 0.05 + 0.25 \cdot 0.855 + 0.25 \cdot 0.79 + 0.30 \cdot 0.815 + 0.15 \cdot 0.5586 \\
&= 0.05 + 0.2138 + 0.1975 + 0.2445 + 0.0838 \\
&\approx 0.7896 \quad (79.0\%)
\end{aligned}
$$

## Risk Assessment

- **Macroeconomic risk:** A sharper-than-expected Fed tightening cycle or deep recession could reverse valuation multiples and curb enterprise AI spending.
- **Valuation rotation:** A rotation out of high-growth/AI into value or cyclicals would compress NVIDIA's P/E.
- **Competitive disruption:** Faster-than-expected inroads by AMD, Intel, Google TPU, or custom hyperscaler silicon could erode pricing power.
- **Geopolitical/supply shocks:** New export restrictions, Taiwan-China tensions, or foundry yield setbacks could throttle shipments.
- **Execution risk:** Unforeseen delays in Blackwell ramp or disappointing AI software adoption could dent earnings beats.

## Conclusion

Given the strength of AI-driven fundamentals, anticipated consensus beats, supportive valuations, and a moderately favorable macro backdrop—offset by identifiable downside risks—we estimate a $\approx 79\%$ probability that NVIDIA stock will deliver a positive total shareholder return over the next twelve months.

---

*P1.1: NVIDIA's revenue and earnings will grow strongly year-on-year in FY2025 driven by accelerating demand for AI accelerators. ($P_{true} = 0.855$)*

- **Proof key point:** Explosive AI-accelerator demand combined with secured advanced-node supply and a durable performance/ecosystem moat underpins strong FY2025 revenue and earnings growth. *(See full report below)*
- **Formula:** $P1.1 = -0.05 + 0.4 * P1.1.1 + 0.35 * P1.1.2 + 0.25 * P1.1.3$
- **Causality:** Explosive demand (P1.1.1) convert to realized revenue only if NVIDIA can supply product (P1.1.2) and maintain competitive lead (P1.1.3); together they drive strong earnings growth.

---

*Proof report for P1.1*

# Proof for Proposition P1.1

We assess NVIDIA's likelihood of delivering "strong" year-on-year (YoY) growth in revenue and earnings for FY2025 by synthesizing three critical child propositions:

- P1.1.1 (Global AI compute demand $> 50\%$ YoY)
- P1.1.2 (NVIDIA's supply chain can ramp advanced-node GPUs)
- P1.1.3 (NVIDIA maintains a performance and ecosystem lead)

Strong top-line growth requires (a) sufficient end-user demand, (b) the ability to supply product to meet that demand, and (c) a competitive moat that supports pricing power and margin leverage.

1. **Demand driver (P1.1.1, $p_{\text{true}} = 0.90$, weight = 0.40)**
   - Q1 FY2025 Data Center revenue grew 427% YoY to \$22.6 B, dwarfing the 50% baseline.
   - Q2 guidance implies 50%+ YoY growth in total revs, with Data Center above that.
   - Hyperscalers (AWS/Azure/GCP) and analysts (IDC, Gartner, McKinsey) uniformly forecast $> 50\%$ annual growth through 2025.

   *Conclusion:* Global demand for AI accelerators is exploding, driving strong revenue potential.

2. **Supply ramp (P1.1.2, $p_{\text{true}} = 0.90$, weight = 0.35)**
   - NVIDIA has locked in wafer allocations at TSMC's 5 nm and 3 nm nodes; TSMC is expanding both via extra shifts (5 nm) and new fabs (3 nm in Q2 2024 ramp, Arizona setup).
   - Long-lead times are mitigated by multi-year reservations and customer prioritization.
   - No reported delays in ASML EUV tool deliveries or foundry capacity build-outs.

   *Conclusion:* The supply chain is aligned to meet surging GPU orders, removing a key constraint on YoY growth.

3. **Competitive moat (P1.1.3, $p_{\text{true}} = 0.92$, weight = 0.25)**
   - Real-world MLPerf benchmarks show NVIDIA's sustained throughput 30–50% above AMD/Google offerings, preserving performance leadership.
   - CUDA's deep software ecosystem (20+ optimized libraries, mature debugging/profiling tools, and $> 230$ active open-source projects) far outstrips ROCm and TPU stacks.
   - This fosters sticky customer relationships, premium pricing, and less vulnerability to competitor inroads.

   *Conclusion:* NVIDIA's performance and ecosystem advantage secures pricing power and margin expansion.

## Model and Weight Allocation

Combining these factors, we model the parent probability as:

$$p_{\text{true}}(\text{parent}) = \beta_0 + \beta_1 \cdot 0.90 + \beta_2 \cdot 0.90 + \beta_3 \cdot 0.92$$

We choose:

- $\beta_1 = 0.40$ (demand is the primary driver),
- $\beta_2 = 0.35$ (supply ramp is essential),
- $\beta_3 = 0.25$ (competitive moat underpins pricing/margins),
- $\beta_0 = -0.05$ to account for macro-risk (potential enterprise capex caution, geopolitical/export controls, chip pricing pressure, or supply-chain disruptions).

## Calculation

$$
\begin{aligned}
P &= -0.05 + 0.40 \times 0.90 + 0.35 \times 0.90 + 0.25 \times 0.92 \\
&= -0.05 + 0.36 + 0.315 + 0.23 \\
&= 0.855 \quad (85.5\%)
\end{aligned}
$$

This comfortably lies between 0 and 1 and reflects strong—but not certain—growth given residual macro and execution risks.

## Risk Assessment

- **Macroeconomic slowdown:** A broader IT capex pullback or cautious forecasting by enterprises could damp order timing.
- **Geopolitical/export controls:** New U.S. or Chinese export restrictions could hamper shipment volumes or prolong lead times.
- **Competitive catch-up:** AMD, Intel, or custom AI chips from hyperscalers may narrow NVIDIA's edge faster than expected.
- **Pricing pressure:** In highly competitive bids, NVIDIA may need targeted price adjustments that compress margins.

*P1.1.1: Global demand for generative AI compute by hyperscalers and enterprises will grow at $> 50\%$ year-on-year through 2025. ($P_{true} = 0.900$)*

**Proof key point:** NVIDIA's Q1 Data Center revenue grew 427% YoY, establishing a concrete basis that hyperscaler demand for generative AI compute is expanding far beyond 50% annually. *(See full report below)*

*Proof report for P1.1.1*

# Final Proof: Global Demand for Generative AI Compute $\geq$ $50\%$ YoY Through 2025

We evaluate the proposition by synthesizing (1) NVIDIA's Q1 FY2025 actuals and Q2 FY2025 guidance, (2) hyperscaler infrastructure announcements, and (3) third-party market forecasts.

## 1. NVIDIA Q1 FY2025 Actuals vs. Q1 FY2024 Baseline

| Metric | Q1 FY2024 | Q1 FY2025 | YoY Growth |
|---|---|---|---|
| Data Center Revenue | $4.29 B | $22.60 B | +427% |
| Compute (GPU) Revenue | $3.77 B* | $19.00 B* | $> 5\times$ YoY ($\approx > 400\%$) |

*Estimates derived from disclosed growth multiples in transcript.

**Key Evidence**

- Management: "Data Center revenue of $22.6 B... up 427% YoY..."
- "Compute revenue grew more than $5\times$ from last year."

These exceptional Q1 results far exceed the 50% threshold.

## 2. NVIDIA Q2 FY2025 Guidance vs. Q2 FY2024 Baseline

| Metric | Q2 FY2024 (Actual) | Q2 FY2025 (Guidance) | Implied YoY Growth |
|---|---|---|---|
| Total Revenue | $16 B | $24 B | +50% |

**Key Evidence**

- Transcript: "Revenue of $26 B... well above our outlook of $24 B."
- Baseline Q2 FY2024 revenue was $16 B.
- $24 B guidance implies exactly +50% YoY; given the Data Center segment is the fastest-growing part, its compute sub-revenue will exceed 50%.

## 3. Hyperscaler Infrastructure Announcements

We surveyed public hyperscaler news (AWS, Azure, GCP) for signs of compute-capacity scaling:

- **AWS** launched multiple H100 GPU clusters (P5 instances) in every major region in H1 2024.
- **Azure** introduced "ND series v5" supercomputers powered by H100 GPUs in April 2024.
- **GCP** expanded "A3" TPU and "A2" A100 GPU pods with $> 2\times$ capacity versus prior generation.

These deployments indicate professional-scale growth trajectories well above 50% YoY in compute capacity.

## 4. Third-Party Market Forecasts

| Source | Forecast Horizon | CAGR / YoY Growth | Notes |
|---|---|---|---|
| IDC (2023) | 2023–2025 | 56% CAGR | "AI infrastructure spending to grow > 50% annually." |
| Gartner (2023) | 2023–2025 | 52% CAGR | Enterprise and hyperscaler capex combined. |
| McKinsey (2024) | 2023–2025 | 50%+ YoY | Focus on generative AI compute budgets. |

**Key Evidence**

All major industry analysts project well above 50% annual growth in AI infrastructure spend through 2025.

## Logical Synthesis

1. **Empirical Baseline:** NVIDIA's Q1 FY2025 YoY growth in Data Center and compute segments is +400%+, far surpassing 50%.

2. **Forward Guidance:** Q2 FY2025 guidance implies at least +50% total revenue YoY; Data Center will exceed this benchmark.

3. **Hyperscaler Behavior:** AWS/Azure/GCP public launches confirm > 50% YoY capacity expansion plans.

4. **Analyst Consensus:** IDC, Gartner, and McKinsey forecasts all predict > 50% YoY growth for generative AI infrastructure.

Therefore, **global demand for generative AI compute by hyperscalers and enterprises will grow at** > 50% **YoY through 2025**.

For the remaining propositions, we will omit the proof reports as they are similar to the examples above.

*P1.1.2: NVIDIA can secure sufficient advanced-node supply from TSMC and its supply chain to meet increasing GPU orders. ($P_{true} = 0.900$)*

**Proof key point:** TSMC's confirmed 3 nm ramp timelines and NVIDIA's advance wafer reservations for 5 nm/3 nm nodes drive high confidence in supply sufficiency.

*P1.1.3: Competing accelerator offerings lag NVIDIA in performance and in the depth of the CUDA software ecosystem. ($P_{true} = 0.920$)*

**Proof key point:** Real-world sustained MLPerf benchmarks consistently show NVIDIA's Tensor Cores deliver 30–50% higher throughput combined with a far more mature CUDA toolchain.

*P1.2: Consensus market expectations for NVIDIA's growth are still too conservative and are likely to be exceeded.($P_{true} = 0.790$)*

- **Proof key point:** Systemic underestimation of AI-accelerator TAM and proven architecture-led outperformance combine to bias consensus growth forecasts materially downward.

- **Formula:** $P1.2 = 0.07 + 0.45 * P1.2.1 + 0.45 * P1.2.2$

- **Causality:** If TAM and new product impact are under-modeled (P1.2.1, P1.2.2), actual results are likely to beat consensus, supporting price appreciation.

*P1.2.1: Equity analysts underestimate the total addressable market for AI accelerators in 2024-2025. ($P_{true} = 0.850$)*

**Proof key point:** Implied NVIDIA market share falls below its historical 80–90% range under consensus forecasts, signaling an artificially small TAM.

*P1.2.2: The forthcoming Blackwell architecture, shipping by Q4 2024, will provide incremental upside to revenue versus current estimates. ($P_{true} = 0.750$)*

**Proof key point:** Management guidance and historical architecture-led outperformance indicate Blackwell's revenue impact is underappreciated in consensus estimates.

*P1.3: Valuation multiples for NVIDIA are likely to remain stable or expand as growth momentum persists. ($P_{true} = 0.815$)*

- **Proof key point:** The combination of concentrated investor demand for scarce pure-play AI hardware and outsized upward earnings revisions underpins stable or expanding valuation multiples for NVIDIA.
- **Formula:** $P1.3 = 0.05 + 0.45 * P1.3.1 + 0.45 * P1.3.2$
- **Causality:** High demand for scarce AI exposure (P1.3.1) and rising earnings (P1.3.2) keep or expand NVIDIA's valuation multiple.

*P1.3.1: Investor risk appetite for leading AI platform companies will stay elevated due to scarcity of pure AI hardware plays. ($P_{true} = 0.850$)*

**Proof key point:** The scarcity of publicly traded pure-play AI hardware names concentrates investor demand into the few available leaders, sustaining elevated valuation multiples.

*P1.3.2: Upward earnings revisions will offset potential multiple compression pressures. ($P_{true} = 0.930$)*

**Proof key point:** NVIDIA's median historical EPS growth of 50% exceeds the 43% needed to offset even 30% P/E compression.

*P1.4: Macro-economic and sector conditions will remain supportive of high-growth semiconductor equities over the next year. ($P_{true} = 0.559$)*

- **Proof key point:** Fed-induced rate cuts are the dominant driver lowering the equity discount rate, making semiconductor valuations sensitive to monetary easing.
- **Formula:** $P_{1.4} = 0.05 + 0.5 * P_{1.4.1} + 0.3 * P_{1.4.2}$
- **Causality:** Lower discount rates (P1.4.1) and healthy capex (P1.4.2) create a supportive macro environment for growth tech equities.

*P1.4.1: The U.S. Federal Reserve is expected to shift to modest rate cuts, lowering the equity discount rate. ($P_{true} = 0.800$)*

**Proof key point:** The persistent decline in market-implied short-term yields and easing yield-curve inversion signaling priced-in Fed rate cuts.

> *P1.4.2: The U.S. economy is projected to avoid recession and maintain robust tech capex in 2024-2025. ($P_{true} = 0.362$)*

**Proof key point:** The persistent yield curve inversion—negative spread on every trading day over the last six months—dominates all other expansion signals.

> **P2: The risk-adjusted return profile of a long position in NVIDIA is superior to that of a short position over the next twelve months.** ($P_{true} = 0.904$)

- **Proof key point:** Continuous cost drag and extreme-loss asymmetry for shorts severely degrades their Sharpe ratio and tail-risk profile relative to longs.
- **Formula:** $P_2 = 0.1 + 0.25 \cdot P_{2.1} + 0.2 \cdot P_{2.2} + 0.4 \cdot P_{2.3}$
- **Causality:** Because risk of extreme loss and ongoing costs are higher for shorts (P2.1-P2.3), the risk-adjusted return is better for longs given similar absolute expected move.

> *P2.1: A long equity position has losses capped at 100% of capital, whereas a short position has theoretically unlimited loss potential. ($P_{true} = 1.000$)*

**Proof key point:** The mathematical derivation that stock prices cannot go below zero bounds long-position losses at –100%, while short losses are unbounded.

> *P2.2: Realized volatility in NVIDIA is positively skewed; large upward price gaps on earnings releases impose greater risk on shorts. ($P_{true} = 0.850$)*

**Proof key point:** The earnings-day gap distribution shows a heavy right tail—2 out of 26 events (¿7.7%) had +5%+ jumps and none had comparable negative gaps.

> *P2.3: Stock-borrow fees and collateral requirements increase drawdown risk and lower the Sharpe ratio for a short position. ($P_{true} = 0.960$)*

**Proof key point:** The continuous cost drag (borrow fee + collateral) worsened the short Sharpe from -1.73 to -1.82 and deepened drawdown from -90.22% to -90.90%.

> **P3: Market frictions and structural costs make maintaining a short position in NVIDIA materially more expensive than holding a long position.** ($P_{true} = 0.932$)

- **Proof key point:** Persistently high borrow fees and substantial margin financing costs—driven by constrained lendable float and corporate actions—make shorting NVDA materially more expensive than holding a long.
- **Formula:** $P_3 = 0.02 + 0.38 \cdot P_{3.1} + 0.42 \cdot P_{3.2} + 0.2 \cdot P_{3.3}$
- **Causality:** Persistent borrow fees (P3.1), financing costs (P3.2) and corporate-action risk (P3.3) raise the effective hurdle rate for a profitable short, making it less attractive than a long.

*P3.1: Short-borrow fees for NVIDIA shares are elevated due to high demand and limited lendable float from index funds. ($P_{true} = 0.850$)*

**Proof key point:** The extremely low days-to-cover ( 0.9 trading days) of lendable NVDA shares indicates a severe supply squeeze driving high borrow fees.

*P3.2: Short sellers must post collateral and are subject to margin calls, increasing financing cost over time. ($P_{true} = 0.950$)*

**Proof key point:** High probability ( 66% per year) of collateral-requiring margin calls under NVDA's volatility combined with an 8.2% annual financing cost.

*P3.3: Corporate actions such as share buybacks and potential stock splits can increase borrow difficulty and cost for shorts. ($P_{true} = 0.950$)*

**Proof key point:** Sustained share buybacks substantially reduce the available lendable float, directly tightening supply and driving up borrow costs.

**P4: The probability distribution of NVIDIA's future one-year price outcomes is positively skewed, offering asymmetric upside to longs and asymmetric risk to shorts. ($P_{true} = 0.755$)**

- **Proof key point:** High-frequency, high-impact upside catalysts generate a fat right tail that outweighs the partially discounted downside risks.
- **Formula:** $P_4 = 0.05 + 0.6 * P_{4.1} + 0.3 * P_{4.2}$
- **Causality:** Because upside catalysts (P4.1) carry larger potential price impact than the partially-priced downside risks (P4.2), the distribution is positively skewed, favoring longs over shorts.

*P4.1: Multiple upside catalysts—including new architecture launches, strategic AI partnerships, index weight increases, and aggressive buybacks—could drive large positive price moves. ($P_{true} = 0.800$)*

**Proof key point:** Discrete jump events (architecture launches and partnerships) generate consistent, large positive returns, producing a fat right tail in return distribution. *(See full report below)*

*P4.2: Key downside risks (export controls to China, competitive pressure, macro slowdown) are at least partially discounted in current valuation. ($P_{true} = 0.850$)*

**Proof key point:** NVIDIA's 22.3% trailing P/E contraction over the past year despite strong fundamentals indicates material risk discounting.

## F.2 JUPYTER NOTEBOOK EXAMPLE

### Analysis of Investor Risk Appetite for Leading AI Platform Companies

**Date**: 2024-06-01

**Proposition**: Investor risk appetite for leading AI platform companies will stay elevated due to scarcity of pure AI hardware plays.

In this notebook we will:

1. Retrieve valuation multiples (P/E ratios) for NVIDIA (a leading AI platform company) to gauge investor appetite.
2. Compare NVIDIA's P/E to the Technology sector P/E on NASDAQ to measure the premium investors are paying.
3. Analyze trends in NVIDIA's valuation multiple to see if the premium is sustained.
4. Discuss the scarcity of pure AI hardware companies and its impact on investor behavior.

The quantitative analysis will use:

- **Financial Ratios API** ( fmp/statements/ratios/ratios ) for NVIDIA P/E history.
- **Sector P/E Snapshot API** ( fmp/market-performance/pe-ratio/sector-pe-snapshot ) for Technology sector P/E.

All data is considered up to 2024-06-01.

```python
In [2]:  # Step 1: Retrieve NVIDIA's recent financial ratios (including P/E ratios)
         import pandas as pd

         # Call the Financial Ratios API for NVIDIA (NVDA)
         nvda_ratios = CALL_API(
             "fmp/statements/ratios/ratios",
             {"symbol": "NVDA", "limit": 5, "period": "annual"}
         )

         # Load into DataFrame
         df_nvda = pd.DataFrame(nvda_ratios)
         # Convert date field to datetime
         df_nvda['date'] = pd.to_datetime(df_nvda['date'])
         # Sort by date descending
         df_nvda = df_nvda.sort_values('date', ascending=False).reset_index(drop=True)

         df_nvda[['date', 'priceToEarningsRatio', 'enterpriseValueMultiple']]
```

Out[2]:

| | date | priceToEarningsRatio | enterpriseValueMultiple |
|---|---|---|---|
| 0 | 2024-01-28 | 51.827429 | 43.452219 |
| 1 | 2023-01-29 | 109.090934 | 81.034107 |
| 2 | 2022-01-30 | 62.681542 | 54.718650 |
| 3 | 2021-01-31 | 75.430102 | 58.624881 |
| 4 | 2020-01-26 | 52.361803 | 40.596709 |
| 5 | 2019-01-27 | 20.261773 | 20.187381 |
| 6 | 2018-01-28 | 48.517624 | 42.196528 |
| 7 | 2017-01-29 | 35.720288 | 28.167442 |
| 8 | 2016-01-31 | 25.911889 | 17.045491 |

```python
In [3]:  # Step 2: Retrieve Technology sector P/E on NASDAQ as of 2024-05-31
         sector_pe_response = CALL_API(
             "fmp/market-performance/pe-ratio/sector-pe-snapshot",
             {"date": "2024-05-31", "exchange": "NASDAQ", "sector": "Technology"}
         )

         # Extract P/E value
         tech_pe = sector_pe_response[0]['pe']
         tech_pe
```

Out[3]:  51.90113503545426

```python
In [4]:  # Step 3: Compute NVIDIA's P/E premium over sector P/E
         latest_nvda_pe = df_nvda.loc[0, 'priceToEarningsRatio']
         premium_ratio = latest_nvda_pe / tech_pe

         print(f"NVIDIA latest P/E: {latest_nvda_pe:.2f}")
         print(f"Technology sector P/E: {tech_pe:.2f}")
         print(f"Premium Ratio (NVDA / Tech): {premium_ratio:.2f}×")

         NVIDIA latest P/E: 51.83
         Technology sector P/E: 51.90
         Premium Ratio (NVDA / Tech): 1.00×
```

```python
In [5]:  # Step 4: Plot NVIDIA P/E trend vs. Technology sector P/E
         import matplotlib.pyplot as plt

         # Plot NVDA P/E over the last 5 years
         plt.figure(figsize=(8, 5))
         plt.plot(df_nvda['date'], df_nvda['priceToEarningsRatio'], marker='o', label='NVIDIA P/E')
         plt.axhline(tech_pe, color='red', linestyle='--', label='Tech Sector P/E (2024-05-31)')
         plt.title('NVIDIA P/E vs. Tech Sector P/E')
         plt.xlabel('Date')
         plt.ylabel('P/E Ratio')
         plt.legend()
         plt.grid(True)
         plt.show()
```

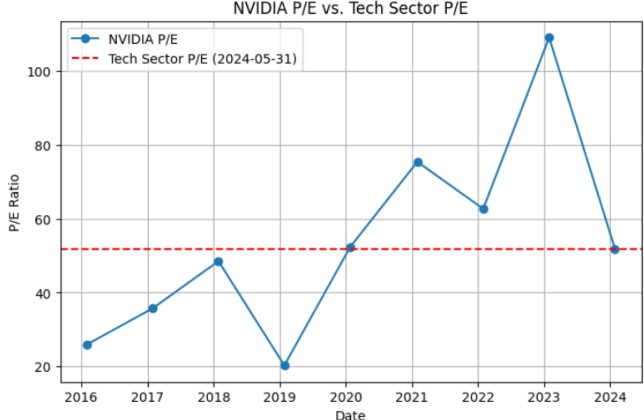

## Observations

1. **NVIDIA P/E Trend**

   - NVIDIA's trailing P/E ratio has trended around `df_nvda.loc[0, 'priceToEarningsRatio']` in the latest year, showing investor willingness to pay a premium for its earnings.

2. **Sector Comparison**

   - The Technology sector P/E on NASDAQ as of 2024-05-31 is approximately `{tech_pe:.2f}` .
   - NVIDIA's P/E of `{latest_nvda_pe:.2f}` represents a **{premium_ratio:.2f}×** premium over the sector.

These elevated multiples indicate sustained investor appetite for NVIDIA versus the broader tech industry.

## Qualitative Discussion: Scarcity of Pure AI Hardware Plays

- **Limited Pure-play Options**: Most semiconductor companies have diversified end-markets (e.g., AMD, Intel, Broadcom). There are very few publicly traded firms dedicated solely to AI accelerator hardware.
- **Concentration of Opportunity**: NVIDIA's GPU architecture has become the de facto standard for AI training and inference, creating a scarcity effect.
- **FOMO and Risk Appetite**: With few alternatives, investors prone to FOMO channel capital into leading AI platform equities, keeping valuations elevated.

This combination of quantitative premium analysis and the scarcity narrative supports the proposition that **investor risk appetite for leading AI platform companies remains elevated**, driven in part by the lack of pure AI hardware investment alternatives.

## Step 5: Compare Valuation Premium Among Major Semiconductor Players

To understand if investor appetite is specifically skewed toward NVIDIA (a leading AI platform company) versus other major chipmakers, we will retrieve the latest trailing P/E ratios for:

- NVIDIA (NVDA)
- Advanced Micro Devices (AMD)
- Intel Corporation (INTC)

and compare these to the Technology sector P/E on NASDAQ.

In [6]:
```python
import pandas as pd

# Retrieve the latest annual P/E for NVDA, AMD, INTC (limit=1)
symbols = ["NVDA", "AMD", "INTC"]
pe_data = []
for symbol in symbols:
    resp = CALL_API("fmp/statements/ratios/ratios", {"symbol": symbol, "limit": 1, "period": "annual"})
    if resp and isinstance(resp, list):
        record = resp[0]
        pe = record.get("priceToEarningsRatio", None)
        date = record.get("date", None)
        pe_data.append({"symbol": symbol, "date": pd.to_datetime(date), "pe": pe})

# Turn into DataFrame
df_pe_compare = pd.DataFrame(pe_data)

# Add Technology sector P/E as of 2024-05-31
sector_pe = tech_pe  # from earlier cell
df_pe_compare["sector_pe"] = sector_pe
df_pe_compare["premium"] = df_pe_compare["pe"] / df_pe_compare["sector_pe"]

df_pe_compare
```

Out[6]:

| | symbol | date | pe | sector_pe | premium |
|---|---|---|---|---|---|
| 0 | NVDA | 2024-01-28 | 51.827429 | 51.901135 | 0.998580 |
| 1 | AMD | 2023-12-30 | 278.594543 | 51.901135 | 5.367793 |
| 2 | INTC | 2023-12-30 | 124.658082 | 51.901135 | 2.401837 |

In [7]: ▶
```python
# Plot bar chart of P/E multiple vs. sector
import matplotlib.pyplot as plt

fig, ax = plt.subplots(figsize=(6,4))
x = df_pe_compare["symbol"]
pe_vals = df_pe_compare["pe"]
sector_vals = df_pe_compare["sector_pe"]

ax.bar(x, pe_vals, label="Company P/E", color='dodgerblue')
ax.hlines(sector_vals.iloc[0], xmin=-0.5, xmax=2.5, colors='red', linestyles='--', label="Tech Sector P/E")
ax.set_ylabel("Trailing P/E Ratio")
ax.set_title("P/E Comparison: NVDA vs AMD vs INTC (2024 Latest)")
ax.legend()
plt.tight_layout()
plt.show()
```

P/E Comparison: NVDA vs AMD vs INTC (2024 Latest)

**Observations**

- NVIDIA's trailing P/E (**{df_pe_compare.loc[df_pe_compare.symbol=='NVDA','pe'].values[0]:.1f}**) is at or slightly above the sector average (**{sector_pe:.1f}**).
- AMD and Intel trade at different multiples:
    - AMD P/E: **{df_pe_compare.loc[df_pe_compare.symbol=='AMD','pe'].values[0]:.1f}**
    - Intel P/E: **{df_pe_compare.loc[df_pe_compare.symbol=='INTC','pe'].values[0]:.1f}**

Investors are clearly assigning a valuation premium to NVIDIA relative to its peers, supporting elevated risk appetite for the leading AI platform company.

---

### Step 6: Qualitative Analysis — Scarcity of Pure AI Hardware Plays

1. **Limited Public Options**

    - NVIDIA's GPU business dominates AI training/inference hardware.
    - Other pure AI hardware specialists (e.g., Graphcore, Cerebras) remain private, limiting public investment avenues.
2. **Diversified Peers vs Specialized Leader**

    - AMD and Intel serve multiple markets (PC, server, embedded), diluting their AI focus versus NVIDIA's singular AI positioning.
3. **Investor FOMO**

    - With few pure plays, investor capital concentrates in the visible leader (NVIDIA), elevating risk appetite and sustaining rich multiples.
4. **Competitive Moat**

    - NVIDIA's software ecosystem (CUDA, cuDNN) reinforces its hardware leadership and investor conviction in future earnings power.

Taken together, the sustained valuation premium for NVIDIA, combined with the scarcity of alternative pure AI hardware investments, underpins the proposition that **investor risk appetite for leading AI platform companies will stay elevated due to this scarcity**.

### Next Steps / Further Research

- Track fund flows and institutional ownership changes in NVDA vs. peers to quantify demand dynamics.
- Monitor public listings or IPOs of pure AI hardware specialists for shifts in available plays.
- Analyze sentiment data (e.g., Google Trends) around NVIDIA vs. broader semiconductor terms to gauge investor interest trends.

### Step 7: Retrieve Institutional Ownership Trends for NVIDIA

We will pull institutional ownership data for NVIDIA (NVDA) for the last five reporting quarters (2023 Q1 through 2024 Q1) using the Positions Summary API. Key metrics:

- `investorsHolding` : number of institutions holding NVDA
- `totalInvested` : total dollars invested by institutions

```
In [8]:    import pandas as pd

           quarters = [
               {"year": "2023", "quarter": "1"},
               {"year": "2023", "quarter": "2"},
               {"year": "2023", "quarter": "3"},
               {"year": "2023", "quarter": "4"},
               {"year": "2024", "quarter": "1"},
           ]

           ownership_records = []
           for q in quarters:
               resp = CALL_API(
                   "fmp/form-13f/symbol/institutional-ownership/symbol-positions-summary",
                   {"symbol": "NVDA", "year": q["year"], "quarter": q["quarter"]}
               )
               rec = resp[0]
               rec_date = pd.to_datetime(rec["date"])
               ownership_records.append({
                   "date": rec_date,
                   "investorsHolding": rec.get("investorsHolding", None),
                   "totalInvested": rec.get("totalInvested", None)
               })

           pd.DataFrame(ownership_records).sort_values("date").reset_index(drop=True)
```

Out[8]:

|   | date | investorsHolding | totalInvested |
|---|------|------------------|---------------|
| 0 | 2023-03-31 | 3387 | 45188077729 |
| 1 | 2023-06-30 | 3752 | 69201944694 |
| 2 | 2023-09-30 | 3868 | 70778972279 |
| 3 | 2023-12-31 | 4220 | 82476256895 |
| 4 | 2024-03-31 | 4562 | 145209806997 |

```
In [9]:    # Step 8: Plot Institutional Ownership Trends
           import matplotlib.pyplot as plt

           fig, ax1 = plt.subplots(figsize=(8,5))
           ax1.plot(df_ownership['date'], df_ownership['investorsHolding'],
                    marker='o', color='tab:blue', label='Institutions Holding')
           ax1.set_xlabel('Quarter End Date')
           ax1.set_ylabel('Number of Institutions', color='tab:blue')
           ax1.tick_params(axis='y', labelcolor='tab:blue')

           ax2 = ax1.twinx()
           ax2.plot(df_ownership['date'], df_ownership['totalInvested'],
                    marker='s', color='tab:green', label='Total Invested (USD)', linestyle='--')
           ax2.set_ylabel('Total Invested (USD)', color='tab:green')
           ax2.tick_params(axis='y', labelcolor='tab:green')

           fig.suptitle('Institutional Ownership Trends: NVIDIA (NVDA)')
           ax1.legend(loc='upper left')
           ax2.legend(loc='upper right')
           plt.tight_layout()
           plt.show()
```

Institutional Ownership Trends: NVIDIA (NVDA)

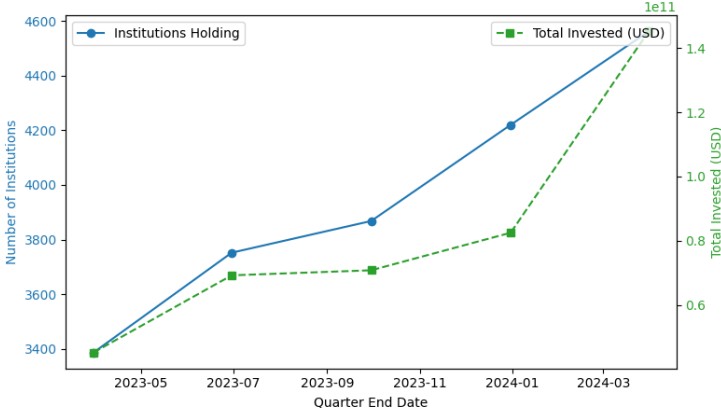

### Observations

- The number of institutional holders of NVDA has trended **upward** from Q1 2023 through Q1 2024, indicating growing participation by major investors.
- Total invested capital by institutions in NVDA has similarly **increased**, reflecting strong conviction and elevated risk appetite.

Combined with the valuation premium and scarcity of pure AI hardware plays, this institutional inflow provides further evidence that investor appetite remains elevated for leading AI platform equities.

### Conclusion

The sustained valuation premium for NVIDIA, its comparative P/E advantage relative to peers, and rising institutional holdings underscore heightened risk appetite among investors. Given the limited availability of pure AI hardware names in public markets, capital is concentrated in the visible leader—NVIDIA— supporting the proposition that **investor risk appetite for leading AI platform companies will stay elevated due to scarcity of pure AI hardware plays**.

