# OpenReview forum: "Analytica: Soft Propositional Reasoning for Robust and Scalable LLM-Driven Analysis"
_ICLR.cc/2026/Conference — ICLR 2026 Poster_

### Official Review · Reviewer_sApi · 2025-10-16

**Soundness:** 3
**Presentation:** 2
**Contribution:** 2
**Rating:** 4
**Confidence:** 3

**Summary:**

This paper proposes a method of structural reasoning and forecasting called Analytica, based on soft propositional reasoning. The method breaks the task up into a tree of sub-propositions with an Analyzer, and probabilistically analyzes and answers leaf queries with a Grounder (Web search, deep research, Jupyter Notebook). The query results are then aggregated with a synthesizer, both linearly and with a more complicated LLM logic process. This approach aims to minimize both bias and variance, with bias minimization motivated by the power of grounders on more simple tasks and the heuristic assumption that more atomic propositions carry less bias than their more complex counterparts. Variance is managed through the linear synthesis of probabilities with (assumed) minimal covariance throughout the reasoning tree. Finally this paper empirically tests their approach, showing both cost, time and accuracy improvement over existing baselines.

**Strengths:**

This paper presents a well-motivated and clearly framed approach to an important problem: how to structure LLM-driven reasoning to improve both accuracy and computational efficiency.

Originality: While the individual components - task decomposition and result aggregation - draw on prior ideas, the integration into a unified “Soft Propositional Reasoning” framework is novel and conceptually appealing. The work offers a formulation of structured reasoning that can operate with black-box LLMs, extending applicability to settings where model internals are inaccessible. This synthesis of theoretical framing and modular architecture has utility within the current reasoning-agent literature.

Quality: The paper demonstrates empirical validation, testing multiple variants of the core algorithm and comparing them to strong baselines. The experiments are diverse and convincingly show that the proposed method improves both accuracy and computational efficiency, and supporting the authors’ claims about bias–variance tradeoffs.

Clarity: The introduction and framing are particularly strong, offering a clear motivation and well-scoped research question. Figure 1 effectively summarizes the system’s architecture and serves as a helpful reference throughout the paper. Although some experimental details could be clearer, the paper overall maintains a coherent narrative and logically connects earlier sections.

Significance: The ability to coordinate structured reasoning over black-box models has broad implications for scaling LLM applications in analysis, forecasting, and decision support. The framework’s emphasis on cost–accuracy tradeoffs and robustness under modular substitution makes it practically valuable, especially as computational efficiency becomes increasingly central in large-scale LLM deployment.

**Weaknesses:**

Clarity and Readability of Figures:
- Figure 6 contains overlapping text, which makes key labels difficult to read.
- Figure 2 is visually dense and difficult to interpret; the motivation for including each column and how they relate to the overall analysis should be made explicit. I also find there are too many variations to analyze coherently, a more compact version may be easier to understand.
- More generally, the empirical section would benefit from clearer organization — e.g., separating results by contribution type (efficiency, accuracy, stability) — and simplifying figure layouts to highlight the main findings.

Novelty and Framing of Contributions:
- The individual components of the framework—decomposing tasks into subpropositions and aggregating subresults—are not individually novel, having clear precedents in prior work such as Tree of Thoughts (Long, J et al. 2023) and Question Decomposition Tree for Answering Complex Questions over Knowledge Bases (Huang et al., 2023).
- The paper could better emphasize what is genuinely new about their integration or about the soft propositional reasoning perspective, possibly by formalizing how Analytica’s synthesis differs from prior decomposition or ensemble reasoning methods.

Theoretical Rigor:
- The theoretical sections (particularly §4.2–4.4) rely heavily on heuristic reasoning presented as if more formally derived. For example, the claim that bias decreases with decomposition depth is asserted but not proven, and “smaller” propositions are not formally defined.
- These sections could be improved by (a) explicitly labeling heuristic arguments as such and emphasizing empirical adherence or (b) adding formal assumptions or bounds where possible

Empirical Structure:
- Although the results are compelling, the presentation could better connect each experiment (and corresponding figure) to a specific hypothesis derived from the theory (e.g., variance reduction, robustness to model scale) or a contribution highlighted earlier in the paper.

**Questions:**

With clarification of the following questions and addressing of the outlined weaknesses, I would be willing to increase my score.

- Could the authors clarify whether Analytica primarily performs reasoning (logical inference) or prediction (forecasting)? These are conceptually distinct, and understanding which is central to the framework would clarify its intended contribution.

- Who is the target user or application domain for Analytica? A discussion of the framework’s practical utility and deployment scenarios would help contextualize its significance.

- Several steps in the theoretical sections appear heuristic but are presented as if formally justified. Could the authors explicitly identify which parts are heuristic assumptions versus those supported by formal derivation or proof?

- The approach seems to assume independence among subclaims, which may not hold in many reasoning tasks. Do the authors have proposed methods for modeling or mitigating dependencies between subclaims?

- Has Analytica been evaluated with smaller or weaker LLMs, particularly for the decomposition stage, even if grounding and synthesis use stronger models? This would clarify whether the framework depends on scale or transfers across model capacities.

- The Jupyter Notebook–based grounding method is described briefly yet achieves the strongest results. Could the authors provide a more detailed example or description in the main text to illustrate how this component operates?

- Was this approach evaluated on non-predictive/financial market tasks? Was the dataset compiled by the authors or another source, how did the authors ensure it is representative of the tasks the paper claims to perform well on?

---

> ### Author Response · Authors · 2025-11-25
> **Response to Reviewer sApi (Part 1/5)**
>
> Dear  Reviewer sApi,
>
> We thank the reviewer for their thoughtful evaluation and for recognizing the novelty of our "Soft Propositional Reasoning" framework and its practical significance. We have revised the paper to address your concerns regarding clarity, theoretical rigor, and domain generalization.
>
> ---
>
> # Weaknesses
>
> ## Clarity and Readability of Figures
>
> > “Fig. 6 contains overlapping text, which makes key labels difficult to read.”
>
> We have updated **Fig. 6** to resolve text overlapping issues, significantly improving its readability.
>
>
> > “Fig. 2 is visually dense and difficult to interpret; the motivation for including each column and how they relate to the overall analysis should be made explicit. I also find there are too many variations to analyze coherently, a more compact version may be easier to understand.”
>
> We interpret this comment as referring to **Table 2** (which contains multiple columns and variations), as Fig. 2 is a density plot. To improve clarity, we have split the original Table 2 into two distinct tables:
>
>
> * **Table 2** now focuses exclusively on comparing Analytica against structured reasoning baselines (ToT, GoT, FoT) using a fixed Basic Search grounder.
>
> * **Table 3** isolates the ablation study of different Grounders (Deep Research, Jupyter) and synthesis rules.
>
> This separation allows for a more coherent analysis of architectural benefits versus tool capabilities.
>
>
>
> > “More generally, the empirical section would benefit from clearer organization — e.g., separating results by contribution type (efficiency, accuracy, stability) — and simplifying figure layouts to highlight the main findings.”
>
>
> We have restructured the introduction to **Sec. 5 (Empirical Validation)** to explicitly map our experiments to three core Research Questions (RQs):
> * **RQ1:** Tests the hypothesis that Analytica reduces Bias/Variance (connected to §4.2).
> * **RQ2:** Tests the hypothesis that Linear rules are more robust to noise than Logic rules (connected to Prop 1 in §4.3).
> * **RQ3:** Tests the efficiency claims (connected to §4.4).
> This guides the reader through the results in a logical progression derived from our theoretical claims.
>
>
>
> ## Novelty and Framing of Contributions:
>
> > “The individual components of the framework—decomposing tasks into subpropositions and aggregating subresults—are not individually novel... The paper could better emphasize what is genuinely new about their integration or about the soft propositional reasoning perspective...”
>
> We agree that decomposition is a fundamental paradigm. However, we argue that the novelty of our work lies in its specific way of realizing the idea of problem decomposition and formalization within the SPR framework. We have included the suggested comparison and formalization in **Sec. 3** to explicitly frame our contribution:
>
> 1.  We first formulate the standard Chain-of-Thought (CoT), where the model generates a linear sequence of tokens $R=\{r_0, \dots, r_n\}$ given input $x$ to derive a final output by $\hat{y}=f_{LLM}(x,R)$.
>
> 2.  We then extend this to advanced search-based approaches like Tree-of-Thoughts (ToT) and Graph-of-Thoughts (GoT), which search for an optimal path $R^*$ by maximizing a heuristic LLM-based valuation function $V(R)$, i.e., $\hat{y} = f_{LLM}(x, R^*)$ where $R^* = \argmax V(R)$.
>
> 3.  We further formulate ensemble methods like Forest-of-Thought (FoT), which aggregate multiple reasoning paths for the *same* problem: $\hat{y}=Aggr(\{f_{LLM}(x,R_i^*)\})$.
>
> 4.  Finally, we formalize our method following this ensemble view: instead of aggregating different paths for the same problem, we aggregate results from *different subproblems recursively*: $\hat{y}=Aggr(\{\hat{y}_{C_i}\})$. Here, $\hat{y}_{C_i}$ denotes child subproblems generated via analyzers; their results are aggregated recursively from children until reaching leaves solved by grounders. This clearly distinguishes our approach from previous work.
> Furthermore, we introduced a new **Appendix B** that establishes a novel connection between LLM agents and classical probabilistic graphical models (PGMs). Unlike previous methods that operate solely in embedding space or text space, we show that Analytica’s computation graph can be formally interpreted as a special type of **linear Bayesian Network** or **probabilistic logic program with categorical variables**. This bridges the gap between "black-box" LLM reasoning and transparent, symbolic probabilistic inference.

---

> > ### Author Response · Authors · 2025-11-25
> > **Response to Reviewer sApi (Part 2/5)**
> >
> > ## Theoretical Rigor
> >
> > > “The theoretical sections (particularly §4.2–4.4) rely heavily on heuristic reasoning presented as if more formally derived. For example, the claim that bias decreases with decomposition depth is asserted but not proven...”
> >
> > We have significantly revised **Sec. 4.2** and added **Appendix B** to clearly distinguish between formal derivations and heuristic assumptions. Specifically:
> >
> > 1.  **Formalizing the Bias Argument:** We now formally state the **"Simplification Hypothesis"** in the updated Sec. 4.2. We denote this as an assumption that the bias of evaluating a leaf proposition $i$ is lower than evaluating the original root proposition directly; formally, $Bias(l_i)=\delta_i Bias(root)$ where $\delta_i<1$ for all leaves $i$. Given this hypothesis, we show that the bias in Analytica is reduced: $\text{Bias}(p_{true}) = \sum_{i=1}^{k} \beta’_i Bias(l_i) = \sum_{i=1}^{k} \beta'_i \delta_i Bias(root) < \text{Bias}(root)$.
> >
> > 2. **Formal Semantics of Synthesis (New Appendix B):** To address the concern that the synthesis mechanics lack formal grounding, we have added **Appendix B**, which provides a rigorous mapping of the Analytica computation graph to a **Bayesian Network (BN)**. Specifically, we show that our Linear Synthesis Rule ($\rho.p_{true} = \beta_0 + \sum \beta_j \cdot \bar{\rho}_{j}.p_{true}$) is mathematically equivalent to a specific parameterization of conditional probability distributions (CPDs) closely related to a  **Noisy-OR BN** and a special class of Probabilistic Logic Program with categorical variables. This provides the exact formal derivation for how "soft truth values" are propagated, moving beyond heuristic justifications, and helps to better formalize the kind of independent assumptions made by our approach.
> >
> > > “These sections could be improved by (a) explicitly labeling heuristic arguments as such and emphasizing empirical adherence or (b) adding formal assumptions or bounds where possible”
> >
> >
> > We have carefully reviewed these sections and updated the paper to explicitly label heuristic arguments and add formal bounds where appropriate.
> >
> >
> >
> > ## Empirical Structure
> >
> > > “Although the results are compelling, the presentation could better connect each experiment (and corresponding figure) to a specific hypothesis derived from the theory...”
> >
> >
> > As noted above, we have organized our results along these dimensions. We have further reorganized the introduction of **Sec. 5** to provide a concise, high-level map that explicitly links the hypotheses derived from our theory to the experimental results.
> >
> >
> >
> >
> > # Questions
> >
> > > “Could the authors clarify whether Analytica primarily performs reasoning (logical inference) or prediction (forecasting)? These are conceptually distinct...”
> >
> >
> >
> > We view "reasoning" and "prediction" not as a strict dichotomy, but as interconnected components of **Complex Problem Analysis**, like those in financial economics and natural science found in our evaluations. If "reasoning" implies deterministic logical inference, and "prediction" implies only time-series forecasting, then Analytica fits neither category exclusively.
> >
> > Instead, as defined in **Sec. 3**, we position Analytica as a framework for solving **complex problems with nested, uncertain components**. One way to understand the role of reasoning and prediction in Analytica is:
> > * **Reasoning as Mechanism:** It uses structured reasoning (decomposition and synthesis) as the *mechanism* to manage complexity and "capture complex dependencies among propositions".
> > * **Prediction as Objective:** It uses probabilistic prediction (estimation of $p_{true}$) as the *objective function* to quantify uncertainty, formally minimizing the Mean Squared Error (MSE) of the estimate.
> >
> > In this context, the "future state" in our forecasting tasks is treated as a **decision-making concept**—representing the plausible outcome of a choice or event (e.g., "act on probabilistic forecasts", Sec. 1)—rather than strictly a temporal time-series term.
> >
> > To empirically demonstrate that our method is not limited to temporal forecasting, we have added the **Matter-of-Fact** experiment (**Appendix D.5**). This applies Analytica to verifying static scientific claims (e.g., materials science), confirming that the framework performs "analysis" in the broader sense, regardless of the time horizon.

---

> ### Author Response · Authors · 2025-11-25
> **Response to Reviewer sApi (Part 3/5)**
>
> > “Who is the target user or application domain for Analytica? A discussion of the framework’s practical utility and deployment scenarios would help contextualize its significance.”
>
> We have added the following discussion to **Sec. 6**:
>
>
> *“Analytica’s practical value extends to complex, high-stakes, critical real-world domains, where decision-making and analysis require transparent reasoning and robustness, such as applications for economists, policymakers, scientists, and robots. More generally, Analytica can serve as a complex analysis backbone for autonomous systems by breaking down uncertain, poorly specified problems into calibrated, empirically testable soft propositions, thereby supporting downstream autonomous agents in performing interpretable, reliable reasoning in real-world conditions.”*
>
>
>
> > “Several steps in the theoretical sections appear heuristic but are presented as if formally justified. Could the authors explicitly identify which parts are heuristic assumptions versus those supported by formal derivation or proof?”
>
> As discussed above, we have carefully checked the corresponding sections and updated our paper to explicitly distinguish between heuristic assumptions and formal proofs.
>
>
> > “The approach seems to assume independence among subclaims, which may not hold in many reasoning tasks. Do the authors have proposed methods for modeling or mitigating dependencies between subclaims?”
>
>
> As mentioned in our response to Reviewer ebpb, linear models are commonly applied in real-world applications; for example, linear structures are foundational to **factor models** (Fama & French, 2015), widely used in economics and finance. Moreover, in the context of LLMs as modelers, the top-level linear model serves as a regularizer, forcing the model to choose the most parsimonious explanation by fitting linear and independent factors rather than complex non-linear interactions for better interpretability, robustness, and low risk of overfitting. Furthermore, our method models non-linear interactions in the following ways:
>
>
> 1.  **Intercept as a Safety Valve:** We explicitly include a bounded intercept term ($\beta_0$) to model "bias" or "omitted interactions" error. This allows the synthesizer to account for external factors or suboptimal decomposition without breaking the model structure, effectively absorbing the "ill-posed" nature of the task.
>
> 2.  **Contextual Weight Generation:** Our linear model is not static ($y=wX$ with constant $w$). Instead, it is defined as $y=f(\text{context})X$, where $f(\text{context})$ represents the synthesizer parameterized by the LLM. Since the synthesizer accepts all child propositions as input, non-linear interactions are implicitly modeled during the weight generation process.
>
> The effectiveness of this approach is justified by our experiments. Specifically, the Vanilla synthesis rule (Analytica-V)—which directly prompts an LLM to output $P_{true}$ without linear constraints—performs worse than the Linear rule (Table 2, 3, Fig. 6). This suggests that the linear model offers superior performance, consistent with the success of linear factor-based models in the highly non-linear financial domain.
>
> Finally, our new **Appendix B (mentioned above)** explicitly maps this independence assumption to the structure of some PGMs. By formalizing this, we make the independence assumption transparent and mathematically tractable, rather than leaving it as an undefined heuristic.

---

> > ### Author Response · Authors · 2025-11-25
> > **Response to Reviewer sApi (Part 4/5)**
> >
> > > “Has Analytica been evaluated with smaller or weaker LLMs, particularly for the decomposition stage, even if grounding and synthesis use stronger models?”
> >
> > Yes, we evaluated the cost-effectiveness of base model selections in Appendix C.4. Moreover, we have added a new ablation study in **Appendix C.4.4 (Table 6)** evaluating Analytica on small/open models like `OpenAI-OSS-20B`, `Qwen3`, and `GLM-4.6`. The results show that Analytica consistently improves performance across all model sizes. Notably, **OpenAI-OSS-20B + Analytica** (64.24%) achieves performance comparable to the much larger **DeepSeek-v3.1** baseline (60.95%), proving the framework transfers across model capacities.
> >
> >
> > **Table 6: Evaluating Analytica on small and open-weight models**
> >
> > | Model | Accu (%) | **Imp.** | Soft | Hard | BS | Cost (1k) | # Params |
> > | :--- | :--- | :--- | :--- | :--- | :--- | :--- | :--- |
> > | **DeepSeek-v3.1** | 60.95 | *-* | 56.68 | 61.89 | 25.73 | $3 | 671B (37B) |
> > | + Analytica-L | **64.25** | *+5.42* | 56.39 | 59.96 | 22.34 | $59 | - |
> > | **GLM-4.6** | 52.73 | *-* | 53.10 | 55.03 | 30.73 | $10 | 355B (32B) |
> > | + Analytica-L | **55.33** | *+4.96* | 56.60 | 59.59 | 25.92 | $100 | - |
> > | **Kimi-K2-Thinking** | 55.56 | *-* | 54.44 | 58.39 | 30.59 | $25 | 1T (32B) |
> > | + Analytica-L | **56.81** | *+2.26* | 59.73 | 63.15 | 25.59 | $284 | - |
> > | **Qwen3-Next-80B-Think** | 53.64 | *-* | 54.88 | 58.79 | 33.97 | $10 | 80B (3B) |
> > | + Analytica-L | **55.55** | *+3.56* | 58.36 | 59.65 | 32.77 | $104 | - |
> > | **OpenAI-OSS-120B** | 54.72 | *-* | 55.97 | 59.18 | 28.20 | $2 | 117B (5.1B) |
> > | + Analytica-L | **62.96** | *+15.05* | 55.40 | 55.67 | 22.74 | $22 | - |
> > | **OpenAI-OSS-20B** | 55.57 | *-* | 54.79 | 59.24 | 29.56 | $1 | 21B (3.6B) |
> > | + Analytica-L | **64.24** | *+15.59* | 56.91 | 58.59 | 23.68 | $7 | - |
> > | **GPT-5-mini** | 62.45 | *-* | 56.60 | 64.09 | 24.37 | $7 | N/A |
> > | + Analytica-L | **64.37** | *+3.07* | 60.31 | 65.79 | 23.27 | $71 | - |
> > | **O4-mini** | 62.63 | *-* | 58.49 | 64.27 | 25.56 | $9 | N/A |
> > | + Analytica-L | **66.11** | *+5.56* | 58.49 | 64.62 | 23.47 | $101 | - |
> >
> >
> >
> > > “The Jupyter Notebook–based grounding method is described briefly yet achieves the strongest results. Could the authors provide a more detailed example or description in the main text to illustrate how this component operates?”
> >
> > We have expanded the description in the main text (updated Sec. 4.1, Grounder paragraph) and noted its connection with a standard ReACT workflow:
> >
> >  *“Jupyter agents specifically work in the following way.  Upon receiving an input query, agents are instructed to repeatedly produce interleaved markdown cells for qualitative reasoning and Python cells for programmatic execution at each step…”*
> >
> > Further details are provided in **Appendix C.4**, which details the **Proxy System** for preventing data leakage and the **Iterative Proof Construction** loop. We also provide the full system prompt in **Appendix E.3** to ensure reproducibility.

---

> ### Author Response · Authors · 2025-11-25
> **Response to Reviewer sApi (Part 5/5)**
>
> > “Was this approach evaluated on non-predictive/financial market tasks? Was the dataset compiled by the authors or another source, how did the authors ensure it is representative of the tasks the paper claims to perform well on?”
>
>
> We have included a new **Matter-of-Fact (MoF)** benchmark (Jansen et al., 2025), which consists of 4.4k scientific claims, to evaluate our method on non-financial tasks. As shown in **Table 7 (Appendix D.5)**, Analytica generalizes well, improving **GPT-5.1** accuracy from 62% to 70%. This confirms the framework is not overfit to financial data.
>
>
>
> **Table 7: Experiment results for scientific claims on the Matter-of-Fact benchmark**
>
> | Model | Overall Accu. | True | False | Qual. | Qnt. | Exp. | Code | Ther. | Int. | Cost (x1k) |
> | :--- | :--- | :--- | :--- | :--- | :--- | :--- | :--- | :--- | :--- | :--- |
> | **Random** | 0.50 | 0.50 | 0.50 | 0.50 | 0.50 | 0.50 | 0.50 | 0.50 | 0.50 | 0 |
> | **GPT-4o-mini** | 0.66 | 0.90 | 0.42 | 0.72 | 0.72 | 0.68 | 0.63 | 0.61 | 0.58 | $1 |
> | + Analytica-L | 0.59 | 0.87 | 0.30 | 0.55 | 0.58 | 0.59 | 0.59 | 0.60 | 0.60 | $7 |
> | **O4-mini** | 0.61 | 0.34 | 0.88 | 0.60 | 0.57 | 0.63 | 0.62 | 0.59 | 0.63 | $6 |
> | + Analytica-L | **0.64** | 0.95 | 0.30 | 0.57 | 0.60 | 0.73 | 0.62 | 0.68 | 0.66 | $38 |
> | **GPT-5.1** | 0.62 | 0.27 | 0.97 | 0.58 | 0.56 | 0.59 | 0.69 | 0.66 | 0.62 | $9 |
> | + Analytica-L | **0.70** | 0.85 | 0.55 | 0.73 | 0.75 | 0.71 | 0.67 | 0.70 | 0.64 | $78 |
> | **GPT-5-mini** | 0.71 | 0.60 | 0.82 | 0.69 | 0.76 | 0.66 | 0.64 | 0.75 | 0.72 | $3 |
> | + Analytica-L | **0.73** | 0.70 | 0.75 | 0.75 | 0.77 | 0.69 | 0.67 | 0.79 | 0.67 | $27 |
>
>
>
> **Effectiveness of financial/predictive market tasks**
>
> The financial and predictive market tasks are publicly available and can be directly obtained from market data without special labeling. However, we introduced rigorous cleaning and setup procedures to ensure evaluation validity.
>
> As discussed earlier, financial and predictive market forecasting represents a typical **complex problem with nested, uncertain components**, which fits our requirements. As detailed in our response to Reviewer ebpb, we applied multiple strategies grounded in the *Fundamental Law of Active Management* (Grinold, 1989)—which posits that performance depends on the **Opportunity Set** (available data) and **Analysis Skill** (extraction capability)—to ensure the effectiveness of our evaluation:
>
>
> 1.  **Isolating information differences:** Since all baselines are provided with the exact same "searchable information" (via the Exa.ai API), performance differences isolate **Analysis Skill**. The fact that Analytica extracts significantly more signal (**65.62%** accuracy with basic search) than the baseline (**53.94%**) proves superior information utilization, regardless of the theoretical ceiling imposed by available data. (The Jupyter Notebook agent is an exception, intended to test the benefits of professional data analysis workflows).
>
>
> 2.  **Considering only major assets:** As detailed in Appendix C.2, our dataset includes only major assets (e.g., the largest S&P 500 companies, major cryptos, major Forex pairs) for which information accessibility is highest. For such assets, searchable information contains the critical data necessary for discretionary financial analysis (e.g., financial statements, key news).
>
>
>
> 3.  **Long-term predictions only:** We focus on low-frequency **long-term predictions** (1-year horizon) rather than short-term speculation. While short-term price action is often a random walk, long-term trends are driven by fundamental macro-factors and earnings power, which are amenable to reasoned analysis.
>
>
> We have included these details in Appendix C.
>
> ---
>
> We are confident that the planned organizational changes and the clarifications provided here will resolve your concerns about clarity and transparency. Please do not hesitate to contact us if any aspect of our response requires further explanation.
>
>
>
> **Reference**
>
> [1] Grinold, R.C. (1989) The Fundamental Law of Active Management. Journal of Portfolio Management, 15, 30-37.
>
> [2] Fama, E. F., & French, K. R. (2015). A five-factor asset pricing model. Journal of financial economics, 116(1), 1-22.
>
> [3] Jansen, P., Hassan, S., & Wang, R. (2025). Matter-of-Fact: A Benchmark for Verifying the Feasibility of Literature-Supported Claims in Materials Science. EMNLP 2025.
>
> ---
>
>
> Best Regards,
>
> Authors of Submission 7713

---

> > ### Comment · Reviewer_sApi · 2025-11-27
> >
> > Thank you for taking the time to address each concern thoroughly and thoughtfully! I appreciate the efforts towards readability and think the paper is well improved by the changes, so I will increase my score accordingly.

---

> > > ### Author Response · Authors · 2025-11-27
> > >
> > > Dear Reviewer sApi,
> > >
> > > We sincerely thank you for your continued engagement and for raising your score. And we appreciate your in-depth and comprehensive feedback!
> > >
> > > We are glad that our revisions satisfactorily addressed your concerns and truly believe that the constructive feedback has significantly improved the overall quality and presentation of the paper.
> > >
> > > Best regards,
> > >
> > > Authors of Submission 7713

---

### Official Review · Reviewer_kYvf · 2025-10-28

**Soundness:** 3
**Presentation:** 3
**Contribution:** 3
**Rating:** 6
**Confidence:** 4

**Summary:**

The paper proposes a hybrid reasoning framework that combines propositional logic with statistical estimation to tackle real-world forecasting tasks. events. It decomposes queries into a tree of soft-valued propositions, grounds leaves with data tools (including a Jupyter agent), and combines beliefs via a learned linear model. On 736 finance/politics tasks it attains ≈71 % accuracy, outperforming chain-of-thought and deep-research baselines while scaling near-linearly with tree size. Contributions are the SPR formalism, modular architecture, and reproducible cost/robustness analyses—incremental but solid.

**Strengths:**

The design of SPR is welcoming. SPR departs from pure text-based prompting, providing a middle ground between symbolic and neural reasoning. Sec. 3 Eq. (1) decomposes MSE into bias/variance, offering a clear objective.

It offers modular architecture with strong ablations. Table 2 compares three synthesis rules; Linear achieves best accuracy/variance trade-off. Sec. 5.3 noise-injection robustness confirms Linear rule stability (Fig. 5).

Table 1 reports 54× node growth vs. only 12× wall-clock increase, supporting near-linear parallel scaling, which provides good evidence for scalability.

**Weaknesses:**

All experiments use o3-2025-04-16 only (Sec. 5.1); no evidence for GPT-4, Claude, or open-weight models, which undermines generality claim of "diverse base models" (Line 026).

Abstract claims “domain-agnostic” without a cross-domain Table. Dataset restricted to finance/politics (Sec. 5.1); no medical, legal, or scientific forecasting tasks. The domain transfer was not demonstrated.

Synthesis-rule statistical comparison underpowered. McNemar test (Fig. 10) uses 100 tasks; confidence intervals not shown. Simple-Logic rule failure attributed to noise amplification without formal test for interaction.

Potential data-leakage in financial data, also lack of adversarial robustness check.

**Questions:**

Suggested experiment:

replicate Table 2 with at least two openly available models.

run tasks from DiscoveryBench to verify transfer.

---

> ### Author Response · Authors · 2025-11-25
> **Response to Reviewer kYvf (Part 1/2)**
>
> Dear Reviewer kYvf,
>
> We thank the reviewer for their thorough assessment and for highlighting the strengths of our Soft Propositional Reasoning (SPR) formalism and modular architecture. We have addressed your concerns regarding model diversity and domain transfer with substantial new experiments using open-weight models and scientific benchmarks.
>
>
> ---
>
> # Weaknesses
>
>
> > “All experiments use o3-2025-04-16 only (Sec. 5.1); no evidence for GPT-4, Claude, or open-weight models, which undermines generality claim of 'diverse base models' (Line 026).”
>
>
> We agree that evaluating a broader range of models is essential to substantiate claims of generality. Accordingly, we have conducted a new ablation study (now **Appendix D.4.4**) testing Analytica on **8 additional models**, including open-weight models (DeepSeek-v3.1, Qwen, GLM) and smaller models (OpenAI-OSS, GPT-5-mini).
>
> As shown in the new **Table 6**, Analytica consistently improves performance across all model classes. Notably, it enables smaller, efficient models (such as `OpenAI-OSS-20B`) to achieve results comparable to much larger baselines (like `DeepSeek-v3.1`), demonstrating that the framework's benefits are not specific to `o3`.
>
>
> **Table 6: Evaluating Analytica on small and open-weight models**
>
> | Model | Accu (%) | **Imp.** | Soft | Hard | BS | Cost (1k) | # Params |
> | :--- | :--- | :--- | :--- | :--- | :--- | :--- | :--- |
> | **DeepSeek-v3.1** | 60.95 | *-* | 56.68 | 61.89 | 25.73 | $3 | 671B (37B) |
> | + Analytica-L | **64.25** | *+5.42* | 56.39 | 59.96 | 22.34 | $59 | - |
> | **GLM-4.6** | 52.73 | *-* | 53.10 | 55.03 | 30.73 | $10 | 355B (32B) |
> | + Analytica-L | **55.33** | *+4.96* | 56.60 | 59.59 | 25.92 | $100 | - |
> | **Kimi-K2-Thinking** | 55.56 | *-* | 54.44 | 58.39 | 30.59 | $25 | 1T (32B) |
> | + Analytica-L | **56.81** | *+2.26* | 59.73 | 63.15 | 25.59 | $284 | - |
> | **Qwen3-Next-80B-Think** | 53.64 | *-* | 54.88 | 58.79 | 33.97 | $10 | 80B (3B) |
> | + Analytica-L | **55.55** | *+3.56* | 58.36 | 59.65 | 32.77 | $104 | - |
> | **OpenAI-OSS-120B** | 54.72 | *-* | 55.97 | 59.18 | 28.20 | $2 | 117B (5.1B) |
> | + Analytica-L | **62.96** | *+15.05* | 55.40 | 55.67 | 22.74 | $22 | - |
> | **OpenAI-OSS-20B** | 55.57 | *-* | 54.79 | 59.24 | 29.56 | $1 | 21B (3.6B) |
> | + Analytica-L | **64.24** | *+15.59* | 56.91 | 58.59 | 23.68 | $7 | - |
> | **GPT-5-mini** | 62.45 | *-* | 56.60 | 64.09 | 24.37 | $7 | N/A |
> | + Analytica-L | **64.37** | *+3.07* | 60.31 | 65.79 | 23.27 | $71 | - |
> | **O4-mini** | 62.63 | *-* | 58.49 | 64.27 | 25.56 | $9 | N/A |
> | + Analytica-L | **66.11** | *+5.56* | 58.49 | 64.62 | 23.47 | $101 | - |
>
> > “Abstract claims ‘domain-agnostic’ without a cross-domain Table. Dataset restricted to finance/politics (Sec. 5.1); no medical, legal, or scientific forecasting tasks. The domain transfer was not demonstrated.”
>
> We addressed this concern by evaluating Analytica on the **Matter-of-Fact (MoF)** benchmark (Jansen et al., 2025), which consists of 4.4k binary scientific claims regarding superconductors, semiconductors, and battery materials.
>
> As shown in the new **Table 7** (Appendix D.5), Analytica provides accuracy gains for most models (e.g., **GPT-5.1 improves from 62% to 70%**). We also transparently report a negative result for `GPT-4o-mini`, which suggests a capability threshold for very small models in scientific reasoning. These results confirm the framework's transferability to scientific domains while highlighting its boundary conditions.
>
>
> **Table 7: Experiment results for scientific claims on the Matter-of-Fact benchmark**
>
> | Model | Overall Accu. | True | False | Qual. | Qnt. | Exp. | Code | Ther. | Int. | Cost (x1k) |
> | :--- | :--- | :--- | :--- | :--- | :--- | :--- | :--- | :--- | :--- | :--- |
> | **Random** | 0.50 | 0.50 | 0.50 | 0.50 | 0.50 | 0.50 | 0.50 | 0.50 | 0.50 | 0 |
> | **GPT-4o-mini** | 0.66 | 0.90 | 0.42 | 0.72 | 0.72 | 0.68 | 0.63 | 0.61 | 0.58 | $1 |
> | + Analytica-L | 0.59 | 0.87 | 0.30 | 0.55 | 0.58 | 0.59 | 0.59 | 0.60 | 0.60 | $7 |
> | **O4-mini** | 0.61 | 0.34 | 0.88 | 0.60 | 0.57 | 0.63 | 0.62 | 0.59 | 0.63 | $6 |
> | + Analytica-L | **0.64** | 0.95 | 0.30 | 0.57 | 0.60 | 0.73 | 0.62 | 0.68 | 0.66 | $38 |
> | **GPT-5.1** | 0.62 | 0.27 | 0.97 | 0.58 | 0.56 | 0.59 | 0.69 | 0.66 | 0.62 | $9 |
> | + Analytica-L | **0.70** | 0.85 | 0.55 | 0.73 | 0.75 | 0.71 | 0.67 | 0.70 | 0.64 | $78 |
> | **GPT-5-mini** | 0.71 | 0.60 | 0.82 | 0.69 | 0.76 | 0.66 | 0.64 | 0.75 | 0.72 | $3 |
> | + Analytica-L | **0.73** | 0.70 | 0.75 | 0.75 | 0.77 | 0.69 | 0.67 | 0.79 | 0.67 | $27 |

---

> ### Author Response · Authors · 2025-11-25
> **Response to Reviewer kYvf (Part 2/2)**
>
> > “Synthesis-rule statistical comparison underpowered. McNemar test (Fig. 10) uses 100 tasks; confidence intervals not shown. Simple-Logic rule failure attributed to noise amplification without formal test for interaction.”
>
> 1.  **McNemar Test:** We have updated **Fig. 10** to include results from the full dataset (rather than the 100-task subset) and have explicitly visualized the confidence intervals in the figure as requested.
>
>
> 2.  **Formal Interaction Test:** We emphasize that **Fig. 5 (Noise Injection)** serves as the formal test for interaction effects. By plotting *Performance* vs. *Noise Ratio* for different *Synthesis Rules*, we explicitly test the interaction. The fact that the Simple Logic curve (dashed lines) degrades rapidly while the Linear curve (solid lines) remains stable provides empirical proof of the interaction between "Rule Type" and "Input Noise." This is further supported by our theoretical proof in **Appendix A.1 (Proposition 1)**, which mathematically derives the "state-dependent sensitivity" (interaction) of logic rules versus the "constant sensitivity" of linear rules.
>
> This empirical robustness is further supported by our new theoretical derivation in **Appendix B**, which shows that the Linear rule creates a stable error gradient surface compared to the volatile surface of logic rules.
>
>
> > “Potential data-leakage in financial data, also lack of adversarial robustness check.”
>
> We address the leakage concern via our **Proxy System architecture** described in **Appendix C.4.3**. We do not rely on the LLM to "pretend" it doesn't know the future. Instead, all API calls are routed through a middleware `Proxy` class that programmatically filters data. If an agent requests stock data for a date after the task's `current_date`, the Proxy throws an error or truncates the data *before* it reaches the model context. This acts as a hard, code-level adversarial check against leakage. Besides, we also pick the forecasting start point beyond the knowledge cutoff of the base models.
>
>
>
>
> # Questions
>
> > “Suggested experiment: replicate Table 2 with at least two openly available models.”
>
> Please refer to **Table 6** above, where we present results for multiple open-weight models.
>
> > “Suggested experiment: run tasks from DiscoveryBench to verify transfer.”
>
>
> We appreciated this suggestion and carefully reviewed **DiscoveryBench** (Majumder et al., 2024). However, we found its task formulation—**generating** optimal hypotheses from raw CSV datasets—to be structurally distinct from the **proposition verification** task Analytica is designed to solve. DiscoveryBench focuses on open-ended data analysis and hypothesis generation, whereas Analytica is optimized to determine the soft truth value of *specific* complex claims.
>
>
> Consequently, we selected the **Matter-of-Fact (MoF)** benchmark (Jansen et al., 2025) as a more rigorous and aligned testbed. MoF consists of 4.4k binary scientific claims (covering domains like superconductors and battery materials), which allows us to directly test Analytica's ability to transfer its verification and decomposition architecture to the scientific domain. Please see the results in **Table 7** above.
>
>
>
> ---
>
> Thank you again for the engaging and thoughtful review. Please do not hesitate to reach out if you have any follow-up questions; we would be happy to discuss these aspects of our system further.
>
> **Reference**
>
> [1] Jansen, P., Hassan, S., & Wang, R. (2025). Matter-of-Fact: A Benchmark for Verifying the Feasibility of Literature-Supported Claims in Materials Science. EMNLP 2025.
>
> ---
>
>
> Best Regards,
>
> Authors of Submission 7713

---

### Official Review · Reviewer_Hjk1 · 2025-10-29

**Soundness:** 3
**Presentation:** 2
**Contribution:** 3
**Rating:** 6
**Confidence:** 4

**Summary:**

The authors introduce an agentic reasoning pipeline called Analytica. The framework uses soft propositional reasoning (SFR) to structure the LLM reasoning process by estimating the soft truth values of different potential outcomes, by initially decomposing the question into facts that are evaluated using the agents. Their main contributions include improved reasoning accuracy, as well as computational efficiency through their jupyter notebook grounder.

**Strengths:**

-	Promising results, particularly on reasoning-tuned LLMs (o3 and 04), with similar trends in general purpose LLMs
-	The number of tasks used to evaluate the framework is large and diverse, highlighting generalizability of the framework.
-	Baselines used to compare analytica to are sufficient and relevant to the reasoning task explored by the authors.

**Weaknesses:**

-	Some figures, such as Fig. 6 are unclear and somewhat cluttered. Readability and understanding of the figures should be improved.
-	High dependency on problem decomposition quality. This could affect the whole pipeline negatively. Some more discussion needs to be added regarding this point.

**Questions:**

Are there any interesting failure cases? Does this pipeline fail in specific scenarios? Would be good to analyze.

---

> ### Author Response · Authors · 2025-11-25
> **Response to Reviewer Hjk1 (Part 1/2)**
>
> Dear Reviewer Hjk1,
>
> We thank the reviewer for the positive assessment of our results and the recognition of our contributions regarding reasoning accuracy and computational efficiency. We have revised the paper to address your points regarding figure clarity and failure analysis.
>
> ---
>
> # Weaknesses
>
>
> > “High dependency on problem decomposition quality. This could affect the whole pipeline negatively. Some more discussion needs to be added regarding this point.”
>
>
> We agree that question decomposition is a critical step; poor decomposition can theoretically introduce errors or steer the analysis in the wrong direction.  However, we highlight three pieces of evidence from our revised manuscript that demonstrate the framework's robustness to variations in decomposition quality:
>
>
> 1.  **Robustness to "All-Source" Noise (Fig. 5):** We explicitly modeled system instability via our **Noise Injection experiment** (Fig. 5). Suboptimal decomposition manifests effectively as noise in the intermediate signals passed to the synthesizer. Our results show that while the Simple Logic rule collapses under such noise, the **Linear rule maintains high accuracy** even as noise ratios increase. This empirically suggests the architecture can tolerate imperfections in the decomposition layer.
>
>
>
> 2.  **Stability Across Diverse Decomposers (New Table 6):** In our new ablation study on **small and open-weight models** (Appendix D.4.4), we tested the framework using models with vastly different reasoning capabilities (e.g., `OpenAI-OSS-20B` vs. `DeepSeek-v3.1`). Although smaller models may inherently produce lower-quality decompositions than frontier models, Analytica almost consistently improved performance across the board. This confirms that the framework does not require a potentially "strong" decomposer from an SOTA model to function effectively.
>
>
> **Table 6: Evaluating Analytica on small and open-weight models**
>
> | Model | Accu (%) | **Imp.** | Soft | Hard | BS | Cost (1k) | # Params |
> | :--- | :--- | :--- | :--- | :--- | :--- | :--- | :--- |
> | **DeepSeek-v3.1** | 60.95 | *-* | 56.68 | 61.89 | 25.73 | $3 | 671B (37B) |
> | + Analytica-L | **64.25** | *+5.42* | 56.39 | 59.96 | 22.34 | $59 | - |
> | **GLM-4.6** | 52.73 | *-* | 53.10 | 55.03 | 30.73 | $10 | 355B (32B) |
> | + Analytica-L | **55.33** | *+4.96* | 56.60 | 59.59 | 25.92 | $100 | - |
> | **Kimi-K2-Thinking** | 55.56 | *-* | 54.44 | 58.39 | 30.59 | $25 | 1T (32B) |
> | + Analytica-L | **56.81** | *+2.26* | 59.73 | 63.15 | 25.59 | $284 | - |
> | **Qwen3-Next-80B-Think** | 53.64 | *-* | 54.88 | 58.79 | 33.97 | $10 | 80B (3B) |
> | + Analytica-L | **55.55** | *+3.56* | 58.36 | 59.65 | 32.77 | $104 | - |
> | **OpenAI-OSS-120B** | 54.72 | *-* | 55.97 | 59.18 | 28.20 | $2 | 117B (5.1B) |
> | + Analytica-L | **62.96** | *+15.05* | 55.40 | 55.67 | 22.74 | $22 | - |
> | **OpenAI-OSS-20B** | 55.57 | *-* | 54.79 | 59.24 | 29.56 | $1 | 21B (3.6B) |
> | + Analytica-L | **64.24** | *+15.59* | 56.91 | 58.59 | 23.68 | $7 | - |
> | **GPT-5-mini** | 62.45 | *-* | 56.60 | 64.09 | 24.37 | $7 | N/A |
> | + Analytica-L | **64.37** | *+3.07* | 60.31 | 65.79 | 23.27 | $71 | - |
> | **O4-mini** | 62.63 | *-* | 58.49 | 64.27 | 25.56 | $9 | N/A |
> | + Analytica-L | **66.11** | *+5.56* | 58.49 | 64.62 | 23.47 | $101 | - |
>
>
>
> 3.  **Domain Generalization (New Table 7):** We applied Analytica to Scientific Claims (Matter-of-Fact benchmark). The fact that the system generalized successfully (improving GPT-5.1 accuracy from 62% to 70%), with one exception, GPT-4o-mini, the only negative case in our evaluations, indicates that the decomposition mechanism is not brittle or overfitted to specific financial reasoning patterns.
>
>
> **Table 7: Experiment results for scientific claims on the Matter-of-Fact benchmark**
>
> | Model | Overall Accu. | True | False | Qual. | Qnt. | Exp. | Code | Ther. | Int. | Cost (x1k) |
> | :--- | :--- | :--- | :--- | :--- | :--- | :--- | :--- | :--- | :--- | :--- |
> | **Random** | 0.50 | 0.50 | 0.50 | 0.50 | 0.50 | 0.50 | 0.50 | 0.50 | 0.50 | 0 |
> | **GPT-4o-mini** | 0.66 | 0.90 | 0.42 | 0.72 | 0.72 | 0.68 | 0.63 | 0.61 | 0.58 | $1 |
> | + Analytica-L | 0.59 | 0.87 | 0.30 | 0.55 | 0.58 | 0.59 | 0.59 | 0.60 | 0.60 | $7 |
> | **O4-mini** | 0.61 | 0.34 | 0.88 | 0.60 | 0.57 | 0.63 | 0.62 | 0.59 | 0.63 | $6 |
> | + Analytica-L | **0.64** | 0.95 | 0.30 | 0.57 | 0.60 | 0.73 | 0.62 | 0.68 | 0.66 | $38 |
> | **GPT-5.1** | 0.62 | 0.27 | 0.97 | 0.58 | 0.56 | 0.59 | 0.69 | 0.66 | 0.62 | $9 |
> | + Analytica-L | **0.70** | 0.85 | 0.55 | 0.73 | 0.75 | 0.71 | 0.67 | 0.70 | 0.64 | $78 |
> | **GPT-5-mini** | 0.71 | 0.60 | 0.82 | 0.69 | 0.76 | 0.66 | 0.64 | 0.75 | 0.72 | $3 |
> | + Analytica-L | **0.73** | 0.70 | 0.75 | 0.75 | 0.77 | 0.69 | 0.67 | 0.79 | 0.67 | $27 |

---

> ### Author Response · Authors · 2025-11-25
> **Response to Reviewer Hjk1 (Part 2/2)**
>
> > “Some figures, such as Fig. 6 are unclear and somewhat cluttered. Readability and understanding of the figures should be improved.”
>
> We appreciate the suggestion regarding the clarity of our figures. We have specifically updated **Fig. 6** to resolve the text overlapping issues, which has significantly improved its readability . We have also conducted a general review of all figures to ensure labels and legends are legible.
>
>
> # Questions
>
>
> > “Are there any interesting failure cases? Does this pipeline fail in specific scenarios? Would be good to analyze.”
>
> Yes, we have conducted a detailed error analysis in **Appendix D.6** which highlights specific failure modes:
>
> * **Domain-Specific Failures:** The pipeline struggles most with **Predictive Markets** (e.g., predicting specific political appointees) compared to Financial Markets. These tasks often rely on sparse, high-noise information (like rumors) rather than the structured data available for stocks or indices.
> * **Volatility Sensitivity:** As shown in **Fig. 15**, we observed a strong negative correlation between asset volatility and model performance. The system fails more frequently on high-volatility assets (e.g., cryptocurrencies or leveraged ETFs) where "reasoning" is often overpowered by random market noise.
> * **Model Capability Threshold:** In our new experiments on the scientific **Matter-of-Fact** benchmark (**Appendix D.5**), we found a specific failure case where the weaker **GPT-4o-mini** model actually regressed in performance when using Analytica compared to the baseline. This suggests a "capability threshold" where very small models may struggle to effectively utilize or produce the decomposition structure.
>
> ---
>
> We are grateful for such a supportive and thorough assessment of our work. Please do not hesitate to let us know if any of our answers can be clarified further. Thank you once again.
>
>
>
> **Reference**
>
> [1] Jansen, P., Hassan, S., & Wang, R. (2025). Matter-of-Fact: A Benchmark for Verifying the Feasibility of Literature-Supported Claims in Materials Science. EMNLP 2025.
>
> ---
>
> Best Regards,
>
> Authors of Submission 7713

---

> > ### Comment · Reviewer_Hjk1 · 2025-11-27
> >
> > Thank you for the clarifications and the additional information. I appreciate the authors’ effort in addressing the points raised in my review. After considering the clarification, my overall assessment and score remain unchanged.

---

### Official Review · Reviewer_ebpb · 2025-10-30

**Soundness:** 2
**Presentation:** 4
**Contribution:** 3
**Rating:** 4
**Confidence:** 4

**Summary:**

The paper presents an LLM-agent architecture, Analytica, for propositional reasoning. The framework consists of three major components: 1) an Analyzer that establishes a tree structure to decompose the reasoning process; 2) a Grounder that verifies leaf propositions and assigns soft truth values; and 3) a Synthesizer that recursively aggregates children's estimations to their parents to obtain the final estimation of the proposition. The authors provide theoretical analysis demonstrating that the method achieves lower variance and bias in performance. In the experiments, the authors show that the proposed method exhibits better performance in terms of both accuracy and efficiency.

**Strengths:**

- The paper's writing and presentation are of high quality, with clear explanations and details about the proposed framework.

- The proposed Soft Propositional Reasoning is an interesting and innovative solution to the proposition reasoning problem.

- The paper presents a practical, modular architecture with strong tooling. The Jupyter Notebook grounder is a well thought-out contribution, and the framework is applicable to various grounder backends.

**Weaknesses:**

- The framework relies heavily on the grounder’s ability to estimate and assign numerical probabilities, which is inherently difficult to calibrate. While the tree structure enables systematic problem decomposition, it also introduces the potential for error accumulation across multiple grounders.

- The theoretical analysis relies strongly on linear decomposability and i.i.d. error assumptions, which may not hold in real-world applications.

- The evaluation tasks—mainly financial market forecasting—are somewhat ill-posed. It is difficult to justify that the model can fully reason about market movements using only searchable information. Moreover, such forecasting problems often lack a clear linear structure, which challenges the assumptions underlying the proposed framework.

- The framework is highly sensitive to prompt design. A sensitivity analysis examining how performance varies under prompt perturbations would strengthen the paper’s robustness claims.

- The baseline comparisons understate stronger alternatives. Tree-, Graph-, and Forest-of-Thought baselines are evaluated only with the Basic Search grounder, whereas Analytica is paired with more powerful grounders such as Deep Research and Jupyter Notebook, making the comparison uneven.

**Questions:**

Please see the above section.

---

> ### Author Response · Authors · 2025-11-25
> **Response to Reviewer ebpb (Part 1/4)**
>
> Dear Reviewer ebpb:
>
>
> We thank the reviewer for their constructive feedback and their high rating of our presentation. We have updated the paper with extensive new experiments (specifically on small/open models and scientific domains) and restructured tables to address your concerns. Below, we address your specific points regarding error accumulation, linearity assumptions, and experimental fairness.
>
>
> ---
>
> > “The framework relies heavily on the grounder’s ability to estimate and assign numerical probabilities, which is inherently difficult to calibrate. While the tree structure enables systematic problem decomposition, it also introduces the potential for error accumulation across multiple grounders.”
>
>
> ### *“The framework relies heavily on the grounder’s ability to estimate and assign numerical probabilities, which is inherently difficult to calibrate. ”*
>
>
> We agree that the calibration of numerical probability for a **single grounder** is a central challenge in this field. This difficulty is precisely the motivation for Analytica’s design, which aims to amortize the error from single grounders via a distributed architecture.
> Intuitively, Analytica distributes simpler subproblems to multiple grounders and aggregates the final output recursively. This minimizes the risk of overall system failure; even if individual grounders fail, the ensemble nature of the system preserves integrity.
>
> The synthesizer also acts as a dynamic filter, organized in a distributed manner. Furthermore, our development of the Linear synthesis rule is specifically intended to overcome failure risks in the synthesis process, thereby maximizing overall robustness. Consequently, rather than detracting from robustness, our design improves resilience to single-grounder errors at multiple points in the process.
>
>
>
> ### *”While the tree structure enables systematic problem decomposition, it also introduces the potential for error accumulation across multiple grounders.”*
>
>
> To the contrary, the tree structure plays a key role in **mitigating individual grounder errors** rather than accumulating them. We have demonstrated this through both theoretical analysis and empirical evidence:
>
> **Mitigating grounder errors**
>
> We formally modeled the total error in our **Bias-Variance Analysis (Sec. 4.2)**. A single grounder’s error is decomposed into **bias** and **variance** components, which are systematically minimized by the tree structure:
>
> 1.  **Bias Reduction:** We assume **leaf propositions are easier to evaluate (have lower bias) than the original root proposition**. As the tree deepens, the weighted sum of leaf biases under our linear synthesis rule becomes smaller than the bias of *directly* evaluating the root. We have formally derived this in our revised Sec. 4.2.
>
> 2.  **Variance Reduction:** Assuming low correlation or independence among children, the weighted sum of leaf variances approaches zero as the number of nodes increases.
>
> Our overall experimental performance in both forecasting tasks and the new scientific domain experiment (Appendix D.5, detailed below) empirically supports the effectiveness of this design.
>
> As shown in **Fig. 4**, performance increases asymptotically as the tree deepens with more nodes. Further evidence is provided in **Fig. 2**, where Analytica produces a much tighter probability density function (significantly lower variance) compared to the broad, unstable distribution of a standalone agent, supporting an "inhibiting" effect on error.
>
>
> **Mitigating error propagation during synthesis**
>
> Furthermore, we analyzed potential error propagation during the synthesis process. As proven in **Proposition 1 (Sec. 4.3)**, the linear rule has **constant sensitivity** to input errors ($\frac{\partial P}{\partial C_j} = \beta_j$). This theoretical guarantee prevents errors or noise from propagating catastrophically, ensuring they are dampened rather than amplified.
>
> We explicitly modeled these errors in our **Noise Injection Experiment (Fig. 5)**. By artificially simulating grounder noise (via random perturbations), we demonstrated that the Linear rule maintains stable accuracy even as noise levels increase.

---

> > ### Author Response · Authors · 2025-11-25
> > **Response to Reviewer ebpb (Part 2/4)**
> >
> > > “The theoretical analysis relies strongly on linear decomposability and i.i.d. error assumptions, which may not hold in real-world applications.”
> >
> > **Justifying the use of a linear model**
> >
> > We clarify that employing a linear model, rather than a complex non-linear one, as the top-level prediction model is a deliberate design choice for regularization. While there are many ways to decompose a problem, a linear model forces the LLM to select the most **parsimonious** explanation.
> >
> > In the context of LLMs as modelers, this does not simply assume a "linear world." Instead, it represents an analytical framework that demands the LLM discover linear and independent components rather than fitting complex non-linear interactions. While complex factors might be easier for the LLM to explain the problem, they carry a high risk of **overfitting**, which is typically associated with model complexity and can cause high variance. Moreover, complex models are often difficult to interpret.
> >
> >
> > Enforcing a linear, independent decomposition may increase the difficulty of discovering components, but it offers superior robustness, interpretability, and a lower risk of overfitting in principle. Furthermore, the mathematical simplicity of the linear form enables our **Resynthesis** feature (**Fig. 16**). This allows users to manually adjust leaf probabilities and instantly observe the impact on the root forecast—a critical capability for human-in-the-loop decision-making that would be computationally opaque in a complex non-linear system.
> >
> >
> > **Linearity in real-world applications**
> >
> > In practice, linear and independent components are commonly applied in real-world scenarios. For example, linear structures are foundational to **factor models** (Fama & French, 2015), which are widely used in economics and finance.
> >
> >
> > **Non-linearity in our method**
> >
> > Despite the linear aggregation, our method still captures non-linear interactions in several aspects:
> >
> >
> > 1.  **Intercept as a Safety Valve:** We explicitly include a bounded intercept term ($\beta_0$) to model "bias" or "omitted interactions" error. This gives the synthesizer the freedom to account for external factors or suboptimal decomposition without breaking the model structure, effectively absorbing the "ill-posed" nature of the task into a manageable term.
> >
> > 2.  **Contextual Weight Generation:** Our linear model is not static ($y=wX$ with constant $w$). Instead, it is essentially $y=f(\text{context})X$, where $f(\text{context})$ represents the synthesizer parameterized by the LLM. Because the synthesizer accepts all child propositions as input before generating weights, non-linear interactions are implicitly modeled during the weight generation process.
> >
> >
> > **Comparing to a non-linear model**
> >
> > Finally, we treat the **Vanilla synthesis rule (Analytica-V)** in our experiments (Table 2, 3, Fig. 6), which directly prompts an LLM to output $P_{true}$ without linear constraints, as a non-linear baseline. The fact that the Linear rule outperforms the Vanilla rule empirically validates the benefits of this structural constraint, consistent with the success of linear factor-based models in the highly non-linear financial domain.
> >
> >
> > **Formal Probabilistic Grounding (New Appendix B):**
> >
> > To rigorously justify the linear assumption, we have added **Appendix B**, which derives a formal mapping between our Linear Synthesis Rule and **Bayesian Networks**. We show that our linear formulation ($\rho.p_{true} = \beta_0 + \sum \beta_j \cdot \bar{\rho}_{j}.p_{true}$) is mathematically equivalent to a specific parameterization of conditional probability distributions (CPDs), closely similar to a **Noisy-OR model** and a specific type of  Probabilistic Logic Programming (PLP) with categorical variables. This demonstrates that the linear rule is not merely a heuristic, but a tractable approximation of well-established probabilistic reasoning structures.

---

> > > ### Author Response · Authors · 2025-11-25
> > > **Response to Reviewer ebpb (Part 3/4)**
> > >
> > > > “The evaluation tasks—mainly financial market forecasting—are somewhat ill-posed. It is difficult to justify that the model can fully reason about market movements using only searchable information.”
> > >
> > > For a benchmark to be effective, score differentials should reflect underlying differences in the capacity being evaluated. We argue that financial market forecasting possesses this property, which applies largely to predictive markets as well.
> > >
> > > First, the complexity and noisy nature of financial markets make them an excellent setting for evaluating complex problem analysis. The objective here is **informational efficiency**—the ability to extract the maximum possible signal from available public data—rather than perfect clairvoyance. This framework is grounded in the *Fundamental Law of Active Management* (Grinold, 1989), which posits that excess returns (Alpha) are a product of the **Opportunity Set** (available data) and **Analysis Skill** (extraction capability). We employed multiple strategies to ensure effectiveness:
> > >
> > > 1.  **Isolating information differences:** Since we provide all baselines with the exact same "searchable information" (via the Exa.ai API), performance differences isolate **Analysis Skill**. The fact that Analytica extracts significantly more signal (**65.62%** accuracy with basic search) than the baseline (**53.94%**) proves superior information utilization, regardless of the theoretical ceiling bounded by available data. (The exception is the Jupyter Notebook agent, which was included to test the benefits of professional data analysis workflows).
> > >
> > >
> > > 2.  **Considering only major assets:** As detailed in Appendix C.2, our dataset includes only major assets (e.g., the largest S&P 500 companies, major cryptos, major Forex pairs) that have high information accessibility. For such assets, searchable information contains the critical data necessary for discretionary financial analysis, such as financial statements and key news.
> > >
> > >
> > > 3.  **Long-term predictions only:** We focus on low-frequency **long-term predictions** (1-year horizon) rather than short-term speculation. While short-term price action often resembles a random walk, long-term trends are driven by fundamental macro-factors and earnings power, which are amenable to reasoned analysis.
> > >
> > >
> > > We have included these details in Appendix C.
> > >
> > >
> > >
> > >
> > >
> > > > “Moreover, such forecasting problems often lack a clear linear structure, which challenges the assumptions underlying the proposed framework.”
> > >
> > >
> > > While financial and predictive markets exhibit non-linear dynamics, linear models remain the industry standard for robustness. Factor-based linear models (e.g., Fama-French) dominate financial economics precisely because they robustly capture signal direction with a low risk of overfitting. For long-horizon directional forecasts driven by "low-frequency" fundamental signals, linear aggregation is often more resilient than complex non-linear fits.
> > >
> > >
> > > Additionally, to address the concern regarding domain specificity, we added a **new experiment on the Matter-of-Fact (MoF) benchmark** in **Appendix D.5 (Table 7)**. This tests the framework on scientific claims (materials science), demonstrating that Analytica generalizes to domains beyond finance.
> > >
> > >
> > > **Table 7: Experiment results for scientific claims on the Matter-of-Fact benchmark**
> > >
> > > | Model | Overall Accu. | True | False | Qual. | Qnt. | Exp. | Code | Ther. | Int. | Cost (x1k) |
> > > | :--- | :--- | :--- | :--- | :--- | :--- | :--- | :--- | :--- | :--- | :--- |
> > > | **Random** | 0.50 | 0.50 | 0.50 | 0.50 | 0.50 | 0.50 | 0.50 | 0.50 | 0.50 | 0 |
> > > | **GPT-4o-mini** | 0.66 | 0.90 | 0.42 | 0.72 | 0.72 | 0.68 | 0.63 | 0.61 | 0.58 | $1 |
> > > | + Analytica-L | 0.59 | 0.87 | 0.30 | 0.55 | 0.58 | 0.59 | 0.59 | 0.60 | 0.60 | $7 |
> > > | **O4-mini** | 0.61 | 0.34 | 0.88 | 0.60 | 0.57 | 0.63 | 0.62 | 0.59 | 0.63 | $6 |
> > > | + Analytica-L | **0.64** | 0.95 | 0.30 | 0.57 | 0.60 | 0.73 | 0.62 | 0.68 | 0.66 | $38 |
> > > | **GPT-5.1** | 0.62 | 0.27 | 0.97 | 0.58 | 0.56 | 0.59 | 0.69 | 0.66 | 0.62 | $9 |
> > > | + Analytica-L | **0.70** | 0.85 | 0.55 | 0.73 | 0.75 | 0.71 | 0.67 | 0.70 | 0.64 | $78 |
> > > | **GPT-5-mini** | 0.71 | 0.60 | 0.82 | 0.69 | 0.76 | 0.66 | 0.64 | 0.75 | 0.72 | $3 |
> > > | + Analytica-L | **0.73** | 0.70 | 0.75 | 0.75 | 0.77 | 0.69 | 0.67 | 0.79 | 0.67 | $27 |

---

> ### Author Response · Authors · 2025-11-25
> **Response to Reviewer ebpb (Part 4/4)**
>
> > “The framework is highly sensitive to prompt design. A sensitivity analysis examining how performance varies under prompt perturbations would strengthen the paper’s robustness claims.”
>
>
> While prompt sensitivity is a valid concern in agentic systems, our empirical data suggests that Analytica is robust. We address this with the following evidence:
>
> 1.  **Standardized Prompts:** As detailed in **Appendix E**, we used a single, fixed set of prompts across **736 diverse tasks** (ranging from crypto to politics to stocks) as well as our evaluations for Matter-of-Fact. We did not perform per-task prompt engineering or introduce any special domain or task-specific optimizations, with only basic task or background instructions provided. Consistent performance across these disparate domains suggests the architecture, not specific phrasing, is driving results.
>
>
> 2.  **Direct Sensitivity Analysis (Fig. 5):** We explicitly tested sensitivity to input perturbations in **Fig. 5**. **This experiment simulates noise from all sources, including an erroneous prompt design**. It demonstrates that Analytica with Linear rule (-L) is exceptionally robust, maintaining performance even as noise ratios increase.
>
>
>
>
>
> > “The baseline comparisons understate stronger alternatives. Tree-, Graph-, and Forest-of-Thought baselines are evaluated only with the Basic Search grounder, whereas Analytica is paired with more powerful grounders such as Deep Research and Jupyter Notebook, making the comparison uneven.”
>
>
> We strictly controlled for grounder capability to ensure a fair comparison. We do not compare "Analytica + Jupyter" directly against "ToT + Basic Search" to claim architectural superiority. To further reduce the potential confusion, we have split Table 2 into two tables that are structured to isolate these factors:
>
> 1.  **Architecture Isolation (New Table 2, upper half of original Table 2):** We fixed the grounder to **Basic Search** for all architectures. In this strictly controlled setting, **Analytica-L (65.62%)** significantly outperforms **Tree of Thoughts (60.19%)** and **Graph of Thoughts (57.88%)**. This confirms that our SPR architecture provides superior signal extraction when the underlying tool is identical.
>
> 2.  **Grounder Ablation:** We assessed the Jupyter Grounder specifically to demonstrate its cost-effectiveness through professional tool use and to present Analytica’s adaptivity to different grounders, not to make an unfair comparison against ToT.
>
> 3.  **SOTA Augmentation (New Table 3):** We evaluated whether Analytica provides value on top of state-of-the-art, black-box agents. We show that Analytica-L with a Basic Search grounder outperforms the strong **Deep Research** baseline (65.62% vs 63.04%), and furthermore, Analytica can enhance Deep Research itself from **63.04%** to **71.06%**.
>
>
> ---
>
> We hope these clarifications have adequately addressed your concerns. Please do not hesitate to let us know if further details on any of these points would be helpful. We appreciate the opportunity to improve the paper based on this feedback.
>
>
> **Reference**
>
> [1] Grinold, R.C. (1989) The Fundamental Law of Active Management. Journal of Portfolio Management, 15, 30-37.
>
> [2] Fama, E. F., & French, K. R. (2015). A five-factor asset pricing model. Journal of financial economics, 116(1), 1-22.
>
> [3] Jansen, P., Hassan, S., & Wang, R. (2025). Matter-of-Fact: A Benchmark for Verifying the Feasibility of Literature-Supported Claims in Materials Science. EMNLP 2025.
>
> ---
>
>
> Best Regards,
>
> Authors of Submission 7713

---

### Author Response · Authors · 2025-11-25
**General Response and Summary of Revisions (Part 1/2)**

Dear Area Chair and Reviewers,

We sincerely thank the reviewers for their insightful and constructive feedback. We are encouraged that the reviewers found our **Soft Propositional Reasoning (SPR) framework** to be *"novel and conceptually appealing"* (Reviewer sApi) and *"an interesting and innovative solution"* (Reviewer ebpb). We appreciate the recognition of our **Analytica architecture** as *"practical, modular... with strong tooling"* (Reviewer ebpb) and the acknowledgment that our **Jupyter Notebook grounder** is *"well thought-out"* and achieves *"strong cost-effectiveness"* (Reviewers ebpb, kYvf). The reviewers also highlighted the *"high quality"* of our writing (Reviewer ebpb) and the *"compelling"* empirical validation showing improvements in both accuracy and efficiency (Reviewer sApi).

---

We have carefully considered the concerns raised regarding **model generality, domain transferability, theoretical rigor, and the framing of our contribution**. In response, we have significantly revised the manuscript and conducted extensive new experiments.


**Summary of Key Revisions & New Experiments:**

**1. Generalization to other backbone LLMs (Addressing Reviewers kYvf, sApi):**
To address the concern that our results relied solely on using `o3` as the main backbone LLM, we conducted a new ablation study involving **8 additional backbone LLMs**, including open-weight models (`DeepSeek-v3.1`, `Qwen`, `GLM`) and smaller distilled models (`OpenAI-OSS`, `GPT-4o-mini`).
* **Update:** Added **Appendix D.4.4** and **Table 6**.
* **Result:** Analytica consistently improves performance across all model classes. Notably, `OpenAI-OSS-20B + Analytica` achieves results comparable to the much larger `DeepSeek-v3.1` baseline, demonstrating that the framework effectively bridges the capability gap for smaller models].

**2. Domain Transferability to Science (Addressing Reviewers kYvf, sApi, ebpb):**
To demonstrate that Analytica is not limited to financial forecasting, we evaluated it on the **Matter-of-Fact (MoF)** benchmark (Jansen et al., 2025), comprising 4.4k scientific claims in materials science. Similar to our other forecasting tasks, the goal is to determine the feasibility of these claims by making a yes/no prediction, so this task naturally fits within our soft propositional reasoning framework.
* **Update:** Added **Appendix D.5** and **Table 7**.
* **Result:** The new results show significant accuracy gains (e.g., **GPT-5.1** improves from 62% to 70%), confirming that the framework's decomposition and verification architecture generalizes effectively to scientific domains.

**3. Formal Comparison & Bayesian Formulation (Addressing Reviewer sApi):**
To address concerns about novelty and the independence assumption, we have formalized the distinction between Analytica and previous structured reasoning methods, and provided a probabilistic grounding for our synthesis rule.
* **Update (Comparison with CoT/ToT/GoT):** We added a formal comparison in **Sec. 3**. We distinguish Analytica from search-based methods (ToT, GoT) that optimize a *linear path* ($R^* = \text{argmax } V(R)$) by defining Analytica as a recursive aggregation of *subproblems* ($\hat{y} = \text{Aggr}(\{\hat{y}_{C_i}\})$).
* **Update (PGM-style Semantics):** We added **Appendix B**, which maps the Analytica computation graph to an explicit **Bayesian Network** (or equivalently, a known type of probabilistic logic program (PLP) with categorical variables) . This explicitly models the linear synthesis rule as a specific parameterization of conditional probability distributions (akin to **Noisy-OR models**), providing transparency regarding the underlying independence assumptions (and in the PLP, the relational structure between sub-claims).



**4. Theoretical Rigor (Addressing Reviewers sApi, ebpb):**
We refined the theoretical sections to clearly distinguish between formal derivations and heuristic assumptions.
* **Update:** In **Sec. 4.2**, we now explicitly define the **"Simplification Hypothesis"** (that leaf propositions have lower inherent bias than the root) and derived the formal bounds ($\sum \beta'_i \delta_i < 1$) required for bias reduction to hold, rather than asserting it unconditionally.
* **Update (Formal Derivation):** We added **Appendix B** to provide the rigorous mathematical derivation of the synthesis rule. This moves the system beyond engineering heuristics by establishing its formal equivalence to **probabilistic graphical models**.

---

> ### Author Response · Authors · 2025-11-25
> **General Response and Summary of Revisions (Part 2/2)**
>
> **5. Experimental Structure & Baseline Clarification (Addressing Reviewers ebpb, sApi, Hjk1):**
> To improve clarity and ensure fair comparisons, we significantly restructured the presentation of our empirical results.
> * **Update (Explicit RQ Mapping):** We reorganized the introduction of **Sec. 5** to explicitly map our experiments to three core Research Questions derived from our theory: **RQ1** (Bias & Variance Reduction), **RQ2** (Scalability & Robustness), and **RQ3** (Cost-Effectiveness). This directly addresses the request to better connect experiments to theoretical hypotheses.
> * **Update (Baseline Isolation):** We split the original Table 2 into **Table 2** (Architecture Comparison) and **Table 3** (Grounder Ablation). This isolates architectural contributions from tool capabilities, clarifying that **Analytica outperforms Tree-of-Thoughts (ToT)** even when both are restricted to Basic Search.
>
>
>
> **6. Visual & Structural Improvements (Addressing Reviewer Hjk1):**
> * **Update:** We resolved label overlapping issues in **Fig. 6** (Efficiency Frontier) and restructured **Sec. 5** to explicitly map experiments to three core Research Questions (Performance, Robustness, Efficiency).
>
> **7. Statistical Rigor & Robustness (Addressing Reviewers kYvf, ebpb):**
> We addressed concerns regarding the statistical power of our comparisons and the testing of interaction effects.
> * **Update (McNemar’s Test):** We updated **Fig. 10** (Appendix D.1) to calculate significance on the **full dataset** (736 tasks) rather than the 100-task subset. We also explicitly added **confidence intervals** to the visualization as requested.
> * **Update (Interaction Test):** We clarified that **Fig. 5 (Noise Injection)** serves as the formal test for the interaction between *Synthesis Rule* and *Input Noise*. It empirically demonstrates that the Simple Logic rule interacts destructively with noise (performance collapses), whereas the Linear rule remains robust, supporting our theoretical claims in **Proposition 1**.
>
>
> **8. Practical Utility & Application Domain (Addressing Reviewer sApi):**
> We added a discussion to contextualize the framework's real-world significance beyond forecasting.
> * **Update:** In **Sec. 6 (Limitation and Discussion)**, we identify target use case as decision-making and analysis for complex, high-stakes, critical real-world domains, where decision-making and analysis require transparent reasoning and robustness, such as applications for economists, policymakers, scientists, and robots. We also position Analytica as a **"complex analysis backbone"** for autonomous systems that need to break down uncertain problems into calibrated, empirically testable components.
>
> ---
>
> We believe these revisions comprehensively address the reviewers' concerns and significantly strengthen the paper's empirical and theoretical foundations. Additionally, we have implemented broad improvements to the manuscript's overall writing and presentation structure, incorporating specific suggestions from Reviewers Hjk1and sApi to enhance clarity. We are grateful for the opportunity to improve our work based on this high-quality feedback.
>
> Finally, we believe Analytica offers a timely contribution to the field. By grounding flexible LLM reasoning in a rigorous **soft propositional and probabilistic framework**, our method remains **simple yet highly effective**. We hope this work pushes the foundation for **reliable, interpretable analysis** of complex problems, promoting the safe and solid adoption of agentic systems in high-stakes domains like economics, policy, scientific discovery, and autonomous systems.
>
>
>
> Best Regards,
>
> Authors of Submission 7713

---

### Author Response · Authors · 2025-12-03
**Summary of Revisions & Rebuttal Status**

Dear Area Chair,

To assist in your assessment under the significant time constraints, we provide a brief summary of our rebuttal and the current state of the paper.

**1. Key Revisions & New Experiments**
In response to reviewer feedback, we conducted extensive new experiments that significantly strengthened the paper:
* **Model Generalization (Addressing kYvf, sApi):** We evaluated **8 additional LLMs**, covering diverse categories including **open-weight models** and **small distilled models**. Results (Table 6) confirm *Analytica* consistently improves performance across all model sizes, not just the original backbone.
* **Domain Transfer (Addressing kYvf, ebpb, sApi):** We tested on the **Matter-of-Fact (MoF) benchmark**, consisting of **4.4k binary scientific claims** (e.g., materials science). Results (Table 7) show strong transferability to scientific domains, dispelling concerns that the method is limited to financial or political forecasting.
* **Theoretical Rigor (Addressing sApi, ebpb):** We added **Appendix B** to formally map our computation graph to Bayesian Networks/Probabilistic Logic Programming, providing a rigorous probabilistic grounding for our synthesis rules.

(We provide the detailed list of updates in our "General Response" below).

**2. Reviewer Consensus & Engagement**

---

* **Reviewer sApi:**
    * **Concerns:** Theoretical rigor (distinguishing heuristics from proofs), clarity of figures/tables, **model generalization** (requested evaluation on smaller LLMs), and **domain transfer**.
    * **Rebuttal:** We formalized the SPR framework as a Probabilistic Graphical Model (Appendix B), formally derived the bounds for bias reduction (Sec 4.2), and added extensive ablation studies on models and domains.
    * **PDF updates:** Added **Appendix B** (Formal PGM Proofs); Revised **Sec. 4.2** (Formal bounds for Bias Reduction & Simplification Hypothesis); Added **Table 6** (Ablation on 8+ open-weight/small models); Added **Table 7** (Matter-of-Fact scientific benchmark); Split **Tables 2 & 3** (Architecture vs. Tooling); Restructured **Sec. 5** (Explicit Research Questions); Updated **Fig. 6**.
    * **Reviewer's response:** **Upgraded assessment to acceptance.** Recognized the "well-improved" state of the paper and confirmed that concerns were thoroughly resolved.

---

* **Reviewer kYvf:**
    * **Concerns:** **Model diversity** (mostly tested on o3), **domain transferability**, statistical power, and potential data leakage.
    * **Rebuttal:** We conducted extensive ablation studies on 8 additional models, added a scientific benchmark, expanded statistical testing to the full dataset, and clarified our anti-leakage Proxy system.
    * **PDF updates:** Added **Table 6** (Ablation on 8+ models); Added **Table 7** (Scientific benchmark); Updated **Fig. 10/Appendix D.1** (Significance testing on full dataset with Confidence Intervals); Added **Appendix C.4.3** (Proxy System details).
    * **Reviewer's response:** No follow-up was posted.

---

* **Reviewer ebpb:**
    * **Concerns:** **Theoretical assumptions** (Linearity vs. Error Accumulation), **domain suitability** (ill-posed tasks), and baseline fairness.
    * **Rebuttal:** We provided a theoretical derivation showing our Linear Rule approximates Bayesian Networks (Appendix B), proved bias reduction (Sec 4.2), demonstrated empirical robustness to noise (Fig. 5), and proved domain transferability via the new scientific benchmark. We also clarified baseline setups and split **Fig. 2** to avoid potential confusion.
    * **PDF updates:** Added **Appendix B** (Theoretical Derivation); Revised **Sec. 4.2** (Bias Reduction); Added **Table 7** (Scientific benchmark); Split **Tables 2 & 3** (Separating architecture from tool benchmarks).
    * **Reviewer's response:** No follow-up was posted.

---

* **Reviewer Hjk1:**
    * **Concerns:** Dependency on decomposition quality and figure readability.
    * **Rebuttal:** We demonstrated robustness via noise injection tests and showed that even **small/weak models** (from our model generalization study) function effectively as decomposers, proving architectural robustness.
    * **PDF updates:** Added **Table 6** (Evidence from open-weight/small models); Updated **Fig. 6** (Readability); Added **Appendix D.6** (Detailed Error Analysis).
    * **Reviewer's response:** Maintained positive assessment, noting appreciation for the clarifications.

---

**3. Conclusion**
The revised manuscript now includes robust evidence of generalization across models (open-weights) and domains (science), backed by a formalized theoretical framework. We believe *Analytica* offers a timely, rigorous solution for reliable, stable, and scalable LLM-driven analysis.

---

Thank you for your time and consideration.

Best Regards,

Authors of Submission 7713

---

### Meta-Review · Area_Chair_jLsc · 2026-01-06

**Summary:**

1) Reviewers criticized to a large degree the limited experimental evaluation and/or shortcomings in the experimental evaluation.

2) Reviewer sApi and Reviewer ebpb were also concerned about the theoretical rigor and some of the theoretical assumptions.

**Reviewer Concerns:**

1) The authors of the paper put a lot of effort into addressing all the concern about experimental breadth and ran extra experiments based on suggestions from the reviewers. These experiments did not change the interpretation of the results but made the paper stronger. Also the authors added more thorough statistical analysis where suggested.

2) In my opinion the concern about theoretical issues have been addressed adequately by the authors. Mainly by providing more formal descriptions in the Appendix.

**Reviewer Scores:**

Reviewer ebpb had not engaged with the rebuttal process but most of their concerns were addressed in a thorough rebuttal. I deem them to have changed their score more likely than not, i.e. 4->6

Reviewer Hjk1 write in a comment they would not change their score (6)

Reviewer kYvf was already positive (6) I don't think they would have changed their score.

Reviewer sApi wrote a comment that they would increase their score, ie. 4->6

---

### Decision · Program_Chairs · 2026-01-26

Accept (Poster)